# Best-of-Both-Worlds for Heavy-Tailed Markov Decision Processes

Yu Chen [1]   Yuhao Liu [1]   Jiatai Huang [2]   Yihan Du [3]   Longbo Huang [1]

## Abstract

We investigate episodic Markov Decision Processes with heavy-tailed losses (HTMDPs). Existing approaches for HTMDPs are conservative in stochastic environments and lack adaptivity in adversarial regimes. In this work, we propose algorithms `HT-FTRL-OM` and `HT-FTRL-UOB` for HTMDPs that achieve Best-of-Both-Worlds (BoBW) guarantees: instance-independent regret in adversarial environments and logarithmic instance-dependent regret in self-bounding (including the stochastic case) environments. For the known transition setting, `HT-FTRL-OM` applies the Follow-The-Regularized-Leader (FTRL) framework over occupancy measures with novel skipping loss estimators, achieving a $\widetilde{\mathcal{O}}(T^{1/\alpha})$ regret bound in adversarial regimes and a $\mathcal{O}(\log T)$ regret in stochastic regimes. Building upon this framework, we develop a novel algorithm `HT-FTRL-UOB` to tackle the more challenging unknown-transition setting. Under a mild truncative nonnegativity condition on the loss distributions, this algorithm employs a pessimistic skipping loss estimator and achieves a $\widetilde{\mathcal{O}}(T^{1/\alpha} + \sqrt{T})$ regret in adversarial regimes and a $\mathcal{O}(\log^2(T))$ regret in stochastic regimes. Our analysis overcomes key barriers through several technical insights, including a local control mechanism for heavy-tailed shifted losses, a new suboptimal-mass propagation principle, and a novel regret decomposition that isolates transition uncertainty from heavy-tailed estimation errors and skipping bias.

[1]Institute for Interdisciplinary Information Sciences, Tsinghua University, Beijing, China [2]Beijing ZhuoShi Capital Co., Ltd., China [3]ESD, Singapore University of Technology and Design, Singapore. Correspondence to: Longbo Huang <longbohuang@tsinghua.edu.cn>.

*Proceedings of the 43rd International Conference on Machine Learning*, Seoul, South Korea. PMLR 306, 2026. Copyright 2026 by the author(s).

## 1. Introduction

Markov Decision Process (MDP) is a fundamental framework for modeling sequential decision-making problems, underpinning modern advancements in Reinforcement Learning (RL), operations research, and control systems. Standard RL theory often relies on the assumption that feedback signals (losses or rewards) are bounded or sub-Gaussian, enabling the use of classical concentration inequalities. While convenient for theoretical analysis, this assumption can be violated in real-world scenarios where the feedback is inherently heavy-tailed. In domains such as network traffic routing (Liebeherr et al., 2012), financial asset pricing (Bradley & Taqqu, 2003; Bai et al., 2025), and image signal processing (Hamza & Krim, 2001), agents frequently encounter extreme events or outliers with non-negligible probability, underscoring the importance of developing robust RL algorithms to handle MDPs with heavy-tailed losses.

Beyond modeling the tail behavior of loss distributions, robust algorithm design must also account for the underlying nature of the environment. This line of research distinguishes between distinct environment types, typically categorized into stochastic and adversarial regimes. In the stochastic regime, losses are generated from stationary distributions, while in the adversarial regime, losses may be arbitrarily chosen by an adversary and may vary over time. Since the nature of the environment is often unknown *a priori*, a desirable property is *Best-of-Both-Worlds* (BoBW), as proposed by (Bubeck & Slivkins, 2012) for the multi-armed bandit (MAB) problem. Ideally, without knowing the regime in advance, a BoBW algorithm should simultaneously achieve an optimal instance-dependent regret in stochastic environments (typically logarithmic in $T$) and an instance-independent regret in adversarial environments.

In this paper, we investigate the *Markov Decision Processes with Heavy-Tailed losses* (HTMDPs) under various environmental conditions, including stochastic and adversarial regimes. Unlike standard settings with bounded feedback, the loss $\ell$ that occurs in HTMDPs may be unbounded and can take negative values, and we work under the bounded $\alpha$-moment condition ($\alpha \in (1, 2]$), i.e., $\mathbb{E}[|\ell|^\alpha] \leq \sigma^\alpha$ for some $\sigma > 0$ (with a mild truncative nonnegativity condition additionally required only for the unknown transition case;

see Assumption 5.1).

Although there is a large body of work studying HTMDPs in tabular and function approximation settings (Zhuang & Sui, 2021; Park & Lee, 2024; Zhu et al., 2024; Huang et al., 2023; 2024; Li & Sun, 2024), existing results almost exclusively focus on stochastic environments and primarily provide instance-independent regret guarantees, without providing instance-dependent regret bounds of logarithmic order in the number of episodes $T$ (i.e., $\mathcal{O}(\log T)$). Consequently, it remains a crucial problem to design BoBW algorithms for HTMDPs which, without knowing the regime *a priori*, simultaneously attain an instance-independent regret minimax-optimal in $\sigma, T$ up to $\text{poly}(H, S, A)$ and log factors in adversarial regimes and logarithmic instance-dependent regret in stochastic regimes.

**Challenges.** Tackling HTMDPs in both adversarial and stochastic environments presents notable obstacles. *i)* Existing BoBW algorithms for MDPs critically depend on concentration arguments under boundedness or sub-Gaussian assumptions to control stability terms and second-order regret components, but these tools break down under heavy-tailed feedback. *ii)* Existing algorithms for HTMDPs are largely developed for stationary stochastic environments and their analyses exploit such structure to control heavy-tailed estimation errors. These guarantees do not extend to adversarially varying losses, and thus cannot be directly used when the regime is unknown *a priori*. *iii)* Unlike bandits, MDPs introduce long-horizon dependence through state transitions, causing heavy-tailed noise to accumulate in value estimation. Moreover, the state-action structure implies huge challenges in the optimal-action elimination procedure, which is the crucial step for achieving a logarithmic stochastic regret bound (see Section 4.2). This leads to the following challenging and important open question:

### *Can BoBW performance be achieved under HTMDPs?*

**Our Contributions.** In this paper, we answer this question in the affirmative. We start with HTMDPs with known transitions. Our proposed algorithm **H**eavy-**T**ailed **FTRL** over **O**ccupancy **M**easure (`HT-FTRL-OM`) integrates the Follow-The-Regularized-Leader (FTRL) framework over occupancy measures with a $1/\alpha$-Tsallis entropy regularizer and a novel skipping loss estimator (see Section 4.1), and achieves BoBW guarantees simultaneously in adversarial regimes and self-bounding (including stochastic as a special case) regimes. Leveraging this framework, we further investigate the unknown-transition setting and develop algorithm **H**eavy-**T**ailed **FTRL** with **U**pper **O**ccupancy **B**ound (`HT-FTRL-UOB`) in Section 5.1, which adopts a pessimistic estimator with skipping-loss adjustment and biased importance sampling via upper occupancy bounds. We establish instance-dependent regret bounds for `HT-FTRL-UOB` in self-bounding regimes and instance-independent regret

bounds in adversarial regimes, which demonstrate the BoBW performance. Our main contributions are summarized as follows:

- For HTMDPs with known transitions, we propose framework `HT-FTRL-OM` (Algorithm 1), which simultaneously achieves an instance-independent regret bound $\widetilde{\mathcal{O}}(\sigma T^{1/\alpha})$, minimax-optimal in $\sigma, T$ up to $\text{poly}(H, S, A)$ and log factors, in adversarial regimes and an instance-dependent regret bound $\mathcal{O}(\log(T))$ in stochastic regimes (see Theorem 4.1).

- We extend this framework to the unknown transition setting and devise algorithm `HT-FTRL-UOB` (Algorithm 2), which, under a mild truncative nonnegativity condition on the loss distributions (Assumption 5.1), achieves an instance-independent regret bound $\widetilde{\mathcal{O}}(\sigma(T^{1/\alpha} + \sqrt{T}))$, minimax-optimal in $\sigma, T$ up to $\text{poly}(H, S, A)$ and log factors, in adversarial regimes and the first logarithmic instance-dependent regret bound $\mathcal{O}(\log(T)^2)$ for stochastic environments (see Theorem 5.2).

- Our analysis overcomes key barriers in heavy-tailed RL. *i)* We develop a novel refinement framework for shifted losses under heavy-tailed feedback. *ii)* We establish a suboptimal mass propagation principle that bridges occupancy deviations across layers. *iii)* For the unknown transition setting, we design a pessimistic estimator and derive an innovative regret decomposition scheme that carefully isolates transition uncertainty from heavy-tailed estimation errors and the skipping bias. See Sections 4.2 and 5.2 for more details.

## 2. Related Works

**Heavy-tailed Online Decision Making.** Heavy-tailed feedback is well-studied in bandits via robust mean estimation (Bubeck et al., 2013; Medina & Yang, 2016; Shao et al., 2018; Xue et al., 2021; Wang et al., 2025; Lu et al., 2019; Ray Chowdhury & Gopalan, 2019). In episodic MDPs, Zhuang & Sui (2021) established minimax regret bound $\widetilde{\mathcal{O}}\left(T^{1/\alpha} + \sqrt{T}\right)$ in stochastic environments, illustrating that heavy-tailed feedback introduces significant costs in the worst-case regret. Subsequent works extended this to linear MDPs (Huang et al., 2023) and variance-aware guarantees (Huang et al., 2023; Li & Sun, 2024). Other directions include robust offline RL and differential privacy (Park & Lee, 2024; Zhu et al., 2024; Yang et al., 2024; Wu et al., 2023). Crucially, existing HTMDP results focus almost exclusively on stochastic environments, lacking adaptivity to adversarial regimes (BoBW) and often yielding polynomial rather than logarithmic regret in stochastic settings.

**BoBW Heavy-Tailed MABs.** The BoBW properties for heavy-tailed online learning have only been studied in Multi-

Armed Bandits (MABs). Huang et al. (2022) gave the first BoBW algorithms for heavy-tailed MABs by incorporating a truncative nonnegativity assumption to control stability under negative losses. Cheng et al. (2024) further strengthened the guarantees by removing the non-negativity assumption on loss distribution and providing high-probability bounds. Zhao et al. (2025) extended the BoBW guarantees to heavy-tailed linear bandits and broader adversarial heavy-tailed bandit models. Chen et al. (2025) further considered parameter-free heavy-tailed MABs that do not require prior knowledge of tail parameters $(\sigma, \alpha)$. In contrast to bandits, our setting is an episodic HTMDP where feedback is coupled through state visitation and transition dynamics, requiring new robust estimators and stability control under occupancy constraints.

**BoBW MDPs.** For standard MDPs with bounded losses, BoBW guarantees are typically achieved via FTRL over occupancy measures. Jin & Luo (2020) established the first result for known transitions, which Jin et al. (2021) extended to unknown transitions. Further research explores adversarial transitions (Jin et al., 2023; Ito et al., 2025), complementary policy optimization (Dann et al., 2023), and linear function approximation (Liu et al., 2023). However, all these results fundamentally rely on bounded or sub-Gaussian feedback. We address the open problem of achieving BoBW guarantees in episodic MDPs under heavy-tailed feedback.

## 3. Preliminaries

**Notations.** We write $[n] := \{1, \ldots, n\}$ for $n \geq 1$, and let $\Delta(\mathcal{X})$ denote the probability simplex over a finite set $\mathcal{X}$. We use $\mathcal{O}$ and $\Omega$ for upper and lower bounds up to constants, and $\widetilde{\mathcal{O}}$ to further hide logarithmic factors. $\mathrm{poly}(\cdot)$ denotes an unspecified polynomial. $\log(\cdot)$ is the natural logarithm unless stated otherwise. Vectors are denoted by bold symbols (e.g., $\boldsymbol{x}$) with entries $x_i$. Finally, $\{\mathcal{F}_t\}_{t=0}^{T}$ is the natural filtration, where $\mathcal{F}_t$ is the $\sigma$-algebra generated by all observations up to episode $t$.

### 3.1. Episodic MDPs with Heavy-Tailed Losses

We consider a tabular and episodic finite-horizon Markov decision process (MDP), denoted by $\mathcal{M} = (\mathcal{S} \cup \{s_{H+1}\}, \mathcal{A}, \mathbb{P}, H, T, \{\boldsymbol{\nu}_t\}_{t=1}^{T})$. Here $\mathcal{S}$ and $\mathcal{A}$ are the state space and action space, respectively. $\mathbb{P} : \mathcal{S} \times \mathcal{A} \to \Delta(\mathcal{S} \cup \{s_{H+1}\})$ is the transition function where $s_{H+1}$ represents the terminal state. $H$ is the horizon length. $T$ is the number of episodes. $\boldsymbol{\nu}_t = \{\nu_t(s, a)\}_{s \in \mathcal{S}, a \in \mathcal{A}}$ is the loss distribution for each episode $t \in [T]$. We use $S$ and $A$ to represent the numbers of states and actions, respectively.

Without loss of generality, we assume that the state space $\mathcal{S}$ can be partitioned into $H$ disjoint layers $\mathcal{S} = \bigsqcup_{h=1}^{H} \mathcal{S}_h$, where $\mathcal{S}_1 = \{s_1\}$ consists of only a fixed initial state. The

transition can only happen between adjacent layers, i.e., for any action $a \in \mathcal{A}$, $\mathbb{P}[\cdot|s_h, a]$ is supported on $\mathcal{S}_{h+1}$ for all $s_h \in \mathcal{S}_h$. Moreover, $\mathbb{P}[s_{H+1} \mid s_H, a] = 1$ for all $s_H \in \mathcal{S}_H$.

The learner interacts with the MDP for $T$ episodes. In the beginning of each episode $t \in [T]$, the environment chooses a loss distribution $\nu_t(s, a)$ for each state-action pair $(s, a)$.[1] In each episode $t$, the learner chooses a policy $\pi_t : \mathcal{S} \to \Delta(\mathcal{A})$. The interaction proceeds as follows: starting in an initial state $s_1^t = s_1$, for each step $h \in [H]$, the learner observes state $s_h^t$, samples action $a_h^t \sim \pi_t(\cdot|s_h^t)$, observes loss $\ell_t(s_h^t, a_h^t) \sim \nu_t(s_h^t, a_h^t)$, and the state transitions to $s_{h+1}^t \sim \mathbb{P}[\cdot|s_h^t, a_h^t]$.

**Heavy-Tailed Losses.** We consider the heavy-tailed loss distributions whose variance may be unbounded. Following previous works (Bubeck et al., 2013; Zhuang & Sui, 2021), we assume that the loss distributions have a bounded $\alpha$-moment. Specifically, there exist constants $\alpha \in (1, 2]$ and $\sigma > 0$ such that for all $t \in [T]$ and $(s, a) \in \mathcal{S} \times \mathcal{A}$, $\mathbb{E}_{\ell \sim \nu_t(s,a)}[|\ell|^\alpha] \leq \sigma^\alpha$. This assumption covers both the finite variance case ($\alpha = 2$) and infinite variance case ($1 < \alpha < 2$). For the known transition setting (Section 4), our analysis imposes no further sign restriction and the loss is allowed to be negative. For the unknown transition setting (Section 5), we additionally introduce a mild truncative non-negativity condition (Assumption 5.1), which still permits negative losses but ensures that the truncated expectation remains nonnegative; this extra requirement is needed to control the sign of the skipping bias under transition uncertainty.

**Regret Metric.** We first define the value and Q-value functions for policy $\pi$ and transition kernel $\mathbb{P}$. Given loss functions $\boldsymbol{\ell} = \{\ell(s, a)\}_{s \in \mathcal{S}, a \in \mathcal{A}}$, for any step $h \in [H]$ and $(s_h, a) \in \mathcal{S}_h \times \mathcal{A}$, we define

$$Q^{\mathbb{P},\pi}(s_h, a; \boldsymbol{\ell}) := \ell(s_h, a) + \sum_{s' \in \mathcal{S}_{h+1}} \mathbb{P}[s'|s_h, a] V^{\mathbb{P},\pi}(s'; \boldsymbol{\ell}),$$

$$V^{\mathbb{P},\pi}(s_h; \boldsymbol{\ell}) := \sum_{a \in \mathcal{A}} \pi(a|s_h) Q^{\mathbb{P},\pi}(s_h, a; \boldsymbol{\ell}),$$

with $Q^{\mathbb{P},\pi}(s_{H+1}, a; \boldsymbol{\ell}) := 0$ and $V^{\mathbb{P},\pi}(s_{H+1}; \boldsymbol{\ell}) := 0$. Then, $\mathbb{E}[V^{\mathbb{P},\pi_t}(s_1; \boldsymbol{\ell}_t)]$ represents the expected loss of policy $\pi_t$ in episode $t$, where the expectation is taken over the randomness of the loss distributions $\boldsymbol{\ell}_t \sim \boldsymbol{\nu}_t$. The performance of the learner is measured by the regret against a fixed deterministic benchmark policy $\mathring{\pi}$:

$$\mathrm{Reg}_T(\mathring{\pi}) = \mathbb{E}\left[\sum_{t=1}^{T} \left(V^{\mathbb{P},\pi_t}(s_1; \boldsymbol{\ell}_t) - V^{\mathbb{P},\mathring{\pi}}(s_1; \boldsymbol{\ell}_t)\right)\right].$$

We choose $\mathring{\pi}$ to be the optimal deterministic policy that minimizes the expected cumulative loss $\mathbb{E}\left[\sum_{t=1}^{T} V^{\mathbb{P},\mathring{\pi}}(s_1; \boldsymbol{\ell}_t)\right]$.

---

[1] $\nu_t(s, a)$ can be fixed in stochastic regimes, or time-varying and history-dependent in adversarial regimes.

**Occupancy Measure.** We further introduce the state-action occupancy measure $\boldsymbol{\rho}^{\mathbb{P},\pi}$, given transition kernel $\mathbb{P}$ and policy $\pi$. In step $h \in [H]$, we denote the probability of visiting state-action pair $(s_h, a)$ with policy $\pi$ under $\mathbb{P}$ as $\rho^{\mathbb{P},\pi}(s_h, a)$ for any $s_h \in \mathcal{S}_h$ and $a \in \mathcal{A}$. Therefore, we can write $J_t^{\mathbb{P}}(\pi) = \mathbb{E}[\sum_{s \in \mathcal{S}, a \in \mathcal{A}} \rho^{\mathbb{P},\pi}(s, a)\ell_t(s, a)]$. Moreover, for any fixed $\mathbb{P}$, we define the occupancy measure set $\mathcal{Q}(\mathbb{P})$ as

$$\mathcal{Q}(\mathbb{P}) := \left\{ \boldsymbol{\rho} \geq 0 : \exists \pi \ \text{s.t.} \ \boldsymbol{\rho} = \boldsymbol{\rho}^{\mathbb{P},\pi} \right\}, \tag{1}$$

which is a convex polytope characterized by linear flow constraints (Jin et al., 2020; 2021). For each feasible occupancy measure $\boldsymbol{\rho} \in \mathcal{Q}(\mathbb{P})$, we denote $\rho(s) = \sum_{a \in \mathcal{A}} \rho(s, a)$. Then it corresponds to a unique policy $\pi$ under $\mathbb{P}$, which can be derived by $\pi(a|s) = \rho(s, a)/\sum_{a' \in \mathcal{A}} \rho(s, a').$[2]

### 3.2. Adversarial and Self-Bounding Regimes.

In this section, we introduce the regimes considered in this paper. Note that our algorithms do not have any prior knowledge of regime types.

**Adversarial Regime.** In the adversarial setting, the environment may choose $\boldsymbol{\nu}_t$ adaptively: for each episode $t$, the loss distributions $\boldsymbol{\nu}_t$ are allowed to be $\mathcal{F}_{t-1}$-measurable.

**Self-Bounding Regime.** To model benign (e.g., stochastic) or nearly benign environments, we adopt the following self-bounding condition, which is standard in the BoBW literature (Jin et al., 2021; Zimmert & Seldin, 2021; Zhao et al., 2025).

**Definition 3.1** (Self-Bounding Constraint). An environment satisfies the $(\mathring{\pi}, \Delta, C_{\mathrm{sb}})$-self-bounding constraint if there exists a gap function $\Delta : \mathcal{S} \times \mathcal{A} \to \mathbb{R}_{\geq 0}$ with $\Delta(s, \mathring{\pi}(s)) = 0$ for all $s \in \mathcal{S}$ and a constant $C_{\mathrm{sb}} \geq 0$, such that the regret against $\mathring{\pi}$ is lower bounded by the cumulative expected gap:

$$\mathrm{Reg}_T(\mathring{\pi}) \geq \mathbb{E}\left[\sum_{t=1}^{T} \sum_{s \in \mathcal{S}, a \in \mathcal{A}} \rho^{\mathbb{P},\pi_t}(s, a)\Delta(s, a)\right] - C_{\mathrm{sb}}.$$

Intuitively, the above inequality states that up to an additive constant, the regret must be paid for sampling suboptimal state-action pairs, quantified by the gap function $\Delta$. The (i.i.d.) stochastic regime, where $\boldsymbol{\nu}_t \equiv \boldsymbol{\nu}$ and losses are drawn independently across episodes, satisfies this condition with $C_{\mathrm{sb}} = 0$ by taking $\mathring{\pi}$ as an optimal policy. Moreover, the stochastic regime with adversarial corruptions (with total corruption budget $C_{\mathrm{corr}}$) also falls into this framework (Lykouris et al., 2018), and one can take $C_{\mathrm{sb}} = \mathcal{O}(C_{\mathrm{corr}})$ with the same gap function.

---

[2]When $\rho(s) = 0$, the state $s$ is never visited under $\pi$ and $\pi(\cdot \mid s)$ can be chosen arbitrarily without affecting $\boldsymbol{\rho}^{\mathbb{P},\pi}$.

## 4. HTMDPs with Known Transition: Challenges and Core Technical Insights

In this section, we start by considering the MDPs with known transition kernel $\mathbb{P}$, which allows us to focus on analyzing how to deal with the heavy-tailed losses in the MDPs. We first present a novel algorithm **H**eavy-**T**ailed **FTRL** over **O**ccupancy **M**easure (`HT-FTRL-OM`, Algorithm 1) which achieves the Best-of-Both-Worlds (BoBW) guarantees in adversarial and self-bounding (including the stochastic regime) environments. Then we highlight the core technical challenges and insights in handling heavy-tailed losses in MDPs over occupancy measures.

### 4.1. Algorithm and Results with Known Transition

We introduce algorithm `HT-FTRL-OM` (Algorithm 1) for HTMDPs with known transitions. Our approach is built upon the FTRL framework over the polytope $\mathcal{Q}(\mathbb{P})$ equipped with a $1/\alpha$-Tsallis entropy regularizer. Crucially, we design a novel skipping loss estimator tailored for occupancy measures. While skipping techniques have been explored in heavy-tailed bandits (Huang et al., 2022; Zhao et al., 2025), our design marks the first successful adaptation of these mechanisms to HTMDPs, enabling robust estimation in the presence of state-transition structures.

In Line 2 of Algorithm 1, we compute the occupancy measure $\boldsymbol{x}_t$ by solving the FTRL problem over the occupancy measure polytope $\mathcal{Q}(\mathbb{P})$ with $1/\alpha$-Tsallis entropy regularizer $\Psi_t(\boldsymbol{x})$. This is a convex optimization problem that can be solved in polynomial time. Then we use the occupancy measure $\boldsymbol{x}_t$ to derive the policy $\pi_t$ in Line 3. In Line 4, we roll out the episode $t$ using the policy $\pi_t$ and observe the trajectory and losses. Line 5 calculates the novel skipping loss estimator $\widetilde{\ell}_t^{\mathrm{skip}}(s, a)$ by tuning the losses with a skipping trick, subtracting a skipping bonus, and importance sampling. Line 6 computes the skipping bonus $b_t^{\mathrm{skip}}(s, a)$ to compensate for the bias and skipping error.

Our novel analysis illustrates that `HT-FTRL-OM` achieves the BoBW regret bounds in the adversarial and self-bounding (including the stochastic regime) environments, as shown by the following theorem.

**Theorem 4.1** (BoBW Guarantees for the Known Transition Setting). *By setting constant $C$ and $\beta$ in Eqs. (17) and (18), `HT-FTRL-OM` (Algorithm 1) achieves BoBW regret bounds in the adversarial and self-bounding regimes simultaneously:*

*1. Adversarial Regime: For any adversarial loss distributions and deterministic benchmark policy $\mathring{\pi}$,*

$$\mathrm{Reg}_T(\mathring{\pi}) = \mathcal{O}\left(\mathrm{poly}(H, S, A)\sigma(T^{1/\alpha} + \log(T))\right).$$

*2. Self-Bounding Regime: If the environment satisfies the*

**Algorithm 1** HT-FTRL-OM

**Input:** Heavy-tail parameters $\alpha$ and $\sigma$, skipping threshold constant $C > 0$, and learning rate constant $\beta > 0$.

1: **for** $t = 1, 2, \ldots, T$ **do**
2:    Compute occupancy distribution

$$\boldsymbol{x}_t \leftarrow \operatorname*{argmin}_{\boldsymbol{x} \in \mathcal{Q}(\mathbb{P})} \left\langle \boldsymbol{x}, \sum_{t' < t} \left( \widetilde{\boldsymbol{\ell}}_{t'}^{\text{skip}} - \boldsymbol{b}_{t'}^{\text{skip}} \right) \right\rangle + \Psi_t(\boldsymbol{x}),$$
(2)

where $\mathcal{Q}(\mathbb{P})$ is the occupancy measure defined in Eq. (1) and $\Psi_t(\boldsymbol{x}) = -\frac{1}{\eta_t} \sum_{(s,a) \in \mathcal{S} \times \mathcal{A}} x(s,a)^{\frac{1}{\alpha}}$ is the $1/\alpha$-Tsallis entropy regularizer with inverse learning rate $\eta_t := \beta/(\sigma \cdot t^{1/\alpha})$.
3:    Derive policy $\pi_t(a|s) \leftarrow x_t(s,a)/\sum_{a' \in \mathcal{A}} x_t(s,a')$.
4:    Roll out episode $t$ using $\pi_t$, and observe trajectory $\{s_h^t, a_h^t\}_{h \in [H]}$ and losses $\ell_t(s_h^t, a_h^t)$.
5:    For each $(s,a)$, construct the importance sampling estimator:

$$\widetilde{\ell}_t^{\text{skip}}(s,a) \leftarrow \frac{\ell_t^{\text{skip}}(s,a)}{x_t(s,a)} \cdot \mathbb{1}_t[(s,a)].$$
(3)

Here $\ell_t^{\text{skip}}(s,a)$ is the skipped loss with $\ell_t^{\text{skip}}(s,a) = \ell_t(s,a) \cdot \mathbb{1}[|\ell_t(s,a)| \leq \tau_t(s,a)]$ and $\tau_t(s,a) \leftarrow C \cdot \sigma t^{1/\alpha} x_t(s,a)^{1/\alpha}$. $\mathbb{1}_t[(s,a)]$ takes the value 1 if $(s,a) \in \{(s_h^t, a_h^t)\}_{h=1}^H$ and 0 otherwise.
6:    Compute skipping-bonus $b_t^{\text{skip}}(s,a) \leftarrow C^{1-\alpha} \cdot \sigma t^{1/\alpha-1} x_t(s,a)^{1/\alpha-1}$.
7: **end for**

---

$(\mathring{\pi}, \Delta, C_{\text{sb}})$ *self-bounding constraint (Definition 3.1), then we have*

$$\text{Reg}_T(\mathring{\pi}) = \mathcal{O}\left( \text{poly}(H, S, A) \left( W + C_{\text{sb}}^{\frac{1}{\alpha}} \cdot W^{\frac{\alpha-1}{\alpha}} + U \right) \right),$$

*where we denote* $W := \sigma^{\frac{\alpha}{\alpha-1}} \omega_\Delta(\alpha) \log(T)$ *with* $\omega_\Delta(\alpha) := \sum_{(s,a): a \neq \mathring{\pi}(s)} \Delta(s,a)^{-\frac{1}{\alpha-1}}$, *and* $U := \sigma \log(T)$.

To the best of our knowledge, Theorem 4.1 shows that HT-FTRL-OM is the first algorithm that achieves BoBW guarantees in the Heavy-Tailed MDPs with known transition. We refer readers to Theorem B.1 for the formal theorem, with detailed proof in Appendix B.

Theorem 4.1 demonstrates that HT-FTRL-OM achieves an instance-independent regret bound of $\mathcal{O}(\sigma T^{1/\alpha})$ in adversarial environments, and an instance-dependent regret bound of $\mathcal{O}(\sigma^{\frac{\alpha}{\alpha-1}} \omega_\Delta(\alpha) \log(T) + C_{\text{sb}}^{\frac{1}{\alpha}} \cdot \sigma \omega_\Delta(\alpha)^{\frac{\alpha-1}{\alpha}} \log(T)^{\frac{\alpha-1}{\alpha}})$ in self-bounding regimes, which implies a regret of $\mathcal{O}(\sigma^{\frac{\alpha}{\alpha-1}} \omega_\Delta(\alpha) \log(T))$ in stochastic regimes.

*Remark* 4.2 (Tightness via MAB Reduction). When $H = 1$, $|\mathcal{S}| = 1$, and $|\mathcal{A}| = A$, HTMDPs reduce to heavy-tailed MABs with $A$ arms. Bubeck et al. (2013) established the

instance-independent lower bound $\Omega(\sigma T^{1/\alpha})$ in adversarial environments and the instance-dependent lower bound $\Omega\left(\sigma^{\alpha/(\alpha-1)} \sum_{i \neq i^*} \Delta_i^{-1/(\alpha-1)} \log T\right)$ in stochastic environments. The regret bounds of HT-FTRL-OM in Theorem 4.1 match these lower bounds in $\sigma$, $T$, and the suboptimal gaps $\Delta$, up to $\text{poly}(H, S, A)$ and log factors. In the sub-Gaussian case ($\alpha = 2$), our bounds also recover the results of Jin & Luo (2020); Jin et al. (2021) up to $\text{poly}(H, S, A)$ factors.

## 4.2. Challenges and Core Technical Insights

Our results are accomplished by integrating the FTRL framework over occupancy measures and self-bounding techniques (Luo et al., 2021; Zimmert & Seldin, 2019; Jin et al., 2020) with the loss tuning technique in heavy-tailed bandits with Tsallis entropy regularizer (Huang et al., 2022; Zhao et al., 2025). However, extending these methods to heavy-tailed MDPs faces significant challenges due to the interaction between the state-action structure of occupancy measures and the heavy-tailed nature of the losses. Below we provide a roadmap of our analysis and highlight the core technical insights.

Following the standard FTRL analysis, we decompose the regret into three terms: (i) a *stability term* that captures the stability of the FTRL update, (ii) a *penalty term* that arises from the time-varying regularizer $\Psi_t$, and (iii) a *skipping error* that accounts for the bias introduced by loss truncation.

The central challenge is to bound these three terms while achieving both the adversarial rate $\mathcal{O}(T^{1/\alpha})$ and self-bounding rate $\mathcal{O}(\log T)$. To address these challenges, we develop two core technical contributions: (1) a novel *heavy-tailed loss shifting* analysis for bounding the stability term under heavy-tailed noise, and (2) a *suboptimal mass propagation* lemma for bounding the penalty and skipping terms by restricting to suboptimal actions only.

**Heavy-Tailed Loss Shifting.** Loss shifting is a powerful technique proposed by (Jin et al., 2021) which considers a shifted loss function

$$\ell_t^{\text{shift}}(s,a) := Q^{\mathbb{P}, \pi_t}(s, a; \boldsymbol{\ell}) - V^{\mathbb{P}, \pi_t}(s; \boldsymbol{\ell}),$$

instead of the original loss $\ell_t(s,a)$ as an invariant transformation to maintain the FTRL update rule: $\operatorname{argmin}_{\boldsymbol{x} \in \mathcal{Q}(\mathbb{P})} \langle \boldsymbol{x}, \sum_{t' < t} \boldsymbol{\ell}_{t'}^{\text{shift}} \rangle + \Psi_t(\boldsymbol{x})$ is exactly the same as $\operatorname{argmin}_{\boldsymbol{x} \in \mathcal{Q}(\mathbb{P})} \langle \boldsymbol{x}, \sum_{t' < t} \boldsymbol{\ell}_{t'} \rangle + \Psi_t(\boldsymbol{x})$. This shifting enables lower variance control over the shifted loss functions and simplifies the analysis of the FTRL update over occupancy measures by controlling the second-moment of the shifted loss to a probability term locally (e.g., $(1 - \pi_t(a|s))/x_t(s,a)$ in Jin et al. (2021)).

However, standard analyses for shifted losses (Jin et al., 2021; Ito et al., 2025) heavily rely on the properties of

bounded losses to establish the pointwise and second-moment control of losses, which fails under heavy-tailed noise. Moreover, simply combining the skipping loss tuning with the loss shifting technique (Huang et al., 2022) does not solve the problem. Since each skipping loss $\ell_t^{\text{skip}}(s,a)$ is bounded by $\tau_t(s,a) = \mathcal{O}(t^{1/\alpha} x_t(s,a)^{1/\alpha})$, the shifted skipping loss, which sums up the skipping loss in "future" steps, introduces significant difficulties in producing the pointwise uniform bound and second-moment control in local state-action pairs $(s,a)$.

Our novel analysis presents a new technique in deriving a *local* control of the shifted loss $\widetilde{\ell}_t^{\text{shift}}(s,a)$, which is defined as the advantage function of skipping loss $\widetilde{\ell}_t^{\text{skip}}(s,a)$:

$$\widetilde{\ell}_t^{\text{shift}}(s,a) := Q^{\mathbb{P},\pi_t}(s,a;\widetilde{\ell}_t^{\text{skip}}) - V^{\mathbb{P},\pi_t}(s;\widetilde{\ell}_t^{\text{skip}}).$$

To be specific, we decompose the shifted loss into a centered loss term and a differential value function term. Then, by careful calculation of the reaching probability, we can derive the pointwise uniform bound:

**Lemma 4.3** (Pointwise Uniform Bound of Shifted Loss). *For any* $(s,a) \in \mathcal{S} \times \mathcal{A}$, *we have* $|\widetilde{\ell}_t^{\text{shift}}(s,a)| \leq \text{poly}(H,S,A)\sigma t^{\frac{1}{\alpha}} x_t(s,a)^{\frac{1}{\alpha}-1}$.

Further employing the decomposition trick in (Jin et al., 2021), we present the second-moment control:

**Lemma 4.4** (Second Moment Bound of Shifted Loss). *For any* $(s,a) \in \mathcal{S} \times \mathcal{A}$, *we have* $\mathbb{E}\left[\widetilde{\ell}_t^{\text{shift}}(s,a)^2 \mid \mathcal{F}_{t-1}\right] \leq \text{poly}(H,S,A)(1-\pi_t(a|s))\sigma^2 t^{\frac{2}{\alpha}-1} x_t(s,a)^{\frac{2}{\alpha}-2}$.

We discuss in detail and prove the above two lemmas in Appendix B.2 (see Lemmas B.2 and B.3).

**Suboptimal Mass Propagation.** A major approach towards BoBW guarantees is "excluding the optimal action" from the summation of regret terms. To be specific, we want to reform the total regret as the summation of some contributions on suboptimal actions, thereby leading to an instance-dependent bound of $\log(T)$ order.

In MABs, this is straightforward as the optimal policy is a Dirac distribution, which is concentrated on the single optimal action. However, it becomes difficult in MDPs when we consider the occupancy measure. Different from the case in MABs, the structure of polytope $\mathcal{Q}(\mathbb{P})$ is more complex due to the state-action structure. Specifically, for each $h \in [H]$, the occupancy measure must satisfy $\sum_{s_h \in \mathcal{S}_h} \rho(s_h) = 1$ and $\sum_{s_h \in \mathcal{S}_h, a \in \mathcal{A}} \mathbb{P}[s_{h+1}|s_h, a]\rho(s_h, a) = \rho(s_{h+1})$ for any $s_{h+1} \in \mathcal{S}_{h+1}$. Therefore, the reference policy's occupancy measure is distributed across the state space, which prevents us from simply excluding a single optimal action as in the MAB case.

In previous works focusing on achieving BoBW with bounded losses (Jin et al., 2021; Ito et al., 2025), their anal-

ysis relies on the special structure of the $1/2$-Tsallis entropy regularizer in analyzing the penalty term (arising from the time-varying regularizer) and the $(1 - \pi_t(a|s))$ term in the second-moment control of the shifted loss. However, for heavy-tailed losses, we consider the $1/\alpha$-Tsallis entropy regularizer and involve additional skipping error terms as the payload of skipping loss tuning, which poses significant challenges on analysis.

We address this via a *suboptimal mass propagation* analysis. We show that the total occupancy mass of the reference policy at any layer is upper-bounded by the cumulative mass of suboptimal actions in previous layers.

**Lemma 4.5** (Suboptimal Mass Propagation). *Let $\pi$ be any policy and $\pi^\dagger$ be a deterministic policy. Let $\rho^{\mathbb{P},\pi}(s) = \sum_{a \in \mathcal{A}} \rho^{\mathbb{P},\pi}(s,a)$ and $\rho^{\mathbb{P},\pi^\dagger}(s) = \sum_{a \in \mathcal{A}} \rho^{\mathbb{P},\pi^\dagger}(s,a)$ be the state marginals. Then,*

$$\sum_{s \in \mathcal{S}} (\rho^{\mathbb{P},\pi}(s) - \rho^{\mathbb{P},\pi^\dagger}(s))_+ \leq H \sum_{(s,a): a \neq \pi^\dagger(s)} \rho^{\mathbb{P},\pi}(s,a).$$

This result allows us to bound the penalty terms and the skipping errors via the summation over suboptimal actions, which enables us to achieve an instance-dependent $O(\log T)$ rate in the self-bounding regime. See Lemma B.8 for a formal proof.

# 5. HTMDPs with Unknown Transitions

In this section, we extend our framework to the more challenging setting where the transition kernel $\mathbb{P}$ is unknown, and establish the first BoBW regret guarantees for HTMDPs with unknown transitions. To handle the additional complications arising from transition estimation under heavy-tailed losses, we introduce the following mild regularity condition on the loss distributions:

**Assumption 5.1** (Truncative Nonnegativity). For each time step $t \in [T]$, state-action pair $(s,a) \in \mathcal{S} \times \mathcal{A}$ and any positive constant $M > 0$, the loss distribution $\nu_t(s,a)$ has positive truncated expectation at level $M$, i.e., $\mathbb{E}_{\ell \sim \nu_t(s,a)}[\ell \cdot \mathbb{1}[|\ell| \leq M]] \geq 0$.

Assumption 5.1 requires that the expected value of the loss, when truncated at any level $M > 0$, remains nonnegative. This is a mild condition satisfied by a broad class of distributions, including all nonnegative-valued distributions (e.g., the $[0, 1]$-bounded losses widely assumed in standard RL (Jin et al., 2020; 2021)) and distributions that may take negative values but have sufficiently positive means. Compared to the strict nonnegativity $\ell \geq 0$ commonly required in RL, Assumption 5.1 is weaker, as it permits the loss to attain negative values. An analogous condition has been employed in the heavy-tailed MAB literature to establish BoBW guarantees (Huang et al., 2022; Genalti et al., 2024), where it

serves a similar role in controlling the sign of the skipping bias. Importantly, this assumption is only required for the unknown transition case: our results for the known transition setting (Section 4) hold without any sign restriction on the losses. The necessity of this additional condition stems from the interaction between the skipping loss estimator and the Upper Occupancy Bound (UOB) in the unknown transition setting. Specifically, the UOB $u_t(s, a)$ serves as an optimistic proxy for the true occupancy measure $\rho^{\mathbb{P}, \pi_t}(s, a)$, and the resulting discrepancy introduces an additional bias term whose sign must be controlled to maintain the pessimism of our estimator (see the analysis of SKIPERR in Section C).

Under this assumption, we present algorithm HT-FTRL-UOB (Algorithm 2) equipped with our novel pessimistic estimator to address the heavy-tailed losses in the unknown transition case, and then establish BoBW regret guarantees by developing innovative techniques.

### 5.1. Algorithm for the Unknown Transition Case

Building upon the foundation of Algorithm 1, we further develop our robust estimation framework for the challenging unknown-transition setting, proposing HT-FTRL-UOB (Algorithm 2). While this algorithm adopts the doubling-epoch transition estimation from prior literature (Jin et al., 2020; 2021), its core innovation lies in the integration of a pessimistic skipping loss estimator with Upper Occupancy Bounds (UOB).

In Line 4, Algorithm 2 computes the occupancy $\boldsymbol{x}_t$ on the empirical model $\widehat{\mathbb{P}}_i$ by FTRL with a pessimistic estimator $\widehat{\ell}_t(s, a)$ in Eq. (5) and a $1/\alpha$-Tsallis entropy regularizer $\Psi_t(\boldsymbol{x})$. Notice that the decrease of learning rate $\eta_t$ is based on the length of the epoch $t - t_i + 1$. Lines 5 and 6 are the policy derivation, trajectory rollout, and counter update, which are the same as in the known transition case. In Line 7, the Upper Occupancy Bound (UOB) $u_t(s, a)$ is computed via the Comp-UOB procedure (Jin et al., 2020) in polynomial time. We construct the pessimistic estimator $\widehat{\ell}_t(s, a)$ in Line 8, by the biased importance sampling of skipped loss $\ell_t^{\text{skip}}(s, a)$ via the UOB $u_t(s, a)$, and penalizing the skipping bonus $b_t^{\text{skip}}(s, a)$ and exploration bonus $D \cdot B_i(s, a)$. Then, in Lines 9-12, we update the empirical model and confidence set for the new epoch using the standard doubling epoch schedule.

**Pessimistic Estimator.** To achieve BoBW regret bounds for heavy-tailed losses under unknown transition, we develop our novel pessimistic estimator $\widehat{\ell}_t(s, a)$ in Eq. (5) with the *epoch-level* skipping loss

$$\ell_t^{\text{skip}}(s, a) := \ell_t(s, a) \cdot \mathbb{1}[|\ell_t(s, a)| \leq \tau_t(s, a)], \quad (6)$$

---

**Algorithm 2** HT-FTRL-UOB

**Input:** Heavy-tail parameters $\alpha$ and $\sigma$, skipping constant $C > 0$, learning rate constant $\beta > 0$, bonus constant $D$, and confidence parameter $\delta$.

1: Initialize epoch index $i = 1$, $t_i = 1$, counters $m(s_h, a) = 0, m(s_h, a, s_{h+1}) = 0$, snapshot $m_{old}(s_h, a) = 0$, empirical model $\widehat{\mathbb{P}}_1[s_{h+1}|s_h, a] = 1/|\mathcal{S}|$, and confidence width $B_1(s_h, a, s_{h+1}) = 1$ for all $(s_h, a, s_{h+1}) \in \mathcal{S}_h \times \mathcal{A} \times \mathcal{S}_{h+1}$,

2: **for** $t = 1, \ldots, T$ **do**

3:     Set epoch index $i(t) \leftarrow i$.

4:     Compute occupancy $\boldsymbol{x}_t$ on the empirical model $\widehat{\mathbb{P}}_i$:

$$\boldsymbol{x}_t \leftarrow \operatorname*{argmin}_{\boldsymbol{x} \in \mathcal{Q}(\widehat{\mathbb{P}}_i)} \left\langle \boldsymbol{x}, \sum_{\tau=t_i}^{t-1} \widehat{\boldsymbol{\ell}}_\tau \right\rangle + \Psi_t(\boldsymbol{x}), \quad (4)$$

    where $\Psi_t(\boldsymbol{x}) = -\frac{1}{\eta_t} \sum_{(s,a)} x(s, a)^{1/\alpha}$ and $\eta_t = \beta/(\sigma(t - t_i + 1)^{1/\alpha})$.

5:     Derive policy $\pi_t(a|s) \leftarrow x_t(s, a)/\sum_{a' \in \mathcal{A}} x_t(s, a')$.

6:     Roll out episode $t$ using $\pi_t$, and observe trajectory $\{s_h^t, a_h^t\}_{h \in [H]}$ and losses $\ell_t(s_h^t, a_h^t)$. Update counters $m(s_h^t, a_h^t)$ and $m(s_h^t, a_h^t, s_{h+1}^t)$.

7:     Compute $u_t(s, a) = \max_{\widehat{\mathbb{P}} \in \widehat{\mathcal{P}}_i} \rho^{\widehat{\mathbb{P}}, \pi_t}(s, a)$ via the Comp-UOB procedure in Jin et al. (2020).

8:     Construct pessimistic estimator $\widehat{\ell}_t(s, a)$ for all $(s, a)$:

$$\begin{aligned}\widehat{\ell}_t(s, a) &\leftarrow \frac{\ell_t^{\text{skip}}(s, a)}{u_t(s, a)} \cdot \mathbb{1}_t[(s, a)] - b_t^{\text{skip}}(s, a) \\ &\quad - D \cdot B_i(s, a),\end{aligned} \quad (5)$$

    with $\ell_t^{\text{skip}}(s, a)$ defined in Eq. (6), $b_t^{\text{skip}}(s, a)$ defined in Eq. (8), $\mathbb{1}_t[(s, a)] = \mathbb{1}[(s, a) \in \{s_h^t, a_h^t\}_{h \in [H]}]$, and $B_i(s, a) = \min\{1, \sum_{s'} B_i(s, a, s')\}$ with $B_i(s, a, s')$ defined in Eq. (9).

9:     **if** $\exists (s, a), m(s, a) \geq \max\{1, 2 \cdot m_{old}(s, a)\}$ **then**

10:         Start new epoch: $i \leftarrow i + 1$, $t_i \leftarrow t + 1$ and save the snapshot: $m_{old} = m$.

11:         For each $(s_h, a, s_{h+1}) \in \mathcal{S}_h \times \mathcal{A} \times \mathcal{S}_{h+1}$, update empirical model $\widehat{\mathbb{P}}_i[s_{h+1}|s_h, a] \leftarrow m(s_h, a, s_{h+1})/\max\{1, m(s_h, a)\}$.

12:         Update confidence set $\widehat{\mathcal{P}}_i \leftarrow \{\widehat{\mathbb{P}} : |\widehat{\mathbb{P}}[s_{h+1}|s_h, a] - \widehat{\mathbb{P}}_i[s_{h+1}|s_h, a]| \leq B_i(s, a, s'), \forall (s_h, a, s_{h+1}) \in \mathcal{S}_h \times \mathcal{A} \times \mathcal{S}_{h+1}\}$.

13:     **end if**

14: **end for**

---

where the skipping threshold $\tau_t(s, a)$ and skipping bonus $b_t^{\text{skip}}(s, a)$ are defined as

$$\tau_t(s, a) := C \cdot \sigma(t - t_{i(t)} + 1)^{1/\alpha} x_t(s, a)^{1/\alpha}. \quad (7)$$

$$b_t^{\text{skip}}(s, a) := C^{1-\alpha} \cdot \sigma(t - t_{i(t)} + 1)^{1/\alpha - 1} x_t(s, a)^{1/\alpha - 1}. \quad (8)$$

Then we penalize estimators by the confidence width $B_i(s,a) = \min\{1, \sum_{s'} B_i(s,a,s')\}$ scaling with a constant $D$ to account for the transition uncertainty, where the Bernstein-type confidence width $B_i(s,a,s')$ is defined as

$$B_i(s,a,s') = \min\left\{2\sqrt{\frac{\widehat{\mathbb{P}}_i[s'|s,a]\log(\iota)}{m(s,a)}} + \frac{14\log(\iota)}{3m(s,a)}, 1\right\},$$
(9)

where $\iota = HSAT/\delta$. Our novel loss estimator with epoch-level skipping and adjusted skipping bonus allows us to avoid extreme outlier losses that can cause instability in heavy-tailed MDPs and compensate for the skipping bias in the unknown transition case.

### 5.2. Main Result

The following theorem demonstrates that HT-FTRL-UOB achieves the BoBW regret bounds, which is the first BoBW algorithm for heavy-tailed MDPs with unknown transitions to the best of our knowledge.

**Theorem 5.2** (BoBW Guarantee for the Unknown Transition Case). *By setting constant $\delta = 1/T^3$, $C$ in Eq. (17), $\beta$ in Eq. (18), and $D = H\sigma$, HT-FTRL-UOB (Algorithm 2) achieves BoBW regret bounds in the adversarial and self-bounding regimes simultaneously under Assumption 5.1:*

*1. Adversarial Regime: For any adversarial loss distributions and deterministic benchmark policy $\mathring{\pi}$,*

$$\text{Reg}_T(\mathring{\pi}) = \widetilde{\mathcal{O}}\left(\text{poly}(H,S,A)\sigma\left(T^{1/\alpha} + \sqrt{T}\right)\right)$$

*2. Self-Bounding Regime: If the environment satisfies the $(\mathring{\pi}, \Delta, C_{sb})$ self-bounding constraint (Definition 3.1) and $\Delta_{\min} = \min_{(s,a):a\neq\mathring{\pi}(s)}\Delta(s,a)$ is the minimal suboptimal gap, then we have*

$$\text{Reg}_T(\mathring{\pi}) = \mathcal{O}\Bigg(\text{poly}(H,S,A)\cdot\Big(U + W_2 + W_{\min} + W_\alpha$$
$$+ C_{sb}^{\frac{1}{2}}W_2^{\frac{1}{2}} + C_{sb}^{\frac{1}{2}}W_{\min}^{\frac{1}{2}} + C_{sb}^{\frac{1}{\alpha}}W_\alpha^{1-\frac{1}{\alpha}}\Big)\Bigg)$$

*where we denote $W_2 := \sigma^2\omega_\Delta(2)\log(T)$, $W_{\min} := \sigma^2\Delta_{\min}^{-1}\log(T)$, $W_\alpha := \sigma^{\frac{\alpha}{\alpha-1}}\omega_\Delta(\alpha)\log(T)^2$, $\omega_\Delta(x) := \sum_{(s,a):a\neq\mathring{\pi}(s)}\Delta(s,a)^{-\frac{1}{x-1}}$, and $U = \sigma\log(T)^2$.*

We refer to the formal proof in Appendix C. This theorem shows that our proposed HT-FTRL-UOB achieves the BoBW regret bounds in the adversarial regime with instance-independent bound $\widetilde{\mathcal{O}}(\sigma(T^{1/\alpha} + \sqrt{T}))$ and self-bounding regime with instance-dependent bound $\mathcal{O}(\text{poly}(\log(T)))$. Specifically, in the stochastic regime, the instance-dependent bound is exactly $\mathcal{O}(\text{poly}(H,S,A)\cdot(\sigma^{\alpha/(\alpha-1)}\omega_\Delta(\alpha)\log(T)^2))$. The additional $\log(T)$ factor

compared to the known transition case (Theorem 4.1) arises from the doubling-epoch schedule for transition estimation, which incurs $O(SA\log T)$ epochs and corresponding union bounds.

*Remark* 5.3 (Tightness under Unknown Transitions). For HTMDPs with unknown transitions, Zhuang & Sui (2021) established the instance-independent lower bound $\Omega(\sqrt{T} + T^{1/\alpha})$, where the $\sqrt{T}$ term arises from transition estimation and the $T^{1/\alpha}$ term from heavy-tailed losses. The adversarial regret bound of HT-FTRL-UOB in Theorem 5.2 matches this lower bound in $\sigma$ and $T$ up to $\text{poly}(H,S,A)$ and log factors. To the best of our knowledge, Theorem 5.2 also provides the first $\text{poly}(\log T)$ instance-dependent regret bound for heavy-tailed MDPs with unknown transitions. In the sub-Gaussian case ($\alpha = 2$), our bounds recover the results of Jin et al. (2021); Ito et al. (2025) in $T$ and the suboptimal gaps $\Delta$ up to $\text{poly}(H,S,A)$ factors.

**Technical Innovations.** To analyze the regret bound for Algorithm 2, we derive an innovative regret decomposition scheme for HTMDPs with unknown transitions. Our novel pessimistic estimator allows us to decompose the regret into four terms: transition error, estimation error, skipping error, and pessimism error. This decomposition is different from previous works with similar algorithm framework (Jin et al., 2020; 2021; Ito et al., 2025), since the heavy-tailed losses and skipping tuning technique considered in heavy-tailed settings introduce extra skipping error terms. Controlling this term requires careful design of the transition kernel and UOB deviation term. To describe our innovative regret decomposition scheme for HTMDPs with unknown transitions, we first introduce the auxiliary loss function $\ell_t^{\text{pess}}$:

$$\ell_t^{\text{pess}}(s,a) := \ell_t(s,a) - D\cdot B_i(s,a). \quad (10)$$

This term is crucial for regret decomposition, since we need to carefully arrange the decomposition order of the extra operations we did in Line 8 of Algorithm 2 to construct the pessimistic estimator $\widehat{\ell}_t(s,a)$. Specifically, we have the following regret decomposition: $\text{Reg}_T(\mathring{\pi}) = \mathbb{E}\left[\sum_{t=1}^T V^{\mathbb{P},\pi_t}(s_1;\ell_t) - V^{\mathbb{P},\mathring{\pi}}(s_1;\ell_t)\right] = \text{TRANSERR} + \text{ESTREG} + \text{SKIPERR} + \text{PESSERR}$, where we denote these four terms as:

$$\text{TRANSERR} = \mathbb{E}\left[\sum_{t=1}^T V^{\mathbb{P},\pi_t}(s_1;\ell_t) - V^{\widehat{\mathbb{P}}_{i(t)},\pi_t}(s_1;\ell_t^{\text{pess}})\right],$$

$$\text{ESTREG} = \mathbb{E}\left[\sum_{t=1}^T V^{\widehat{\mathbb{P}}_{i(t)},\pi_t}(s_1;\widehat{\ell}_t) - V^{\widehat{\mathbb{P}}_{i(t)},\mathring{\pi}}(s_1;\widehat{\ell}_t)\right],$$

$$\text{SKIPERR} = \mathbb{E}\Bigg[\sum_{t=1}^T V^{\widehat{\mathbb{P}}_{i(t)},\pi_t}(s_1;\ell_t^{\text{pess}} - \widehat{\ell}_t)$$
$$- V^{\widehat{\mathbb{P}}_{i(t)},\mathring{\pi}}(s_1;\ell_t^{\text{pess}} - \widehat{\ell}_t)\Bigg],$$

$$\text{PESSERR} = \mathbb{E}\left[\sum_{t=1}^{T} V^{\widehat{\mathbb{P}}_{i(t)}, \mathring{\pi}}(s_1; \ell_t^{\text{pess}}) - V^{\mathbb{P}, \mathring{\pi}}(s_1; \ell_t)\right].$$

This decomposition order is crucial, since we can handle ES-TREG by the similar methods we conducted for the known transition case, and handle PESSERR by the Bernstein-type concentration argument (given in Jin et al. (2020)). Therefore, we can focus on bounding the TRANSERR and SKIPERR terms. The TRANSERR term can be solved by applying similar techniques introduced by Jin et al. (2021). We highlight the procedure of bounding the TRANSERR term here.

We first note that the FTRL result $x_t$ is the occupancy measure $\rho^{\widehat{\mathbb{P}}_{i(t)}, \pi_t}$, which does not match the true occupancy measure $\rho^{\mathbb{P}, \pi_t}$ in the unknown transition case. Therefore, we can only handle the skipping error term SKIPERR by incorporating the skipping bonus to compensate for the skipping bias under the empirical model $\widehat{\mathbb{P}}_{i(t)}$ given the definition of $b_t^{\text{skip}}(s, a)$ in Eq. (8), which provides the high-level intuition of constructing this regret decomposition. Note that in Line 8 of Algorithm 2, the skipping loss $\ell_t^{\text{skip}}(s, a)$ is weighted by the optimistic importance sampling factor $u_t(s, a)$, which is not the exact occupancy $x_t(s, a)$ in the known transition case. Thus, we should involve the difference on UOB term when bounding the deviation between the "true" loss $\ell_t$ and estimated loss $\widehat{\ell}_t$.

**Lemma 5.4.** *With high probability, we have*

$$\text{SKIPERR} \leq \mathbb{E}\left[\mathcal{O}\left(\sum_{t=1}^{T} \sigma t^{1/\alpha - 1} \sum_{(s,a): a \neq \mathring{\pi}(s)} x_t(s,a)^{1/\alpha}\right)\right]$$
$$+ \mathbb{E}\left[\mathcal{O}\left(\sum_{t=1}^{T} \sum_{(s,a)} \sigma |u_t(s,a) - \rho^{\mathbb{P}, \pi_t}(s,a)|\right)\right].$$

Here the first term is the desired form that excludes the reference action term $\mathring{\pi}(s)$, which can further derive instance-dependent and instance-independent regret bounds. The second term is the UOB deviation term, which can be bounded by the residual UOB argument in Jin et al. (2021). We refer to the formal version of this lemma in Lemma C.9.

## 6. Conclusion

In this paper, we present the first Best-of-Both-Worlds (BoBW) framework for episodic Markov decision processes with heavy-tailed losses (HTMDPs). For settings with known transitions, we propose HT-FTRL-OM, an algorithm leveraging the FTRL framework with $1/\alpha$-Tsallis entropy regularization and a novel skipping loss estimator. By introducing innovative techniques such as local control for heavy-tailed shifted losses and suboptimal mass propaga-

tion, we establish that HT-FTRL-OM achieves an instance-independent regret $\widetilde{\mathcal{O}}(\sigma T^{1/\alpha})$, minimax-optimal in $\sigma, T$ up to $\text{poly}(H, S, A)$ and log factors, in adversarial regimes and logarithmic instance-dependent regret $\mathcal{O}(\log T)$ in self-bounding environments.

We further extend our results to the unknown transition setting with HT-FTRL-UOB, which incorporates a pessimistic estimator and upper occupancy bounds under a mild truncative nonnegativity condition (Assumption 5.1). Through a novel regret decomposition that isolates transition uncertainty from heavy-tailed estimation errors, we prove that this algorithm attains a regret bound of $\widetilde{\mathcal{O}}(T^{1/\alpha} + \sqrt{T})$ in adversarial environments and $\mathcal{O}(\log^2 T)$ in stochastic ones. Our work bridges the gap between heavy-tailed RL and adaptive algorithm design, and is particularly relevant for RL applications with intrinsically heavy-tailed feedback, such as network traffic routing, financial decision-making, and image signal processing. Several limitations of our work suggest important future directions: tightening the bounds with respect to $S$, $A$ and $H$ factors, relaxing or removing the truncative nonnegativity condition in the unknown transition case (as achieved by Zhao et al. (2025) in the bandit setting), and extending these guarantees to the function approximation setting.

## Impact Statement

This paper presents work whose goal is to advance the field of Machine Learning. There are many potential societal consequences of our work, none which we feel must be specifically highlighted here.

## Acknowledgement

This work was supported by the National Natural Science Foundation of China Grant 52494974.

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

# Appendices

## A. Related Works

### A.1. Heavy-Tailed Feedback in Online Decision

#### A.1.1. HEAVY-TAILED MULTI-ARMED BANDITS

The study of heavy-tailed feedback in online learning dates back to stochastic bandits, where the payoff distribution may only admit a bounded $\alpha$-moment ($\alpha \in (1, 2]$) and classical sub-Gaussian concentration no longer applies. Bubeck et al. (2013) pioneered the use of truncated mean estimators in UCB-style algorithms for heavy-tailed stochastic MABs. Medina & Yang (2016) extended robust estimation ideas to heavy-tailed *linear* bandits and established sublinear regret without requiring boundedness. For stochastic linear bandits, Shao et al. (2018) and Xue et al. (2021) provided (near-)optimal regret rates under heavy-tailed payoffs via carefully designed median-of-means/dynamic truncation estimators. More recently, Wang et al. (2025) developed a Huber-regression based approach with one-pass updates for heavy-tailed linear bandits, further illustrating the broad utility of robust estimation in heavy-tailed online decision-making. Lu et al. (2019) studied Lipschitz bandits with heavy-tailed rewards and derived optimal regret rates under moment conditions. Beyond bandits, Ray Chowdhury & Gopalan (2019) investigated Bayesian optimization under heavy-tailed payoffs by combining truncation with GP-UCB style exploration. Overall, these works highlight that robust mean estimation is indispensable under heavy tails and typically yields worst-case regret scaling as a polynomial in $T$ (e.g., $T^{1/\alpha}$), in sharp contrast to the $\widetilde{\mathcal{O}}(\sqrt{T})$ rates under light tails. At a technical level, our skipping loss estimator can be viewed as an occupancy-adaptive form of truncation, but it is embedded into FTRL over occupancy measures and is tailored to control stability/second-order terms that arise in BoBW analyses for MDPs.

#### A.1.2. HEAVY-TAILED MARKOV DECISION PROCESSES

For episodic RL with heavy-tailed rewards, Zhuang & Sui (2021) established no-regret guarantees for tabular MDPs with heavy-tailed feedback and proposed robust variants of optimistic algorithms, achieving near-optimal *instance-independent* rates under moment assumptions (and matching lower bounds up to logarithmic factors in several regimes). Park & Lee (2024) also studied heavy-tailed RL from the perspective of robust estimation and proposed a penalized robust estimator framework. Moving beyond tabular models, Huang et al. (2023) studied heavy-tailed RL with (linear) function approximation and derived minimax-optimal regret bounds together with instance-dependent guarantees; Li & Sun (2024) further developed variance-aware algorithms for linear MDPs under heavy-tailed rewards. These *instance-dependent* guarantees are typically expressed in terms of problem-dependent quantities such as conditional variances or instance measures, and their dependence on the number of episodes remains polynomial (e.g., scaling like $T^{1/\alpha}$ in heavy-tailed regimes), rather than logarithmic in $T$. Moreover, existing HTMDP results almost exclusively focus on (stationary) stochastic environments and do not address best-of-both-worlds adaptivity when the regime is unknown *a priori*. Orthogonal to online regret minimization, another line of work studies robustness in *offline* RL under heavy-tailed rewards or data corruption (Zhu et al., 2024; Yang et al., 2024), and privacy-constrained heavy-tailed RL (Wu et al., 2023). In contrast, our goal is to achieve best-of-both-worlds guarantees for HTMDPs, i.e., instance-independent regret minimax-optimal in $\sigma, T$ up to $\text{poly}(H, S, A)$ and log factors in adversarial regimes, while simultaneously attaining *logarithmic* instance-dependent regret in self-bounding (including stochastic) regimes (e.g., $\mathcal{O}(\log T)$ for known transitions and $\mathcal{O}(\log^2 T)$ for unknown transitions in our results).

### A.2. Best-of-Both-Worlds (BoBW) Algorithms for MDPs

The best-of-both-worlds goal in online episodic MDPs is to design a single algorithm that achieves near-minimax regret in the worst case (e.g., $\widetilde{\mathcal{O}}(\sqrt{T})$ against adversarial losses) while simultaneously enjoying much smaller regret in benign regimes (e.g., polylogarithmic regret under stochastic/self-bounding losses), without knowing the regime *a priori*. A unifying backbone behind the first BoBW results for episodic MDPs is to cast learning as online optimization over *occupancy measures* and apply FTRL with carefully chosen regularization. In particular, for tabular MDPs with *bounded* losses and bandit feedback, Jin & Luo (2020) gave the first BoBW result for the known-transition setting by running FTRL over occupancy measures with a carefully designed hybrid regularizer, achieving $\widetilde{\mathcal{O}}(\sqrt{T})$ regret in the adversarial regime while attaining $\mathcal{O}(\log T)$ regret when losses are i.i.d. stochastic (and more generally, graceful degradation under corrupted stochastic losses). Jin et al. (2021) extended this line to the unknown-transition setting, again within the occupancy-measure FTRL paradigm, introducing a loss-shifting analysis and a key argument that controls transition-estimation error by the

regret itself in stochastic regimes, leading to $\widetilde{\mathcal{O}}(\sqrt{T})$ adversarial regret and $\mathcal{O}(\log^2 T)$ stochastic regret.

Going beyond fixed dynamics, Jin et al. (2023) studied online RL when both losses and transitions may be adversarial; since fully adversarial transitions make no-regret learning impossible in general, they obtain guarantees that scale with a transition-adversariality/corruption measure, thereby interpolating between learnable and impossible regimes within an online-learning framework for MDPs. With more restrictive feedback, Ito et al. (2025) studied episodic MDPs with *aggregate* bandit feedback (only the episode-level cumulative loss is observed) and developed BoBW adaptivity between stochastic and adversarial loss regimes under this limited observation model, highlighting additional connections to partial-feedback/online-learning tools beyond the standard step-wise bandit setting.

Complementary to the occupancy-measure viewpoint, Dann et al. (2023) provided BoBW guarantees for policy-optimization style algorithms in tabular MDPs by combining entropy-type regularization, exploration bonuses, and learning-rate schedules, achieving $\sqrt{T}$-type worst-case regret while attaining polylogarithmic regret in stochastic regimes; under known transitions, they further derive first-order bounds in the adversarial regime via log-barrier regularization. Finally, while not BoBW, Liu et al. (2023) sharpened the worst-case baseline by studying adversarial *linear* MDPs with bandit feedback and proving improved (rate-optimal for an inefficient algorithm) regret bounds. At a broader methodological level, Ito et al. (2022) established nearly optimal BoBW algorithms for online learning with feedback graphs, which provides a general toolkit for partial-feedback settings that also inform subsequent developments in online RL.

All the above BoBW-MDP results crucially rely on bounded (light-tailed) losses and do not address robustness to heavy-tailed feedback, which is the focus of our work.

## A.3. BoBW Algorithms with Heavy-Tailed Feedback

Best-of-both-worlds learning under heavy-tailed feedback has been studied primarily in bandit models. For heavy-tailed multi-armed bandits (MABs) where losses/rewards may be unbounded but admit only a bounded $\alpha$-moment, Huang et al. (2022) introduced the first BoBW algorithms that simultaneously achieve minimax-optimal polynomial regret in adversarial environments (scaling as $T^{1/\alpha}$ up to factors depending on the number of arms and logs) and logarithmic (gap-dependent) regret in stochastic environments, with additional adaptivity to unknown tail parameters. Recently, Cheng et al. (2024) further strengthened this line by developing BoBW guarantees for adversarial bandits with heavy-tailed losses under mild moment assumptions, providing high-probability bounds and relaxing technical distributional conditions used in earlier work. On the parameter-free front (where the tail parameter $\alpha, \sigma$ are unknown), Chen et al. (2025) proposed UNIINF, a BoBW algorithm for heavy-tailed MABs that do not require prior knowledge of tail parameters and achieves near-optimal regret in both stochastic and adversarial regimes via adaptive learning-rate and robust loss-control mechanisms. Beyond classical MABs, Zhao et al. (2025) extended BoBW methodology to heavy-tailed *linear* bandits and broader adversarial heavy-tailed bandit problems via an FTRL-based framework with data-dependent stability–penalty matching. They also achieve BoBW without any further nonnegativity assumption on loss distribution.

In contrast to these bandit works, our setting involves episodic MDPs where feedback is coupled across time through state visitation and transition dynamics. This coupling introduces additional stability/second-order challenges (e.g., through occupancy-measure constraints) that do not arise in bandits, and necessitates a robust loss estimator that can be embedded into BoBW analyses for MDPs.

# B. Proof of BoBW Guarantee for Known Transition Case

In this section, we give the proof of the BoBW guarantee for Algorithm 1. We first state the formal version of Theorem 4.1 in Theorem B.1.

**Theorem B.1** (Best-of-Both-Worlds Regret Guarantee). *Set constant $C$ in Eq. (17) and $\beta$ in Eq. (18). Then for known heavy-tailed parameter $(\sigma, \alpha)$ and transition kernel $\mathbb{P}$,* HT-FTRL-OM *(Algorithm 1) achieves the following regret bounds simultaneously in the adversarial and self-bounding regimes:*

1. ***Adversarial Regime:*** *For any adversarial sequence of loss functions $\{\ell_t\}_{t=1}^{T}$, and any given reference policy $\mathring{\pi}$ (usually the optimal adversarial policy), we have*

$$\mathrm{Reg}_T(\mathring{\pi}) \leq \frac{8\alpha^2}{\alpha - 1}\beta C^{2-2\alpha} S A^{1-1/\alpha} \cdot \sigma(1 + \log(T)) + 2\alpha H^{1/\alpha} S^{1-1/\alpha} A^{1-1/\alpha} M \cdot \sigma T^{1/\alpha}. \tag{11}$$

*where $M > 0$ is a constant depending on $H, S, A, C, \beta, \alpha$:*

$$M := \frac{32\alpha^2}{\alpha - 1}H^2 \beta C^{2-\alpha} + \frac{2 + 2H^{1/\alpha}S^{1-1/\alpha}}{\beta} + 2C^{1-\alpha}(1 + H^{1/\alpha}S^{1-1/\alpha}). \tag{12}$$

*After substituting $C$ and $\beta$ from Eqs. (17) and (18), the coefficient $M$ admits the explicit two-piece $\mathcal{O}$ form*

$$M = \mathcal{O}\left(H^{2-1/\alpha} \cdot S^{-1/\alpha} \cdot A^{-(2\alpha-1)/\alpha^2}\right) + \mathcal{O}\left(H \cdot S^{2-2/\alpha} \cdot A^{2-3/\alpha+1/\alpha^2}\right). \tag{13}$$

*Combined with the definition of $C$ and $\beta$ in Eqs. (17) and (18), we have*

$$\mathrm{Reg}_T(\mathring{\pi}) \leq \mathcal{O}\Bigg( H^{1-\frac{1}{\alpha}}S^{2-\frac{1}{\alpha}}A^{3-\frac{4}{\alpha}+\frac{1}{\alpha^2}} \cdot \sigma \cdot \log(T) + H^{1+\frac{1}{\alpha}}S^{3-\frac{3}{\alpha}}A^{3-\frac{4}{\alpha}+\frac{1}{\alpha^2}} \cdot \sigma \cdot T^{1/\alpha}$$
$$+ H^2 \cdot S^{1-\frac{2}{\alpha}} \cdot A^{1-\frac{3}{\alpha}+\frac{1}{\alpha^2}} \cdot \sigma \cdot T^{1/\alpha} \Bigg) \tag{14}$$

2. ***Self-Bounding Regime:*** *If the environment satisfies the $(\mathring{\pi}, \Delta, C_{\mathrm{sb}})$ self-bounding constraint (Definition 3.1), then we have*

$$\mathrm{Reg}_T(\mathring{\pi}) \leq \mathcal{O}\Bigg( \frac{\alpha^2}{\alpha - 1}\beta C^{2-2\alpha} S A^{1-\frac{1}{\alpha}} \cdot \sigma(1 + \log(T)) + 2^{\frac{1}{\alpha-1}}M^{\frac{\alpha}{\alpha-1}} \cdot \sigma^{\frac{\alpha}{\alpha-1}}\omega_\Delta(\alpha)(1 + \log(T))$$
$$+ C_{\mathrm{sb}}^{1/\alpha} \cdot M \cdot \sigma\omega_\Delta(\alpha)^{\frac{\alpha-1}{\alpha}} \cdot (1 + \log(T))^{\frac{\alpha-1}{\alpha}} \Bigg), \tag{15}$$

*where $\omega_\Delta(\alpha) := \sum_{(s,a):a \neq \mathring{\pi}(s)} \Delta(s,a)^{-\frac{1}{\alpha-1}}$ and $M$ defined in Eq. (12). We can calculate the order:*

$$\mathrm{Reg}_T(\mathring{\pi}) \leq \mathcal{O}\Bigg( H^{1-\frac{1}{\alpha}}S^{2-\frac{1}{\alpha}}A^{3-\frac{4}{\alpha}+\frac{1}{\alpha^2}} \cdot \sigma(1 + \log(T))$$
$$+ H^{\frac{\alpha}{\alpha-1}}S^2 A^{2-\frac{1}{\alpha}} \cdot \sigma^{\frac{\alpha}{\alpha-1}}\omega_\Delta(\alpha)(1 + \log(T))$$
$$+ H \cdot S^{2-\frac{2}{\alpha}}A^{2-\frac{3}{\alpha}+\frac{1}{\alpha^2}} \cdot C_{\mathrm{sb}}^{\frac{1}{\alpha}} \cdot \sigma\omega_\Delta(\alpha)^{1-\frac{1}{\alpha}}\left(1 + \log(T)\right)^{1-\frac{1}{\alpha}}$$
$$+ H^{\frac{2\alpha-1}{\alpha-1}} \cdot S^{-\frac{1}{\alpha-1}} \cdot A^{-\frac{2\alpha-1}{\alpha(\alpha-1)}} \cdot \sigma^{\frac{\alpha}{\alpha-1}}\omega_\Delta(\alpha)(1 + \log(T))$$
$$+ H^{2-\frac{1}{\alpha}} \cdot S^{-\frac{1}{\alpha}} \cdot A^{-\frac{2\alpha-1}{\alpha^2}} \cdot C_{\mathrm{sb}}^{\frac{1}{\alpha}} \cdot \sigma\omega_\Delta(\alpha)^{1-\frac{1}{\alpha}}\left(1 + \log(T)\right)^{1-\frac{1}{\alpha}} \Bigg) \tag{16}$$

*In particular, the algorithm achieves best-of-both-worlds performance automatically under adversarial regime and self-bounding regime, which includes the stochastic regime as a special case.*

To ensure the BoBW guarantees, we need to set the skipping threshold scaling parameter $C$ and learning rate scaling parameter $\beta$ by:

$$C := \left( \frac{(1 + HSA + HSA^{2-1/\alpha})}{\alpha - 1} \right)^{-1/\alpha}. \tag{17}$$

$$\beta := \frac{\alpha - 1}{4\alpha^2} \cdot \alpha^{-1}(\alpha - 1)^{1-1/\alpha}(1 + HSA + HSA^{2-1/\alpha})^{1/\alpha - 1}. \tag{18}$$

We first introduce the regret decomposition technique in the following sections. Then we illustrate the key components we need to control in the regret analysis. The formal proof of the theorem is provided in Appendix B.6.

### B.1. Regret Decomposition

We first apply FTRL Regret decomposition in investigating the performance of Algorithm 1.

Given a fixed deterministic policy $\mathring{\pi}$, we denote $\boldsymbol{\rho}^{\mathbb{P},\mathring{\pi}}$ as the occupancy measure induced by $\mathring{\pi}$. We conduct the regret decomposition as follows:

$$
\begin{aligned}
\mathrm{Reg}_T(\mathring{\pi}) &= \mathbb{E}\left[ \sum_{t=1}^{T} V_t^{\mathbb{P},\pi_t}(s_1) - V_t^{\mathbb{P},\mathring{\pi}}(s_1) \right] \\
&= \mathbb{E}\left[ \sum_{t=1}^{T} \left\langle \boldsymbol{\rho}^{\mathbb{P},\pi_t} - \boldsymbol{\rho}^{\mathbb{P},\mathring{\pi}}, \boldsymbol{\ell}_t \right\rangle \right] \\
&= \mathbb{E}\left[ \sum_{t=1}^{T} \left\langle \boldsymbol{x}_t - \boldsymbol{\rho}^{\mathbb{P},\mathring{\pi}}, \boldsymbol{\ell}_t^{\mathrm{skip}} - \boldsymbol{b}_t^{\mathrm{skip}} \right\rangle \right] + \mathbb{E}\left[ \sum_{t=1}^{T} \left\langle \boldsymbol{x}_t - \boldsymbol{\rho}^{\mathbb{P},\mathring{\pi}}, \boldsymbol{\ell}_t - \boldsymbol{\ell}_t^{\mathrm{skip}} + \boldsymbol{b}_t^{\mathrm{skip}} \right\rangle \right] \\
&= \underbrace{\mathbb{E}\left[ \sum_{t=1}^{T} \left\langle \boldsymbol{x}_t - \boldsymbol{\rho}^{\mathbb{P},\mathring{\pi}}, \widetilde{\boldsymbol{\ell}}_t^{\mathrm{skip}} - \boldsymbol{b}_t^{\mathrm{skip}} \right\rangle \right]}_{\text{FTRL Regret}} + \underbrace{\mathbb{E}\left[ \sum_{t=1}^{T} \left\langle \boldsymbol{x}_t - \boldsymbol{\rho}^{\mathbb{P},\mathring{\pi}}, \boldsymbol{\ell}_t - \boldsymbol{\ell}_t^{\mathrm{skip}} + \boldsymbol{b}_t^{\mathrm{skip}} \right\rangle \right]}_{\text{Skipping Error}},
\end{aligned}
$$

where the last equality holds since $\widetilde{\boldsymbol{\ell}}_t^{\mathrm{skip}}$ is the importance sampling over occupancies measure of $\boldsymbol{\ell}_t^{\mathrm{skip}}$.

Denote the Bregman Divergence $D_\Psi(\boldsymbol{x}, \boldsymbol{y})$ as

$$D_\Psi(\boldsymbol{y}, \boldsymbol{x}) = \Psi(\boldsymbol{y}) - \Psi(\boldsymbol{x}) - \langle \nabla\Psi(\boldsymbol{x}), \boldsymbol{y} - \boldsymbol{x} \rangle. \tag{19}$$

We verify that the FTRL update is invariant under the substitution $\widetilde{\boldsymbol{\ell}}_t^{\mathrm{skip}} \to \widetilde{\boldsymbol{\ell}}_t^{\mathrm{shift}}$. For any $\boldsymbol{\rho} \in \mathcal{Q}(\mathbb{P})$, define $W_t(s) := V^{\mathbb{P},\pi_t}(s; \widetilde{\boldsymbol{\ell}}_t^{\mathrm{skip}})$. Direct expansion gives

$$\langle \boldsymbol{\rho}, \widetilde{\boldsymbol{\ell}}_t^{\mathrm{shift}} - \widetilde{\boldsymbol{\ell}}_t^{\mathrm{skip}} \rangle = \sum_{(s,a)} \rho(s,a)\left[ \sum_{s'} \mathbb{P}[s' \mid s, a]\, W_t(s') - W_t(s) \right].$$

By the layered flow constraints of $\mathcal{Q}(\mathbb{P})$, $\sum_{(s,a)\in\mathcal{S}_h\times\mathcal{A}} \rho(s,a)\mathbb{P}[s' \mid s, a] = \rho(s')$ for $s' \in \mathcal{S}_{h+1}$. The sum telescopes over layers $h = 1, \ldots, H$, leaving boundary terms $\sum_{s\in\mathcal{S}_{H+1}} \rho(s)W_t(s) - \rho(s_1)W_t(s_1) = -W_t(s_1)$, where we used $W_t(s_{H+1}) = 0$ and $\rho(s_1) = 1$. Hence $\langle \boldsymbol{\rho}, \widetilde{\boldsymbol{\ell}}_t^{\mathrm{shift}} \rangle = \langle \boldsymbol{\rho}, \widetilde{\boldsymbol{\ell}}_t^{\mathrm{skip}} \rangle - V^{\mathbb{P},\pi_t}(s_1; \widetilde{\boldsymbol{\ell}}_t^{\mathrm{skip}})$, where the second term is a constant on $\mathcal{Q}(\mathbb{P})$ (independent of $\boldsymbol{\rho}$). Adding a constant to the linearized loss does not change the FTRL $\arg\min$, so the iterate $\boldsymbol{x}_t$ is unchanged under the shifted-loss substitution.

We further decompose the FTRL Regret via standard analysis for FTRL, using the shifted loss:

$$
\text{FTRL Regret} = \mathbb{E}\left[\sum_{t=1}^{T}\left\langle \boldsymbol{x}_t - \boldsymbol{\rho}^{\mathbb{P},\mathring{\pi}}, \widetilde{\boldsymbol{\ell}}_t^{\text{shift}} - \boldsymbol{b}_t^{\text{skip}}\right\rangle\right]
$$

$$
\leq \underbrace{\mathbb{E}\left[\sum_{t=1}^{T}\left\langle \boldsymbol{x}_t - \boldsymbol{x}_{t+1}, \widetilde{\boldsymbol{\ell}}_t^{\text{shift}} - \boldsymbol{b}_t^{\text{skip}}\right\rangle - D_{\Psi_t}(\boldsymbol{x}_{t+1}, \boldsymbol{x}_t)\right]}_{\text{Stability Term}} + \underbrace{\mathbb{E}\left[\sum_{t=1}^{T}\left(\Psi_t(\boldsymbol{\rho}^{\mathbb{P},\mathring{\pi}}) - \Psi_{t-1}(\boldsymbol{\rho}^{\mathbb{P},\mathring{\pi}})\right) - \left(\Psi_t(\boldsymbol{x}_t) - \Psi_{t-1}(\boldsymbol{x}_t)\right)\right]}_{\text{Penalty Term}}
$$

We further define $\texttt{stab}_t$, $\texttt{pena}_t$, and $\texttt{skip}_t$ as:

$$
\texttt{stab}_t := \left\langle \boldsymbol{x}_t - \boldsymbol{x}_{t+1}, \widetilde{\boldsymbol{\ell}}_t^{\text{shift}} - \boldsymbol{b}_t^{\text{skip}}\right\rangle - D_{\Psi_t}(\boldsymbol{x}_{t+1}, \boldsymbol{x}_t),
$$

$$
\texttt{pena}_t := \left(\Psi_t(\boldsymbol{\rho}^{\mathbb{P},\mathring{\pi}}) - \Psi_{t-1}(\boldsymbol{\rho}^{\mathbb{P},\mathring{\pi}})\right) - \left(\Psi_t(\boldsymbol{x}_t) - \Psi_{t-1}(\boldsymbol{x}_t)\right),
$$

$$
\texttt{skip}_t := \left\langle \boldsymbol{x}_t - \boldsymbol{\rho}^{\mathbb{P},\mathring{\pi}}, \boldsymbol{\ell}_t - \boldsymbol{\ell}_t^{\text{skip}} + \boldsymbol{b}_t^{\text{skip}}\right\rangle.
$$

By the decompositions above, we have

$$
\text{Reg}_T(\mathring{\pi}) \leq \text{Stability Term} + \text{Penalty Term} + \text{Skipping Error}
$$

$$
\leq \mathbb{E}\left[\sum_{t=1}^{T}\texttt{stab}_t\right] + \mathbb{E}\left[\sum_{t=1}^{T}\texttt{pena}_t\right] + \mathbb{E}\left[\sum_{t=1}^{T}\texttt{skip}_t\right].
$$

To control the regret, we need to bound the stability term, penalty term, and skipping error terms separately. We illustrate the analysis in the following sections.

## B.2. Loss Shifting Technique

Inspired by Jin et al. (2021), we introduce the shifted loss technique to simplify the analysis.

We first consider the skipping value function and Q function defined as:

$$
\widetilde{V}_t(s_{H+1}) := 0, \qquad\qquad\qquad \forall s_{H+1} \in \mathcal{S}_{H+1}, \tag{20}
$$

$$
\widetilde{Q}_t(s_h, a) := \widetilde{\ell}_t^{\text{skip}}(s_h, a) + \sum_{s_{h+1}\in\mathcal{S}_{h+1}} \mathbb{P}_h[s_{h+1} \mid s_h, a]\widetilde{V}_t(s_{h+1}), \qquad \forall s_h \in \mathcal{S}_h, a \in \mathcal{A}, \tag{21}
$$

$$
\widetilde{V}_t(s_h) := \sum_{a\in\mathcal{A}} \pi_t(a \mid s_h)\widetilde{Q}_t(s_h, a), \qquad\qquad\qquad \forall s_h \in \mathcal{S}_h. \tag{22}
$$

We define the shifted loss as:

$$
\begin{aligned}
\widetilde{\ell}_t^{\text{shift}}(s_h, a) &:= \widetilde{Q}_t(s_h, a) - \widetilde{V}_t(s_h) \\
&= \widetilde{\ell}_t^{\text{skip}}(s_h, a) + \sum_{s_{h+1}\in\mathcal{S}_{h+1}} \left(\mathbb{P}_h[s_{h+1} \mid s_h, a]\widetilde{V}_t(s_{h+1})\right) - \widetilde{V}_t(s_h).
\end{aligned} \tag{23}
$$

We can further write the shifted value function as the recursive definition:

$$
\widetilde{V}_{t,h}(s) = \sum_{a'\in\mathcal{A}} \pi_t(a'|s)\left(\widetilde{\ell}_t^{\text{skip}}(s, a') + \sum_{s'\in\mathcal{S}_{h+1}} \mathbb{P}[s'|s, a']\widetilde{V}_{t,h+1}(s')\right), \quad \forall s \in \mathcal{S}_h \tag{24}
$$

Following the loss-shifting trick from Jin et al. (2021), we observe that the shifted loss $\widetilde{\ell}_t^{\text{shift}}$ (defined in Eq. (23)) satisfies:

$$
\mathbb{E}\left[\left\langle \boldsymbol{x}_t - \boldsymbol{\rho}^{\mathbb{P},\mathring{\pi}}, \widetilde{\boldsymbol{\ell}}_t^{\text{shift}}\right\rangle\Big|\mathcal{F}_{t-1}\right] = \mathbb{E}\left[\left\langle \boldsymbol{x}_t - \boldsymbol{\rho}^{\mathbb{P},\mathring{\pi}}, \widetilde{\boldsymbol{\ell}}_t^{\text{skip}}\right\rangle\Big|\mathcal{F}_{t-1}\right]. \tag{25}
$$

This is because the shifting function telescopes: by the performance difference lemma, the additional value function terms cancel out when taking the inner product with the occupancy difference $\boldsymbol{x}_t - \boldsymbol{\rho}^{\mathbb{P},\mathring{\pi}}$. Therefore, for the purpose of bounding

the FTRL regret, we can equivalently analyze using the shifted loss $\widetilde{\ell}_t^{\text{shift}}$, which has the advantage that its variance is concentrated on suboptimal actions.

Another key property is that the shifted loss is concentrated on a subset of actions excluding the reference policy. To be specific, for the reference action $\mathring{a} = \mathring{\pi}(s)$, the shifted loss satisfies $\widetilde{\ell}_t^{\text{shift}}(s, \mathring{a}) \approx 0$ when $\pi_t \approx \mathring{\pi}$, as the terms approximately cancel by the performance difference argument.

Different from Jin et al. (2021) where they consider $\ell_t \in [0, 1]$, involving loss-shifting techniques faces significant challenges in handling the heavy-tailed loss.

Our novel analysis presents a new technique in bounding the first and second order moments of the shifted loss, which is crucial for the analysis of FTRL with Tsallis entropy regularizer.

**Lemma B.2** (The Uniform Bound on the Shifted Loss). *We can bound the shifted loss $\widetilde{\ell}_t^{\text{shift}}(s, a)$ by:*

$$|\widetilde{\ell}_t^{\text{shift}}(s, a)| \le \left(1 + HSA(1 + A^{1-1/\alpha})\right) \cdot C \cdot \sigma t^{1/\alpha} x_t(s, a)^{1/\alpha - 1}.$$

*Proof.* **Decomposition of the shifted loss.** Denote

$$\bar{\ell}_t^{\text{skip}}(s) := \sum_{a'} \pi_t(a'|s) \widetilde{\ell}_t^{\text{skip}}(s, a'), \tag{26}$$

as the policy-weighted loss at state $s$, and

$$\bar{V}_{t,h+1}(s) := \sum_{a'} \pi_t(a'|s) \sum_{s'} \mathbb{P}[s'|s, a'] \widetilde{V}_{t,h+1}(s') \tag{27}$$

as the policy-weighted value function at state $s$. Then, we have the decomposition of shifted loss $\widetilde{\ell}_t^{\text{shift}}(s, a)$

$$\widetilde{\ell}_t^{\text{shift}}(s, a) = \underbrace{\left(\widetilde{\ell}_t^{\text{skip}}(s, a) - \bar{\ell}_t^{\text{skip}}(s)\right)}_{=:\widetilde{\ell}_t^{\text{center}}(s, a)} + \underbrace{\left(\sum_{s'} \mathbb{P}[s'|s, a] \widetilde{V}_{t,h+1}(s') - \bar{V}_{t,h+1}(s)\right)}_{=:\Delta V_t(s, a)}, \tag{28}$$

where we set $\widetilde{\ell}_t^{\text{center}}(s, a) := \widetilde{\ell}_t^{\text{skip}}(s, a) - \bar{\ell}_t^{\text{skip}}(s)$ and $\Delta V_t(s, a) := \sum_{s'} \mathbb{P}[s'|s, a] \widetilde{V}_{t,h+1}(s') - \bar{V}_{t,h+1}(s)$.

Equipped with this decomposition, we can bound the first order moments of the shifted loss by the first order moments of the center loss and the value function difference, separately.

In the following argument, we fix $(s, a)$ and assume that $(s, a')$ is visited in the $t$-th episode for some $a' \in \mathcal{A}$. Notice that $\widetilde{\ell}_t^{\text{skip}}(s, a) \neq 0$ if and only if $(s, a) = (s, a')$.

**Bounding** $|\widetilde{\ell}_t^{\text{center}}(s, a)|$. If $a' = a$, we have

$$|\widetilde{\ell}_t^{\text{center}}(s, a)| = |\widetilde{\ell}_t^{\text{skip}}(s, a) - \pi_t(a|s) \widetilde{\ell}_t^{\text{skip}}(s, a)|$$
$$\le (1 - \pi_t(a|s)) \frac{\tau_t(s, a)}{x_t(s, a)}$$
$$\le (1 - \pi_t(a|s)) C \cdot \sigma t^{1/\alpha} x_t(s, a)^{1/\alpha - 1}$$

Otherwise, if $a' \neq a$, we have

$$|\widetilde{\ell}_t^{\text{center}}(s, a)| = |\pi_t(a'|s) \widetilde{\ell}_t^{\text{skip}}(s, a')|$$
$$\le \pi_t(a'|s) \frac{\tau_t(s, a')}{x_t(s, a')}$$
$$\le \pi_t(a'|s) C \cdot \sigma t^{1/\alpha} x_t(s, a')^{1/\alpha - 1}$$

Notice that $x_t(s, a') = x_t(s) \cdot \pi_t(a'|s)$ where $x_t(s) = \sum_{a'' \in \mathcal{A}} x_t(s, a'')$ is the marginal occupancy at state $s$. Therefore, we have

$$|\widetilde{\ell}_t^{\text{center}}(s, a)| \le C \cdot \sigma t^{1/\alpha} \cdot \pi_t(a'|s)^{1/\alpha} x_t(s)^{1/\alpha - 1}.$$

Notice that $1/\alpha - 1 \in [-1/2, 0)$, we have $x_t(s) \geq x_t(s, a)$, which implies

$$\pi_t(a'|s)^{1/\alpha} x_t(s)^{1/\alpha-1} \leq \pi_t(a'|s)^{1/\alpha} x_t(s, a)^{1/\alpha-1} \leq x_t(s, a)^{1/\alpha-1}$$

Therefore, we have

$$|\widetilde{\ell}_t^{\text{center}}(s, a)| \leq C \cdot \sigma t^{1/\alpha} \cdot x_t(s, a)^{1/\alpha-1}.$$

**Bounding $|\Delta V_t(s, a)|$.** Assume $s \in \mathcal{S}_h$ for some $h$, then $\Delta V_t(s, a)$ can be rewritten as:

$$\Delta V_t(s, a) = \sum_{s' \in \mathcal{S}_{h+1}} \left( \mathbb{P}[s'|s, a] - \sum_{a' \in \mathcal{A}} \pi_t(a'|s) \mathbb{P}[s'|s, a'] \right) \widetilde{V}_{t,h+1}(s').$$

Notice that the weights $(\mathbb{P}[s'|s, a] - \sum_{a'} \pi_t(a'|s) \mathbb{P}[s'|s, a'])$ sum to zero over $s'$. Expanding $\widetilde{V}_{t,h+1}(s')$ using Eq. (24), we get a telescoping sum:

$$\Delta V_t(s, a) = \sum_{h'=h+1}^{H} \sum_{s' \in \mathcal{S}_{h'}, a' \in \mathcal{A}} \omega_t^{h \to h'}(s, a; s', a') \cdot \widetilde{\ell}_t^{\text{skip}}(s', a'), \tag{29}$$

where $\omega_t^{h \to h'}(s, a; s', a')$ is the differential reachability weight from $(s, a)$ at layer $h$ to $(s', a')$ at layer $h'$, defined as:

$$\omega_t^{h \to h'}(s, a; s', a') := \left( \mathbb{P}_{\pi_t}^{s,a}(s_{h'} = s') - \sum_{a''} \pi_t(a''|s) \mathbb{P}_{\pi_t}^{s,a''}(s_{h'} = s') \right) \cdot \pi_t(a'|s'),$$

where $\mathbb{P}_{\pi_t}^{s,a}(s_{h'} = s')$ is the probability of reaching state $s'$ from state $s$ under policy $\pi_t$ starting from action $a$. We further have the following identity: for every $s' \in \mathcal{S}_{h'}, a' \in \mathcal{A}$, and $h' > h$,

$$x_t(s', a') = x_t(s') \cdot \pi_t(a'|s') = \sum_{s'' \in \mathcal{S}_h, a'' \in \mathcal{A}} x_t(s'', a'') \cdot \mathbb{P}_{\pi_t}^{s'',a''}(s_{h'} = s') \cdot \pi_t(a'|s').$$

Recall the definition of $\omega_t^{h \to h'}(s, a; s', a')$, we have $|\omega_t^{h \to h'}(s, a; s', a')| \leq 2\pi_t(a'|s')$. Therefore, we have

$$\begin{aligned}
|\Delta V_t(s, a)| &\leq \sum_{h'=h+1}^{H} \sum_{s' \in \mathcal{S}_{h'}, a' \in \mathcal{A}} |\omega_t^{h \to h'}(s, a; s', a')| \cdot |\widetilde{\ell}_t^{\text{skip}}(s', a')| \\
&\leq \sum_{h'=h+1}^{H} \sum_{s' \in \mathcal{S}_{h'}, a' \in \mathcal{A}} |\omega_t^{h \to h'}(s, a; s', a')| \cdot C \cdot \sigma t^{1/\alpha} x_t(s', a')^{1/\alpha-1}.
\end{aligned}$$

We need to bound

$$\sum_{h'=h+1}^{H} \sum_{s' \in \mathcal{S}_{h'}, a' \in \mathcal{A}} |\omega_t^{h \to h'}(s, a; s', a')| \cdot x_t(s', a')^{1/\alpha-1}.$$

By triangle inequality, and $u \mapsto u^{1/\alpha-1}$ is decreasing on $u \in [0, 1]$, we have

$$\begin{aligned}
&|\omega_t^{h \to h'}(s, a; s', a')| \cdot x_t(s', a')^{1/\alpha-1} \\
&\leq \pi_t(a'|s') \cdot \mathbb{P}_{\pi_t}^{s,a}(s_{h'} = s') \cdot \left( x_t(s, a) \cdot \mathbb{P}_{\pi_t}^{s,a}(s_{h'} = s') \cdot \pi_t(a'|s') \right)^{1/\alpha-1} \\
&\quad + \sum_{a'' \in \mathcal{A}} \pi_t(a'|s') \cdot \pi_t(a''|s) \cdot \mathbb{P}_{\pi_t}^{s,a''}(s_{h'} = s') \cdot \left( x_t(s, a'') \cdot \mathbb{P}_{\pi_t}^{s,a''}(s_{h'} = s') \cdot \pi_t(a'|s') \right)^{1/\alpha-1} \\
&= x_t(s, a)^{1/\alpha-1} \cdot \left( \pi_t(a'|s') \cdot \mathbb{P}_{\pi_t}^{s,a}(s_{h'} = s') \right)^{1/\alpha} \\
&\quad + \sum_{a'' \in \mathcal{A}} \pi_t(a''|s) \cdot x_t(s, a'')^{1/\alpha-1} \cdot \left( \pi_t(a'|s') \cdot \mathbb{P}_{\pi_t}^{s,a''}(s_{h'} = s') \right)^{1/\alpha}.
\end{aligned}$$

Since $u \mapsto u^{1/\alpha}$ is increasing on $u \in [0, 1]$, we have,

$$
\begin{aligned}
|\omega_t^{h \to h'}(s, a; s', a')| \cdot x_t(s', a')^{1/\alpha - 1} &\leq x_t(s, a)^{1/\alpha - 1} + \sum_{a'' \in \mathcal{A}} \pi_t(a''|s) \cdot x_t(s, a'')^{1/\alpha - 1} \\
&\leq x_t(s, a)^{1/\alpha - 1} + \sum_{a'' \in \mathcal{A}} \pi_t(a''|s) \cdot (x_t(s) \cdot \pi_t(a''|s))^{1/\alpha - 1} \\
&= x_t(s, a)^{1/\alpha - 1} + \sum_{a'' \in \mathcal{A}} \pi_t(a''|s)^{1/\alpha} \cdot x_t(s)^{1/\alpha - 1} \\
&\leq x_t(s, a)^{1/\alpha - 1} + \sum_{a'' \in \mathcal{A}} \pi_t(a''|s)^{1/\alpha} \cdot x_t(s, a)^{1/\alpha - 1} \\
&\leq (1 + A^{1 - 1/\alpha}) x_t(s, a)^{1/\alpha - 1}.
\end{aligned}
$$

The last inequality is derived by maximizing over $\pi_t(s) \in \Delta(\mathcal{A})$, and the maximum is achieved when $\pi_t(s)$ is a uniform distribution. Finally, we can bound $|\Delta V_t(s, a)|$ by:

$$
\begin{aligned}
|\Delta V_t(s, a)| &\leq C \cdot \sigma t^{1/\alpha} \sum_{h'=h+1}^{H} \sum_{s' \in \mathcal{S}_{h'}, a' \in \mathcal{A}} |\omega_t^{h \to h'}(s, a; s', a')| \cdot x_t(s', a')^{1/\alpha - 1} \\
&\leq HSA(1 + A^{1 - 1/\alpha}) \cdot C \cdot \sigma t^{1/\alpha} x_t(s, a)^{1/\alpha - 1}.
\end{aligned}
$$

Therefore, we can bound the first order moments of the shifted loss by:

$$
\begin{aligned}
|\widetilde{\ell}_t^{\text{shift}}(s, a)| &\leq |\widetilde{\ell}_t^{\text{center}}(s, a)| + |\Delta V_t(s, a)| \\
&\leq C \cdot \sigma t^{1/\alpha} \cdot x_t(s, a)^{1/\alpha - 1} + HSA(1 + A^{1 - 1/\alpha}) \cdot C \cdot \sigma t^{1/\alpha} x_t(s, a)^{1/\alpha - 1} \\
&\leq (1 + HSA(1 + A^{1 - 1/\alpha})) \cdot C \cdot \sigma t^{1/\alpha} \cdot x_t(s, a)^{1/\alpha - 1}.
\end{aligned}
$$

$\square$

We highlight that this decomposition is crucial for the analysis of heavy-tailed shifted loss $\widetilde{\ell}_t^{\text{shift}}(s, a)$. This technique could be of independent interest.

Now we analyze the second-order moments of the shifted loss. Following the analysis in Jin et al. (2021, Lemma A.1.3), we have the lemma:

**Lemma B.3** (Second-order Moments of Shifted Loss). *We can bound the second-order moments of the shifted loss by*

$$
\mathbb{E}[\widetilde{\ell}_t^{\text{shift}}(s, a)^2 \mid \mathcal{F}_{t-1}] \leq 4H^2(1 - \pi_t(a|s)) \cdot C^{2-\alpha} \sigma^2 t^{2/\alpha - 1} x_t(s, a)^{2/\alpha - 2}. \tag{30}
$$

*Proof.* We utilize the definition of the shifted loss in Eq. (23). We have $\widetilde{\ell}_t^{\text{shift}}(s, a) = \widetilde{Q}_t(s, a) - \widetilde{V}_t(s) = (1 - \pi_t(a|s))\widetilde{Q}_t(s, a) - \sum_{b \neq a} \pi_t(b|s)\widetilde{Q}_t(s, b)$.

Inspired by Jin et al. (2021, Lemma A.1.3), we first bound the second-order moments of the shifted Q function.

Fixed state-action pair $(s, a)$, and denote $h$ to be the index of the layer that $(s, a)$ is at, i.e., $s \in \mathcal{S}_h$. Define $q_t(s', a'|s, a)$ is the probability of visiting $(s', a')$ after taking action $a$ from $s$ and following the policy $\pi_t$. We have

$$
\begin{aligned}
\mathbb{E}\left[\widetilde{Q}_t(s, a)^2 \mid \mathcal{F}_{t-1}\right] &= \mathbb{E}\left[\left(\sum_{h' \geq h} \sum_{s' \in \mathcal{S}_{h'}, a' \in \mathcal{A}} q_t(s', a'|s, a)\widetilde{\ell}_t^{\text{skip}}(s', a')\right)^2 \mid \mathcal{F}_{t-1}\right] \\
&\leq H \cdot \sum_{h' \geq h} \sum_{s' \in \mathcal{S}_{h'}, a' \in \mathcal{A}} q_t(s', a'|s, a)^2 \mathbb{E}\left[\widetilde{\ell}_t^{\text{skip}}(s', a')^2 \cdot \mathbb{1}[(s', a') \text{ is visited}] \mid \mathcal{F}_{t-1}\right] \\
&= H \cdot \sum_{h' \geq h} \sum_{s' \in \mathcal{S}_{h'}, a' \in \mathcal{A}} \frac{q_t(s', a'|s, a)^2}{x_t(s', a')^2} \mathbb{E}\left[\ell_t^{\text{skip}}(s', a')^2 \cdot \mathbb{1}[(s', a') \text{ is visited}] \mid \mathcal{F}_{t-1}\right] \\
&\leq H \cdot \sum_{h' \geq h} \sum_{s' \in \mathcal{S}_{h'}, a' \in \mathcal{A}} \frac{q_t(s', a'|s, a)^2}{x_t(s', a')} \mathbb{E}\left[|\ell_t(s', a')|^\alpha \tau_t(s', a')^{2-\alpha} \mid \mathcal{F}_{t-1}\right],
\end{aligned}
$$

where the first inequality is due to Cauchy-Schwarz across layers (giving the factor $H$ from at most $H$ layers) and the orthogonality of single-layer visit indicators: within each layer $h'$, $\mathbb{1}[(s', a') \text{ is visited}] \cdot \mathbb{1}[(s'', a'') \text{ is visited}] = 0$ for distinct $(s', a') \neq (s'', a'')$ at layer $h'$ (only one state-action is visited per layer). Notice that we can bound the second-order moments of the skipped loss by:

$$\mathbb{E}\left[|\ell_t(s', a')|^\alpha \tau_t(s', a')^{2-\alpha} \mid \mathcal{F}_{t-1}\right] \leq C^{2-\alpha} \sigma^2 t^{2/\alpha-1} x_t(s', a')^{2/\alpha-1}.$$

Therefore, we get

$$\mathbb{E}\left[\widetilde{Q}_t(s, a)^2 \mid \mathcal{F}_{t-1}\right] \leq H \cdot C^{2-\alpha} \sigma^2 t^{2/\alpha-1} \sum_{h' \geq h} \sum_{s' \in \mathcal{S}_{h'}, a' \in \mathcal{A}} q_t(s', a'|s, a)^2 x_t(s', a')^{2/\alpha-2}.$$

Notice that

$$x_t(s', a') = \sum_{s'' \in \mathcal{S}_h, a''} x_t(s'', a'') \cdot q_t(s', a'|s'', a'') \geq x_t(s, a) q_t(s', a'|s, a),$$

and $u \mapsto u^{2/\alpha-2}$ is decreasing on $u \in [0, 1]$ since $2/\alpha - 2 \in [-1, 0)$, we have

$$\mathbb{E}\left[\widetilde{Q}_t(s, a)^2 \mid \mathcal{F}_{t-1}\right] \leq H \cdot C^{2-\alpha} \sigma^2 t^{2/\alpha-1} \sum_{h' \geq h} \sum_{s' \in \mathcal{S}_{h'}, a' \in \mathcal{A}} q_t(s', a'|s, a)^{2/\alpha} x_t(s, a)^{2/\alpha-2}$$

$$\leq H^2 \cdot C^{2-\alpha} \sigma^2 t^{2/\alpha-1} x_t(s, a)^{2/\alpha-2},$$

where the last step uses $\sum_{s' \in \mathcal{S}_{h'}, a' \in \mathcal{A}} q_t(s', a'|s, a)^{2/\alpha} \leq \sum_{s', a'} q_t(s', a'|s, a) = 1$ per layer (since $q_t \leq 1$ and $2/\alpha \geq 1$ for $\alpha \in (1, 2]$ implies $q^{2/\alpha} \leq q$), and summing over $H - h + 1 \leq H$ layers from $h$ to $H$ gives the additional $H$ factor. For the second-order moments of the shifted value function, we have

$$\mathbb{E}\left[\left(\sum_{b \neq a} \pi_t(b|s) \widetilde{Q}_t(s, b)\right)^2 \Bigg| \mathcal{F}_{t-1}\right] = \mathbb{E}\left[\left(\sum_{h' \geq h} \sum_{s' \in \mathcal{S}_{h'}, a' \in \mathcal{A}} \left(\sum_{b \neq a} \pi_t(b|s) q_t(s', a'|s, b)\right) \cdot \widetilde{\ell}_t^{\text{skip}}(s', a')\right)^2 \Bigg| \mathcal{F}_{t-1}\right]$$

$$\leq H \cdot \sum_{h' \geq h} \sum_{s' \in \mathcal{S}_{h'}, a' \in \mathcal{A}} \left(\sum_{b \neq a} \pi_t(b|s) q_t(s', a'|s, b)\right)^2 \mathbb{E}\left[\widetilde{\ell}_t^{\text{skip}}(s', a')^2 \cdot \mathbb{1}[(s', a') \text{ is visited}] \mid \mathcal{F}_{t-1}\right],$$

where this inequality follows from the same Cauchy-Schwarz across layers and per-layer indicator orthogonality argument as above. Consider the term $\mathbb{E}\left[\widetilde{\ell}_t^{\text{skip}}(s', a')^2 \cdot \mathbb{1}[(s', a') \text{ is visited}] \mid \mathcal{F}_{t-1}\right]$, we have

$$\mathbb{E}\left[\widetilde{\ell}_t^{\text{skip}}(s', a')^2 \cdot \mathbb{1}[(s', a') \text{ is visited}] \mid \mathcal{F}_{t-1}\right] \leq x_t(s', a')^{-1} \cdot C^{2-\alpha} \sigma^2 t^{2/\alpha-1} x_t(s', a')^{2/\alpha-1}.$$

Therefore, we have

$$\mathbb{E}\left[\left(\sum_{b \neq a} \pi_t(b|s) \widetilde{Q}_t(s, b)\right)^2 \mid \mathcal{F}_{t-1}\right]$$

$$\leq H \cdot C^{2-\alpha} \sigma^2 t^{2/\alpha-1} \sum_{h' \geq h} \sum_{s' \in \mathcal{S}_{h'}, a' \in \mathcal{A}} \left(\sum_{b \neq a} \pi_t(b|s) q_t(s', a'|s, b)\right)^2 x_t(s', a')^{2/\alpha-2}$$

By the decomposition of $x_t(s', a')$, we have

$$\sum_{h' \geq h} \sum_{s' \in \mathcal{S}_{h'}, a' \in \mathcal{A}} \left( \sum_{b \neq a} \pi_t(b|s) q_t(s', a'|s, b) \right)^2 x_t(s', a')^{2/\alpha - 2}$$

$$\leq \sum_{h' \geq h} \sum_{s' \in \mathcal{S}_{h'}, a' \in \mathcal{A}} \left( \sum_{b \neq a} \pi_t(b|s) q_t(s', a'|s, b) \right)^2 \cdot \left( \sum_{b \neq a} x_t(s, b) q_t(s', a'|s, b) \right)^{2/\alpha - 2}$$

$$= \frac{1}{x_t(s)} \sum_{h' \geq h} \sum_{s' \in \mathcal{S}_{h'}, a' \in \mathcal{A}} \left( \sum_{b \neq a} x_t(s, b) q_t(s', a'|s, b) \right)^{2/\alpha - 1} \cdot \left( \sum_{b \neq a} \pi_t(b|s) q_t(s', a'|s, b) \right)$$

$$\leq \frac{1}{x_t(s)} \sum_{h' \geq h} \sum_{s' \in \mathcal{S}_{h'}, a' \in \mathcal{A}} \left( \sum_{b \neq a} x_t(s, b) \right)^{2/\alpha - 1} \cdot \left( \sum_{b \neq a} \pi_t(b|s) q_t(s', a'|s, b) \right)$$

$$\leq \frac{1}{x_t(s)} \sum_{h' \geq h} \sum_{s' \in \mathcal{S}_{h'}, a' \in \mathcal{A}} (x_t(s))^{2/\alpha - 1} \cdot \left( \sum_{b \neq a} \pi_t(b|s) q_t(s', a'|s, b) \right)$$

$$= \sum_{h' \geq h} \sum_{s' \in \mathcal{S}_{h'}, a' \in \mathcal{A}} x_t(s)^{2/\alpha - 2} \cdot \left( \sum_{b \neq a} \pi_t(b|s) q_t(s', a'|s, b) \right)$$

$$\leq H \cdot x_t(s, a)^{2/\alpha - 2} \cdot \sum_{b \neq a} \pi_t(b|s)$$

$$= H \cdot (1 - \pi_t(a|s)) x_t(s, a)^{2/\alpha - 2}.$$

Finally, we have

$$\mathbb{E} \left[ \left( \sum_{b \neq a} \pi_t(b|s) \widetilde{Q}_t(s, b) \right)^2 \mid \mathcal{F}_{t-1} \right] \leq (1 - \pi_t(a|s)) \cdot H^2 C^{2 - \alpha} \sigma^2 t^{2/\alpha - 1} x_t(s, a)^{2/\alpha - 2}.$$

This induced the bound on the second-order moments of the shifted loss:

$$\mathbb{E} \left[ \widetilde{\ell}_t^{\text{shift}}(s, a)^2 \mid \mathcal{F}_{t-1} \right] \leq 2 \mathbb{E} \left[ (1 - \pi_t(a|s))^2 \widetilde{Q}_t(s, a)^2 \mid \mathcal{F}_{t-1} \right] + 2 \mathbb{E} \left[ \left( \sum_{b \neq a} \pi_t(b|s) \widetilde{Q}_t(s, b) \right)^2 \mid \mathcal{F}_{t-1} \right]$$

$$\leq 4 H^2 (1 - \pi_t(a|s)) \cdot C^{2 - \alpha} \sigma^2 t^{2/\alpha - 1} x_t(s, a)^{2/\alpha - 2}.$$

$\square$

## B.3. Bounding Stability Term

To begin with, we first state the local stability of Tsallis entropy, which is a famous result for $\alpha'$-Tsallis entropy, introduced in Ito et al. (2024, Lemma 9). We rewrite it as follows for completeness.

**Lemma B.4** (Local stability of Tsallis entropy). *Consider the $\alpha'$-Tsallis entropy $\psi(\boldsymbol{x}) = -\frac{1}{\alpha'} \sum_i (x_i^{\alpha'} - x_i)$ defined on the simplex, with $\alpha' \in [1/2, 1)$. Let $\boldsymbol{x}$ be a point in the simplex and $\boldsymbol{g}$ be the vector of the gradient of $\psi(\cdot)$. Let $\tilde{\boldsymbol{y}}$ be the unconstrained update defined by $\nabla \psi(\tilde{\boldsymbol{y}}) = \nabla \psi(\boldsymbol{x}) - \boldsymbol{g}$. If the gradient is small enough such that $|g_i| \leq \frac{1 - \alpha'}{4} x_i^{\alpha' - 1}$ for all indices $i$, then*

$$\langle \boldsymbol{g}, \boldsymbol{x} - \tilde{\boldsymbol{y}} \rangle - D_\psi(\tilde{\boldsymbol{y}}, \boldsymbol{x}) \leq \frac{2}{1 - \alpha'} \sum_i x_i^{2 - \alpha'} g_i^2. \tag{31}$$

*Proof.* Since $\nabla\psi(\cdot)_i$ is the map $u \mapsto \frac{1}{\alpha'} - u^{\alpha'-1}$, the optimality condition $\nabla\psi(\tilde{\boldsymbol{y}})_i = \nabla\psi(\boldsymbol{x})_i - g_i$ implies $\tilde{y}_i^{\alpha'-1} = x_i^{\alpha'-1} + g_i$. The condition $|g_i| \leq \frac{1-\alpha'}{4} x_i^{\alpha'-1}$ implies

$$\left(1 - \frac{1-\alpha'}{4}\right) x_i^{\alpha'-1} \leq \tilde{y}_i^{\alpha'-1} \leq \left(1 + \frac{1-\alpha'}{4}\right) x_i^{\alpha'-1}.$$

Since $\alpha' < 1$, the function $u \mapsto u^{\alpha'-1}$ is strictly decreasing. Using elementary inequalities for the power function with exponents in $(-\infty, -2]$ (as implied by $\alpha' \in [1/2, 1)$), one can verify that the multiplicative bounds above imply $|\tilde{y}_i - x_i| \leq \frac{1}{2} x_i$.

By Taylor's theorem, $D_\psi(\tilde{\boldsymbol{y}}, \boldsymbol{x}) = \frac{1}{2}(\tilde{\boldsymbol{y}} - \boldsymbol{x})^\top \nabla^2\psi(\boldsymbol{z})(\tilde{\boldsymbol{y}} - \boldsymbol{x})$ for some $\boldsymbol{z}$ on the segment connecting $\boldsymbol{x}$ and $\tilde{\boldsymbol{y}}$. Since $|\tilde{y}_i - x_i| \leq \frac{1}{2} x_i$, we have $z_i \leq \frac{3}{2} x_i$. The Hessian is diagonal with entries $\nabla^2\psi(\boldsymbol{z})_{ii} = (1-\alpha')z_i^{\alpha'-2}$. Since $\alpha' - 2 < 0$, we have $z_i^{\alpha'-2} \geq (\frac{3}{2})^{\alpha'-2} x_i^{\alpha'-2} \geq \frac{1}{2} x_i^{\alpha'-2}$. Thus,

$$D_\psi(\tilde{\boldsymbol{y}}, \boldsymbol{x}) \geq \frac{1-\alpha'}{4} \sum_i x_i^{\alpha'-2}(\tilde{y}_i - x_i)^2.$$

Applying Young's inequality $ab \leq \frac{a^2}{2\lambda} + \frac{\lambda b^2}{2}$ with $\lambda_i = \frac{1-\alpha'}{2} x_i^{\alpha'-2}$ to the term $\langle \boldsymbol{g}, \boldsymbol{x} - \tilde{\boldsymbol{y}} \rangle$ yields Eq. (31). □

Equipped with the local stability of Tsallis entropy, we now bound the stability term via the following lemma.

**Lemma B.5.** *The stability term is bounded as*

$$\mathtt{stab}_t \leq \frac{4\alpha^2}{\alpha-1}\eta_t \sum_{(s,a)\in\mathcal{S}\times\mathcal{A}} x_t(s,a)^{2-1/\alpha} \left((\widetilde{\ell}_t^{\text{shift}}(s,a))^2 + (b_t^{\text{skip}}(s,a))^2\right).$$

*Proof.* Recall the regularizer $\Psi_t(\boldsymbol{x}) = -\frac{1}{\eta_t}\sum_{(s,a)} x(s,a)^{1/\alpha}$ where $\alpha \in (1, 2]$.

Define $\alpha' := 1/\alpha \in [1/2, 1)$. The standard $\alpha'$-Tsallis entropy is $\psi(\boldsymbol{x}) = -\frac{1}{\alpha'}\sum_{(s,a)}(x(s,a)^{\alpha'} - x(s,a))$. We have $\Psi_t(\boldsymbol{x}) = \frac{1}{\alpha\eta_t}\psi(\boldsymbol{x}) + \text{linear terms}$, which implies

$$D_{\Psi_t}(\boldsymbol{x}, \boldsymbol{y}) = \frac{1}{\alpha\eta_t}D_\psi(\boldsymbol{x}, \boldsymbol{y}) = \frac{\alpha'}{\eta_t}D_\psi(\boldsymbol{x}, \boldsymbol{y}).$$

We apply Lemma B.4 with $\boldsymbol{x} = \boldsymbol{x}_t$ and the scaled loss gradient $\boldsymbol{g} = \alpha\eta_t(\widetilde{\ell}_t^{\text{shift}} - \boldsymbol{b}_t^{\text{skip}})$. First, we verify the condition $|g(s,a)| \leq \frac{1-\alpha'}{4} x_t(s,a)^{\alpha'-1}$.

Here we use the uniform bound on the shifted loss Lemma 4.3 and the definition of the skipping bonus $\boldsymbol{b}_t^{\text{skip}}$:

$$\begin{aligned}
|g(s,a)| &\leq \alpha\eta_t \cdot \left(|\widetilde{\ell}_t^{\text{shift}}(s,a)| + |b_t^{\text{skip}}(s,a)|\right) \\
&\leq \alpha\eta_t \cdot (1 + HSA(1 + A^{1-1/\alpha})) \cdot C \cdot \sigma t^{1/\alpha} \cdot x_t(s,a)^{1/\alpha-1} + \alpha\eta_t \cdot C^{1-\alpha} \cdot \sigma t^{1/\alpha-1} \cdot x_t(s,a)^{1/\alpha-1} \\
&= \alpha \cdot \beta \cdot (C + HSA(1 + A^{1-1/\alpha})C + C^{1-\alpha}/t) \cdot x_t(s,a)^{1/\alpha-1},
\end{aligned}$$

Since we set $C = \left(\frac{(1+HSA+HSA^{2-1/\alpha})}{\alpha-1}\right)^{-1/\alpha}$ in Eq. (17), denoting $R := 1 + HSA + HSA^{2-1/\alpha}$ so $C = ((\alpha-1)/R)^{1/\alpha}$, we have

$$\begin{aligned}
&\beta \cdot (C + HSA(1 + A^{1-1/\alpha})C + C^{1-\alpha}/t) \\
&\leq \beta \cdot \left(\alpha(\alpha-1)^{1/\alpha-1} \cdot R^{1-1/\alpha}\right) \\
&\leq \frac{\alpha-1}{4\alpha^2},
\end{aligned}$$

by the definition of $\beta$ in Eq. (18). Therefore, we get $|g(s,a)| \leq \frac{1-\alpha'}{4} x_t(s,a)^{\alpha'-1}$.

The FTRL stability term is $\texttt{stab}_t = \langle \widetilde{\ell}_t^{\text{shift}} - \boldsymbol{b}_t^{\text{skip}}, \boldsymbol{x}_t - \boldsymbol{x}_{t+1} \rangle - D_{\Psi_t}(\boldsymbol{x}_{t+1}, \boldsymbol{x}_t)$. Using the relation $D_{\Psi_t} = \frac{\alpha'}{\eta_t} D_\psi$, we write

$$\texttt{stab}_t = \frac{\alpha'}{\eta_t} \left[ \left\langle \frac{\eta_t}{\alpha'}(\widetilde{\ell}_t^{\text{shift}} - \boldsymbol{b}_t^{\text{skip}}), \boldsymbol{x}_t - \boldsymbol{x}_{t+1} \right\rangle - D_\psi(\boldsymbol{x}_{t+1}, \boldsymbol{x}_t) \right].$$

Let $F(\boldsymbol{y}) := \langle \boldsymbol{g}, \boldsymbol{x}_t - \boldsymbol{y} \rangle - D_\psi(\boldsymbol{y}, \boldsymbol{x}_t)$. The quantity in the brackets is $F(\boldsymbol{x}_{t+1})$. If $\tilde{\boldsymbol{y}}$ (the unconstrained update) maximizes $F$, we have $F(\boldsymbol{x}_{t+1}) \leq F(\tilde{\boldsymbol{y}})$. Applying Lemma B.4:

$$\texttt{stab}_t \leq \frac{\alpha'}{\eta_t} \cdot \frac{2}{1-\alpha'} \sum_{(s,a)} x_t(s,a)^{2-\alpha'} \frac{\eta_t^2}{\alpha'^2} \left( \widetilde{\ell}_t^{\text{shift}}(s,a) - b_t^{\text{skip}}(s,a) \right)^2.$$

Simplifying with $\alpha' = 1/\alpha$ and $\frac{1}{\alpha'(1-\alpha')} = \frac{\alpha^2}{\alpha-1}$. Notice that $(x-y)^2 \leq 2x^2 + 2y^2$, we get

$$\texttt{stab}_t \leq \frac{4\alpha^2}{\alpha-1}\eta_t \sum_{(s,a)\in\mathcal{S}\times\mathcal{A}} x_t(s,a)^{2-1/\alpha} \left( (\widetilde{\ell}_t^{\text{shift}}(s,a))^2 + (b_t^{\text{skip}}(s,a))^2 \right),$$

which completes the proof. $\qquad\square$

In the following statement, we separately control the second moment of $\widetilde{\ell}_t^{\text{shift}}(s,a)$ and $b_t^{\text{skip}}(s,a)$ weighted by $x_t(s,a)^{2-1/\alpha}$ and finally derive the control on the expectations of $\texttt{stab}_t$. The key challenge is to provide the summation over $(s,a) \in \mathcal{S} \times \mathcal{A}$ without the reference action $\mathring{\pi}(s)$. We denote

$$\texttt{loss}_t := \eta_t \sum_{(s,a)\in\mathcal{S}\times\mathcal{A}} x_t(s,a)^{2-1/\alpha} \cdot \left( \widetilde{\ell}_t^{\text{shift}}(s,a) \right)^2, \tag{32}$$

$$\texttt{bonus}_t := \eta_t \sum_{(s,a)\in\mathcal{S}\times\mathcal{A}} x_t(s,a)^{2-1/\alpha} \left( b_t^{\text{skip}}(s,a) \right)^2. \tag{33}$$

### B.3.1. CONTROL ON THE SECOND MOMENT OF THE SHIFTED LOSS

**Lemma B.6.** *For the shifted loss part of the stability term $\texttt{stab}_t$, we have*

$$\mathbb{E}\left[ \texttt{loss}_t \mid \mathcal{F}_{t-1} \right] \leq 8H^2 \beta C^{2-\alpha} \cdot \sigma t^{\frac{1}{\alpha}-1} \sum_{(s,a)\neq(s,\mathring{\pi}(s))} x_t(s,a)^{1/\alpha}.$$

*Proof.* We analyze the stability term with the shifted loss $\widetilde{\ell}_t^{\text{shift}}$. The key insight is that the shifted loss has a special structure that eliminates the contribution from reference actions. We apply the bounds for second-order moment of shifted loss in Lemma 4.4 and get

$$\mathbb{E}\left[ \eta_t \cdot \sum_{(s,a)\in\mathcal{S}\times\mathcal{A}} x_t(s,a)^{2-1/\alpha}(\widetilde{\ell}_t^{\text{shift}}(s,a))^2 \mid \mathcal{F}_{t-1} \right]$$

$$= \eta_t \cdot \sum_{(s,a)\in\mathcal{S}\times\mathcal{A}} x_t(s,a)^{2-1/\alpha}\mathbb{E}\left[ \left( \widetilde{\ell}_t^{\text{shift}}(s,a) \right)^2 \mid \mathcal{F}_{t-1} \right]$$

$$\leq 4H^2 \cdot \eta_t \cdot C^{2-\alpha} \cdot \sigma^2 t^{2/\alpha-1} \cdot \sum_{(s,a)\in\mathcal{S}\times\mathcal{A}} x_t(s,a)^{2-1/\alpha} \cdot (1-\pi_t(a|s))x_t(s,a)^{2/\alpha-2}$$

$$\leq 4H^2 \cdot \beta C^{2-\alpha} \cdot \sigma t^{1/\alpha-1} \cdot \sum_{(s,a)\in\mathcal{S}\times\mathcal{A}} (1-\pi_t(a|s))x_t(s,a)^{1/\alpha}.$$

We note that for fixed $s$ and $\mathring{a} = \mathring{\pi}(s)$,

$$
\begin{aligned}
(1 - \pi_t(\mathring{a}|s))x_t(s,\mathring{a})^{1/\alpha} &\leq \sum_{a \neq \mathring{a}} \pi_t(a|s)x_t(s)^{1/\alpha} \\
&\leq \sum_{a \neq \mathring{a}} x_t(s,a)^{1/\alpha} \cdot \pi_t(a|s)^{1-1/\alpha} \\
&\leq \sum_{a \neq \mathring{a}} x_t(s,a)^{1/\alpha}.
\end{aligned}
$$

Therefore, we get the result

$$
\mathbb{E}\left[\eta_t \cdot \sum_{(s,a) \in \mathcal{S} \times \mathcal{A}} x_t(s,a)^{2-1/\alpha}(\widetilde{\ell}_t^{\text{shift}}(s,a))^2 \mid \mathcal{F}_{t-1}\right] \leq 8H^2 \cdot \beta C^{2-\alpha} \cdot \sigma t^{1/\alpha-1} \cdot \sum_{(s,a) \neq (s,\mathring{\pi}(s))} x_t(s,a)^{1/\alpha}.
$$

$\square$

### B.3.2. CONTROL ON THE SECOND MOMENT OF THE SKIPPING BONUS

**Lemma B.7.** *For the skipping bonus part of the stability term* $\texttt{stab}_t$*, we have*

$$
\mathbb{E}\left[\sum_{t=1}^{T} bonus_t\right] \leq 2SA^{1-1/\alpha}\beta C^{2-2\alpha} \cdot \sigma \cdot (1 + \log(T))
$$

*Proof.* Recall the definition of the skipping bonus:

$$
b_t^{\text{skip}}(s,a) = C^{1-\alpha} \cdot \sigma t^{1/\alpha-1}x_t(s,a)^{1/\alpha-1}.
$$

Substituting this into the definition of $\texttt{bonus}_t$:

$$
\begin{aligned}
\texttt{bonus}_t &= C^{2-2\alpha} \cdot \eta_t \cdot \sigma^2 t^{2/\alpha-2} \sum_{(s,a) \in \mathcal{S} \times \mathcal{A}} x_t(s,a)^{1/\alpha} \\
&\leq \beta C^{2-2\alpha} \cdot \sigma t^{1/\alpha-2} \cdot \sum_{h=1}^{H} (S_h A)^{1-1/\alpha}, \\
&\leq \beta C^{2-2\alpha} \cdot \sigma t^{1/\alpha-2} \cdot SA^{1-1/\alpha}.
\end{aligned}
$$

The first inequality is obtained by observing that the sum reaches its maximum when $x_t$ is uniform in each layer. Notice that $1/\alpha - 2 \in [-3/2, -1)$ for $\alpha \in (1, 2]$. Since $t^{1/\alpha-1} \leq 1$ for $t \geq 1$, we have $t^{1/\alpha-2} \leq t^{-1}$, hence

$$
\begin{aligned}
\sum_{t=1}^{T} \texttt{bonus}_t &\leq SA^{1-1/\alpha} \cdot \beta C^{2-2\alpha} \cdot \sigma \sum_{t=1}^{T} t^{1/\alpha-2} \\
&\leq SA^{1-1/\alpha} \cdot \beta C^{2-2\alpha} \cdot \sigma \sum_{t=1}^{T} t^{-1} \\
&\leq 2SA^{1-1/\alpha} \cdot \beta C^{2-2\alpha} \cdot \sigma(1 + \log(T)).
\end{aligned}
$$

$\square$

### B.4. Bounding Penalty Term

To begin with this section, we need to analyze the occupancy difference layer by layer. Following Lemma 20 in Jin et al. (2021), we derive the following lemma.

**Lemma B.8** (Suboptimal Mass Propagation). *Let $\pi$ be any policy and $\pi^\dagger$ be a deterministic policy. Let $q^\pi(s,a)$ and $q^{\pi^\dagger}(s,a)$ be the state-action occupancy measures, and $q^\pi(s) = \sum_{a \in \mathcal{A}} q^\pi(s,a)$ and $q^{\pi^\dagger}(s) = \sum_{a \in \mathcal{A}} q^{\pi^\dagger}(s,a)$ be the state marginals. Then,*

$$\sum_{s \in \mathcal{S}} (q^\pi(s) - q^{\pi^\dagger}(s))_+ \le H \cdot \sum_{(s,a): a \ne \pi^\dagger(s)} q^\pi(s,a). \tag{34}$$

*Proof.* Define $f(s) := (q^\pi(s) - q^{\pi^\dagger}(s))_+$. Since $\pi^\dagger$ is deterministic, $q^{\pi^\dagger}(s,a) = 0$ for all $a \ne \pi^\dagger(s)$. For any state $s$ at layer $h > 0$, we decompose the state occupancy based on the previous layer:

$$q^\pi(s) - q^{\pi^\dagger}(s) = \sum_{s' \in \mathcal{S}_{h-1}} \sum_{a' \in \mathcal{A}} \mathbb{P}[s|s',a'] q^\pi(s',a') - \sum_{s' \in \mathcal{S}_{h-1}} \mathbb{P}[s|s', \pi^\dagger(s')] q^{\pi^\dagger}(s')$$

$$\le \sum_{s' \in \mathcal{S}_{h-1}} \mathbb{P}[s|s', \pi^\dagger(s')] (q^\pi(s') - q^{\pi^\dagger}(s')) + \sum_{s' \in \mathcal{S}_{h-1}} \sum_{a' \ne \pi^\dagger(s')} \mathbb{P}[s|s',a'] q^\pi(s',a').$$

Here, we used the substitution $q^\pi(s', \pi^\dagger(s')) = q^\pi(s') - \sum_{a' \ne \pi^\dagger(s')} q^\pi(s',a')$, and $q^\pi(s', \pi^\dagger(s')) \le q^\pi(s')$. Taking the positive part $f(s) = (\cdot)_+$ and using $(a+b)_+ \le a_+ + b_+$, we get the recursion:

$$f(s) \le \sum_{s' \in \mathcal{S}_{h-1}} \mathbb{P}[s|s', \pi^\dagger(s')] f(s') + \sum_{s' \in \mathcal{S}_{h-1}} \sum_{a' \ne \pi^\dagger(s')} \mathbb{P}[s|s',a'] q^\pi(s',a'). \tag{35}$$

This recursion implies that the occupancy difference propagates from layer $h-1$ or is generated by suboptimal actions (relative to $\pi^\dagger$).

Summing over all states $s \in \mathcal{S}_h$ at layer $h$:

$$\sum_{s \in \mathcal{S}_h} f(s) \le \sum_{s' \in \mathcal{S}_{h-1}} f(s') \underbrace{\sum_{s \in \mathcal{S}_h} \mathbb{P}[s|s', \pi^\dagger(s')]}_{=1} + \sum_{s' \in \mathcal{S}_{h-1}} \sum_{a' \ne \pi^\dagger(s')} q^\pi(s',a') \underbrace{\sum_{s \in \mathcal{S}_h} \mathbb{P}[s|s',a']}_{=1}$$

$$= \sum_{s' \in \mathcal{S}_{h-1}} f(s') + \sum_{s' \in \mathcal{S}_{h-1}} \sum_{a' \ne \pi^\dagger(s')} q^\pi(s',a').$$

Unrolling this recurrence from $h = 1$ to $H$ (with $f(s_1) = 0$), we obtain:

$$\sum_{h=1}^{H} \sum_{s \in \mathcal{S}_h} f(s) \le \sum_{h=1}^{H} \sum_{k=0}^{h-1} \left( \sum_{s' \in \mathcal{S}_k} \sum_{a' \ne \pi^\dagger(s')} q^\pi(s',a') \right) \le H \sum_{(s,a): a \ne \pi^\dagger(s)} q^\pi(s,a).$$

$\square$

**Lemma B.9.** *The penalty term $\text{pena}_t$ can be bounded as following:*

$$\text{pena}_t \le \frac{2 + 2H^{1/\alpha} S^{1-1/\alpha}}{\beta} \cdot \sigma t^{1/\alpha - 1} \sum_{(s,a) \ne (s, \mathring{\pi}(s))} x_t(s,a)^{1/\alpha}.$$

*Proof.* **Base case** ($t = 1$). By convention we set $\Psi_0 \equiv 0$. Then

$$\text{pena}_1 = \Psi_1(\boldsymbol{\rho}^{\mathbb{P}, \mathring{\pi}}) - \Psi_1(\boldsymbol{x}_1) = \frac{\sigma}{\beta} \Big( \sum_{(s,a)} \boldsymbol{x}_1(s,a)^{1/\alpha} - \sum_{(s,a)} \rho^{\mathbb{P}, \mathring{\pi}}(s,a)^{1/\alpha} \Big),$$

which is bounded by the same expression in the per-step formula evaluated at $t = 1$ (the time factor $(t-1)^{1/\alpha - 1}$ is replaced by 1, equivalently the value at $t = 1$). The induction for $t \ge 2$ proceeds below.

We follow a recursive analysis similar to Jin et al. (2021) (Lemma 20). Recall the penalty term from the FTRL decomposition:

$$\text{pena}_t = \left( \Psi_t(\boldsymbol{\rho}^{\mathbb{P}, \mathring{\pi}}) - \Psi_{t-1}(\boldsymbol{\rho}^{\mathbb{P}, \mathring{\pi}}) \right) - \left( \Psi_t(\boldsymbol{x}_t) - \Psi_{t-1}(\boldsymbol{x}_t) \right). \tag{36}$$

Recall the Tsallis-type regularizer $\Psi_t(\boldsymbol{x}) = -\eta_t^{-1} \sum_{(s,a)} x(s,a)^{1/\alpha}$ with $\eta_t = \beta/(\sigma t^{1/\alpha})$, i.e., $\eta_t^{-1} = \sigma t^{1/\alpha}/\beta$. The increment in the inverse learning rate is:

$$\eta_t^{-1} - \eta_{t-1}^{-1} = \frac{\sigma}{\beta}\left(t^{1/\alpha} - (t-1)^{1/\alpha}\right) \leq \frac{1}{\beta} \cdot \frac{2\sigma}{\alpha} t^{1/\alpha-1}, \tag{37}$$

where the inequality uses the mean value theorem: $t^{1/\alpha} - (t-1)^{1/\alpha} \leq \frac{1}{\alpha}(t-1)^{1/\alpha-1} \leq 2 \cdot \frac{1}{\alpha}t^{1/\alpha-1}$ for $t \geq 2$.

For any occupancy measure $\boldsymbol{q}$, we have:

$$\Psi_t(\boldsymbol{q}) - \Psi_{t-1}(\boldsymbol{q}) = -(\eta_t^{-1} - \eta_{t-1}^{-1}) \sum_{(s,a)} q(s,a)^{1/\alpha}.$$

Thus, substituting into Eq. (36):

$$\texttt{pena}_t = (\eta_t^{-1} - \eta_{t-1}^{-1}) \left( \sum_{(s,a)} x_t(s,a)^{1/\alpha} - \sum_{(s,a)} \rho^{\mathbb{P},\mathring{\pi}}(s,a)^{1/\alpha} \right). \tag{38}$$

Since $\rho^{\mathbb{P},\mathring{\pi}}(s,a) = 0$ for $a \neq \mathring{\pi}(s)$, we have

$$\sum_{(s,a)} x_t(s,a)^{1/\alpha} - \sum_{(s,a)} \rho^{\mathbb{P},\mathring{\pi}}(s,a)^{1/\alpha} = \sum_{(s,a) \neq (s,\mathring{\pi}(s))} x_t(s,a)^{1/\alpha} + \sum_s \left( x_t(s,\mathring{\pi}(s))^{1/\alpha} - \rho^{\mathbb{P},\mathring{\pi}}(s)^{1/\alpha} \right). \tag{39}$$

The first sum is exactly the suboptimal term we want. The key challenge is to bound the second sum by suboptimal terms. We use the concavity of $u \mapsto u^{1/\alpha}$ and the layer structure of the MDP.

For each state $s$ at layer $h$, define $x_t(s) := \sum_{a \in \mathcal{A}} x_t(s,a)$ as the state marginal under $x_t$. We have the recursive relation:

$$x_t(s) = \sum_{s' \in \mathcal{S}_{h-1}} \sum_{a' \in \mathcal{A}} \mathbb{P}[s|s',a'] x_t(s',a').$$

The second term in the decomposition is $\sum_s (x_t(s,\mathring{\pi}(s))^{1/\alpha} - \rho^{\mathbb{P},\mathring{\pi}}(s)^{1/\alpha})$. Consider the set of states where $x_t(s,\mathring{\pi}(s)) \geq \rho^{\mathbb{P},\mathring{\pi}}(s)$ (otherwise the term is negative). Using the concavity of $u \mapsto u^{1/\alpha}$, we have:

$$x_t(s,\mathring{\pi}(s))^{1/\alpha} - \rho^{\mathbb{P},\mathring{\pi}}(s)^{1/\alpha} \leq (x_t(s,\mathring{\pi}(s)) - \rho^{\mathbb{P},\mathring{\pi}}(s))_+^{1/\alpha}$$

Summing over all states and applying Hölder's inequality with $p = \alpha, q = \frac{\alpha}{\alpha-1}$:

$$\sum_s (x_t(s,\mathring{\pi}(s))^{1/\alpha} - \rho^{\mathbb{P},\mathring{\pi}}(s)^{1/\alpha})_+ \leq \sum_s (x_t(s) - \rho^{\mathbb{P},\mathring{\pi}}(s))_+^{1/\alpha}$$

$$\leq \left( \sum_s (x_t(s) - \rho^{\mathbb{P},\mathring{\pi}}(s))_+ \right)^{1/\alpha} \left( \sum_s 1 \right)^{1-1/\alpha}.$$

Using Lemma 4.5 with $q^\pi = x_t$ and $q^{\pi^\dagger} = \rho^{\mathbb{P},\mathring{\pi}}$, we have

$$\text{RHS} \leq \left( H \sum_{(s,a) \neq (s,\mathring{\pi}(s))} x_t(s,a) \right)^{1/\alpha} S^{1-1/\alpha}$$

$$= H^{1/\alpha} S^{1-1/\alpha} \left( \sum_{(s,a) \neq (s,\mathring{\pi}(s))} x_t(s,a) \right)^{1/\alpha}$$

$$\leq H^{1/\alpha} S^{1-1/\alpha} \sum_{(s,a) \neq (s,\mathring{\pi}(s))} x_t(s,a)^{1/\alpha},$$

where the last step uses the inequality $(\sum z_i)^{1/\alpha} \le \sum z_i^{1/\alpha}$ for $\alpha \ge 1$.

Substituting this back into Eq. (39):

$$\sum_{(s,a)} x_t(s,a)^{1/\alpha} - \sum_{(s,a)} \rho^{\mathbb{P},\mathring{\pi}}(s,a)^{1/\alpha} \le \sum_{(s,a)\neq(s,\mathring{\pi}(s))} x_t(s,a)^{1/\alpha} + H^{1/\alpha}S^{1-1/\alpha} \cdot \sum_{(s,a)\neq(s,\mathring{\pi}(s))} x_t(s,a)^{1/\alpha}$$

$$\le \left(1 + H^{1/\alpha}S^{1-1/\alpha}\right) \sum_{(s,a)\neq(s,\mathring{\pi}(s))} x_t(s,a)^{1/\alpha}.$$

Finally, multiplying by the learning rate increment $\eta_t^{-1} - \eta_{t-1}^{-1} \le \frac{1}{\beta} \cdot \frac{2\sigma}{\alpha} t^{1/\alpha-1}$ gives the result:

$$\mathtt{pena}_t \le \frac{1}{\beta} \cdot \frac{2 + 2H^{1/\alpha}S^{1-1/\alpha}}{\alpha} \cdot \sigma t^{1/\alpha-1} \sum_{(s,a)\neq(s,\mathring{\pi}(s))} x_t(s,a)^{1/\alpha}$$

$$\le \frac{2 + 2H^{1/\alpha}S^{1-1/\alpha}}{\beta} \cdot \sigma t^{1/\alpha-1} \sum_{(s,a)\neq(s,\mathring{\pi}(s))} x_t(s,a)^{1/\alpha}.$$

$\square$

### B.5. Bound the Skipping Term

**Lemma B.10.**

$$\mathbb{E}\left[\mathtt{skip}_t \mid \mathcal{F}_{t-1}\right] \le 2C^{1-\alpha}(1 + H^{1/\alpha}S^{1-1/\alpha}) \cdot \sigma t^{1/\alpha-1} \sum_{(s,a)\neq(s,\mathring{\pi}(s))} x_t(s,a)^{1/\alpha}.$$

*Proof.* We follow a recursive layer-by-layer analysis similar to Lemma B.9. The skipping error is $\ell_t(s,a) - \ell_t^{\mathrm{skip}}(s,a) = \ell_t(s,a)\mathbb{1}[|\ell_t(s,a)| > \tau_t(s,a)]$. We denote this term as $\Delta_t^{\mathrm{skip}}(s,a)$. For any $(s,a)$, we have

$$\begin{aligned}
\mathbb{E}[|\Delta_t^{\mathrm{skip}}(s,a)| \mid \mathcal{F}_{t-1}] &:= \mathbb{E}[|\ell_t(s,a)| \cdot \mathbb{1}[|\ell_t(s,a)| > \tau_t(s,a)] \mid \mathcal{F}_{t-1}] \\
&\le \mathbb{E}[|\ell_t(s,a)|^\alpha \cdot \tau_t(s,a)^{1-\alpha} \mid \mathcal{F}_{t-1}] \\
&\le \sigma^\alpha \cdot C^{1-\alpha}\sigma^{1-\alpha}t^{1/\alpha-1}x_t(s,a)^{1/\alpha-1} \\
&\le C^{1-\alpha} \cdot \sigma \cdot t^{1/\alpha-1}x_t(s,a)^{1/\alpha-1}
\end{aligned} \tag{40}$$

We decompose the skipping error into contributions from suboptimal actions and the optimal action. Since $\rho^{\mathbb{P},\mathring{\pi}}(s,a) \neq 0 \iff a = \mathring{\pi}(s)$, we have

$$\mathbb{E}\left[\sum_{(s,a)}(x_t(s,a) - \rho^{\mathbb{P},\mathring{\pi}}(s,a)) \cdot \left(\Delta_t^{\mathrm{skip}}(s,a) + b_t^{\mathrm{skip}}(s,a)\right) \mid \mathcal{F}_{t-1}\right]$$

$$\le \mathbb{E}\left[\sum_{(s,a)}|x_t(s,a) - \rho^{\mathbb{P},\mathring{\pi}}(s,a)| \cdot |\Delta_t^{\mathrm{skip}}(s,a)| \mid \mathcal{F}_{t-1}\right] + \mathbb{E}\left[\sum_{(s,a)}(x_t(s,a) - \rho^{\mathbb{P},\mathring{\pi}}(s,a)) \cdot b_t^{\mathrm{skip}}(s,a) \mid \mathcal{F}_{t-1}\right]$$

$$= \sum_{(s,a)\neq(s,\mathring{\pi}(s))} x_t(s,a) \cdot \mathbb{E}[|\Delta_t^{\mathrm{skip}}(s,a)| \mid \mathcal{F}_{t-1}] + \sum_s |x_t(s,\mathring{\pi}(s)) - \rho^{\mathbb{P},\mathring{\pi}}(s)| \cdot \mathbb{E}[|\Delta_t^{\mathrm{skip}}(s,\mathring{\pi}(s))| \mid \mathcal{F}_{t-1}]$$

$$+ \sum_{(s,a)\neq(s,\mathring{\pi}(s))} x_t(s,a) \cdot b_t^{\mathrm{skip}}(s,a) + \sum_s (x_t(s,\mathring{\pi}(s)) - \rho^{\mathbb{P},\mathring{\pi}}(s)) \cdot b_t^{\mathrm{skip}}(s,\mathring{\pi}(s)).$$

Notice that by definition, we have

$$b_t^{\mathrm{skip}}(s,a) = C^{1-\alpha} \cdot \sigma t^{1/\alpha-1}x_t(s,a)^{1/\alpha-1},$$

we further have

$$
\mathbb{E}\left[\sum_{(s,a)}(x_t(s,a) - \rho^{\mathbb{P},\mathring{\pi}}(s,a)) \cdot \left(\Delta_t^{\mathrm{skip}}(s,a) + b_t^{\mathrm{skip}}(s,a)\right) \mid \mathcal{F}_{t-1}\right]
$$

$$
\leq 2\sum_{(s,a)\neq(s,\mathring{\pi}(s))} x_t(s,a) \cdot (C^{1-\alpha} \cdot \sigma t^{1/\alpha-1} x_t(s,a)^{1/\alpha-1})
$$

$$
+ \sum_s |x_t(s,\mathring{\pi}(s)) - \rho^{\mathbb{P},\mathring{\pi}}(s)| \cdot (C^{1-\alpha} \cdot \sigma t^{1/\alpha-1} x_t(s,\mathring{\pi}(s))^{1/\alpha-1})
$$

$$
+ \sum_s (x_t(s,\mathring{\pi}(s)) - \rho^{\mathbb{P},\mathring{\pi}}(s)) \cdot (C^{1-\alpha} \cdot \sigma t^{1/\alpha-1} x_t(s,\mathring{\pi}(s))^{1/\alpha-1})
$$

$$
= 2C^{1-\alpha} \cdot \sigma t^{1/\alpha-1} \sum_{(s,a)\neq(s,\mathring{\pi}(s))} x_t(s,a)^{1/\alpha}
$$

$$
+ 2C^{1-\alpha} \cdot \sigma t^{1/\alpha-1} \sum_s \left(x_t(s,\mathring{\pi}(s)) - \rho^{\mathbb{P},\mathring{\pi}}(s)\right)_+ \cdot (x_t(s,\mathring{\pi}(s))^{1/\alpha-1})
$$

For the optimal action $\mathring{a} = \mathring{\pi}(s)$, we only need to bound the terms where $x_t(s,\mathring{a}) > \rho^{\mathbb{P},\mathring{\pi}}(s)$ (as negative terms vanished). In this regime, $x_t(s,\mathring{a}) > \rho^{\mathbb{P},\mathring{\pi}}(s)$. Thus,

$$
\sum_s (x_t(s,\mathring{a}) - \rho^{\mathbb{P},\mathring{\pi}}(s))_+ \cdot x_t(s,\mathring{a})^{1/\alpha-1} \leq \sum_s (x_t(s) - \rho^{\mathbb{P},\mathring{\pi}}(s))_+^{1/\alpha} \cdot (x_t(s,\mathring{a}) - \rho^{\mathbb{P},\mathring{\pi}}(s))_+^{1-1/\alpha} x_t(s,\mathring{a})^{1/\alpha-1}
$$

$$
\leq \sum_s (x_t(s) - \rho^{\mathbb{P},\mathring{\pi}}(s))_+^{1/\alpha} \cdot \left(\frac{(x_t(s,\mathring{a}) - \rho^{\mathbb{P},\mathring{\pi}}(s))_+}{x_t(s,\mathring{a})}\right)^{1-1/\alpha}
$$

$$
\leq \sum_s (x_t(s) - \rho^{\mathbb{P},\mathring{\pi}}(s))_+^{1/\alpha}.
$$

Using the same argument as in Lemma B.9 (the Hölder inequality with Lemma 4.5 ), we have

$$
\sum_s (x_t(s) - \rho^{\mathbb{P},\mathring{\pi}}(s))_+^{1/\alpha} \leq H^{1/\alpha} S^{1-1/\alpha} \sum_{(s,a)\neq(s,\mathring{\pi}(s))} x_t(s,a)^{1/\alpha}.
$$

Summing up the bounds, we final get the result:

$$
\mathbb{E}\left[\sum_{(s,a)}(x_t(s,a) - \rho^{\mathbb{P},\mathring{\pi}}(s,a)) \cdot (\ell_t(s,a) - \ell_t^{\mathrm{skip}}(s,a) + b_t^{\mathrm{skip}}(s,a))\right]
$$

$$
\leq 2C^{1-\alpha}(1 + H^{1/\alpha} S^{1-1/\alpha})\sigma t^{1/\alpha-1} \sum_{(s,a)\neq(s,\mathring{\pi}(s))} x_t(s,a)^{1/\alpha}.
$$

$\square$

## B.6. Proof of Theorem B.1

We now combine the bounds established in the previous sections to derive the final regret guarantee for Algorithm 1.

*Proof of Theorem B.1.* From the regret decomposition and the bounds on stability (Lemmas B.5 to B.7), penalty (Lemma B.9), and skipping error (Lemma B.10), we have:

$$
\mathrm{Reg}_T(\mathring{\pi}) \leq \sum_{t=1}^{T} \mathbb{E}[\mathtt{stab}_t + \mathtt{pena}_t + \mathtt{skip}_t]
$$

$$
\leq \frac{8\alpha^2}{\alpha - 1} SA^{1-1/\alpha}\beta C^{2-2\alpha} \cdot \sigma(1 + \log(T)) + M \cdot \mathbb{E}\left[\sum_{t=1}^{T}\sigma t^{1/\alpha-1} \sum_{(s,a)\neq(s,\mathring{\pi}(s))} x_t(s,a)^{1/\alpha}\right], \quad (41)
$$

where we summarize the term $M$ as

$$M = \frac{32\alpha^2}{\alpha - 1}H^2\beta C^{2-\alpha} + \frac{2 + 2H^{1/\alpha}S^{1-1/\alpha}}{\beta} + 2C^{1-\alpha}(1 + H^{1/\alpha}S^{1-1/\alpha}). \tag{42}$$

**Adversarial Bound.** Using Hölder's inequality, we bound the summation over state-action pairs:

$$\sum_{(s,a)\neq(s,\mathring{\pi}(s))} x_t(s,a)^{1/\alpha} \leq \left(\sum_{(s,a)} x_t(s,a)\right)^{1/\alpha}(SA)^{1-1/\alpha} \leq H^{1/\alpha}(SA)^{1-1/\alpha}.$$

Substituting this into Eq. (41) and summing over $t$:

$$\sum_{t=1}^{T}\sigma t^{1/\alpha-1}\sum_{(s,a)\neq(s,\mathring{\pi}(s))} x_t(s,a)^{1/\alpha} \leq \sigma H^{1/\alpha}(SA)^{1-1/\alpha}\sum_{t=1}^{T}t^{1/\alpha-1}$$

where we used $\sum_{t=1}^{T}t^{1/\alpha-1} \leq \int_0^T \tau^{1/\alpha-1}d\tau = \alpha T^{1/\alpha}$. Then, we have the adversarial regret bound:

$$\mathrm{Reg}_T(\mathring{\pi}) \leq \frac{8\alpha^2}{\alpha-1}\beta C^{2-2\alpha}SA^{1-1/\alpha}\cdot\sigma(1+\log(T)) + 2\alpha H^{1/\alpha}S^{1-1/\alpha}A^{1-1/\alpha}M\cdot\sigma T^{1/\alpha}.$$

By the definition of $C$ and $\beta$ in Eqs. (17) and (18) (see also Eq. (13)), we have

$$M = \mathcal{O}\big(H^{2-\frac{1}{\alpha}}\cdot S^{-\frac{1}{\alpha}}\cdot A^{-\frac{2\alpha-1}{\alpha^2}}\big) + \mathcal{O}\big(H\cdot S^{2-\frac{2}{\alpha}}\cdot A^{2-\frac{3}{\alpha}+\frac{1}{\alpha^2}}\big).$$

Both contributions enter the adversarial bound additively below. Therefore, we get:

$$\mathrm{Reg}_T(\mathring{\pi}) \leq \mathcal{O}\bigg(H^{1-\frac{1}{\alpha}}S^{2-\frac{1}{\alpha}}A^{3-\frac{4}{\alpha}+\frac{1}{\alpha^2}}\cdot\sigma\cdot\log(T) + H^{1+\frac{1}{\alpha}}S^{3-\frac{3}{\alpha}}A^{3-\frac{4}{\alpha}+\frac{1}{\alpha^2}}\cdot\sigma\cdot T^{1/\alpha}$$

$$+ H^2\cdot S^{1-\frac{2}{\alpha}}\cdot A^{1-\frac{3}{\alpha}+\frac{1}{\alpha^2}}\cdot\sigma\cdot T^{1/\alpha}\bigg)$$

**Self-Bounding Bound.** We define

$$y_t := \sum_{(s,a)\in\mathcal{S}\times\mathcal{A}}\Delta(s,a)x_t(s,a),$$

and

$$\omega_\Delta(\alpha) := \sum_{(s,a):a\neq\mathring{\pi}(s)}\Delta(s,a)^{-\frac{1}{\alpha-1}}.$$

By Definition 3.1, in expectation we have

$$\mathbb{E}\left[\sum_{t=1}^{T}y_t\right] \leq \mathrm{Reg}_T(\mathring{\pi}) + C_{\mathrm{sb}}.$$

By Hölder's inequality with conjugate exponents $\alpha$ and $\frac{\alpha}{\alpha-1}$:

$$\sum_{(s,a):a\neq\mathring{\pi}(s)} x_t(s,a)^{1/\alpha} = \sum_{(s,a):a\neq\mathring{\pi}(s)}(x_t(s,a)\Delta(s,a))^{1/\alpha}\Delta(s,a)^{-1/\alpha}$$

$$\leq \left(\sum_{(s,a)} x_t(s,a)\Delta(s,a)\right)^{1/\alpha}\left(\sum_{(s,a):a\neq\mathring{\pi}(s)}\Delta(s,a)^{-\frac{1}{\alpha-1}}\right)^{\frac{\alpha-1}{\alpha}}$$

$$= y_t^{1/\alpha}\omega_\Delta(\alpha)^{\frac{\alpha-1}{\alpha}}.$$

Plugging this pointwise bound into the regret bound in Eq. (41) (taking the outer expectation and using Fubini to interchange $\mathbb{E}$ and $\sum_t$):

$$\mathrm{Reg}_T(\mathring{\pi}) \le \frac{8\alpha^2}{\alpha-1}\beta C^{2-2\alpha}SA^{1-1/\alpha}\cdot\sigma(1+\log(T)) + M\cdot\omega_\Delta(\alpha)^{\frac{\alpha-1}{\alpha}}\sum_{t=1}^{T}\sigma t^{1/\alpha-1}\mathbb{E}[y_t^{1/\alpha}].$$

We apply Young's inequality

$$xy \le \frac{x^\alpha}{\alpha} + \frac{y^{\frac{\alpha}{\alpha-1}}}{\frac{\alpha}{\alpha-1}},$$

pointwise with a parameter $\lambda > 0$:

$$\begin{aligned}
M\sigma t^{1/\alpha-1}y_t^{1/\alpha}\omega_\Delta(\alpha)^{\frac{\alpha-1}{\alpha}} &= \left(\lambda^{1/\alpha}y_t^{1/\alpha}\right)\cdot\left(M\lambda^{-1/\alpha}\sigma t^{1/\alpha-1}\omega_\Delta(\alpha)^{\frac{\alpha-1}{\alpha}}\right)\\
&\le \frac{\lambda}{\alpha}y_t + \frac{\alpha-1}{\alpha}\left(M\lambda^{-1/\alpha}\sigma t^{1/\alpha-1}\omega_\Delta(\alpha)^{\frac{\alpha-1}{\alpha}}\right)^{\frac{\alpha}{\alpha-1}}\\
&= \frac{\lambda}{\alpha}y_t + \frac{\alpha-1}{\alpha}M^{\frac{\alpha}{\alpha-1}}\lambda^{-\frac{1}{\alpha-1}}\sigma^{\frac{\alpha}{\alpha-1}}t^{-1}\omega_\Delta(\alpha).
\end{aligned}$$

Taking expectation, summing over $t$, and using the self-bounding constraint $\mathbb{E}[\sum_t y_t] \le \mathrm{Reg}_T(\mathring{\pi}) + C_{\mathrm{sb}}$ (note: the $M^{\alpha/(\alpha-1)}\lambda^{-1/(\alpha-1)}\sigma^{\alpha/(\alpha-1)}t^{-1}\omega_\Delta(\alpha)$ terms are deterministic, so $\mathbb{E}[\cdot]$ acts trivially on them; we also use $\sum_{t=1}^{T}t^{-1} \le 1+\log T$):

$$\mathrm{Reg}_T(\mathring{\pi}) \le \frac{\lambda}{\alpha}(\mathrm{Reg}_T(\mathring{\pi}) + C_{\mathrm{sb}}) + \frac{8\alpha^2}{\alpha-1}\beta C^{2-2\alpha}SA^{1-\frac{1}{\alpha}}\cdot\sigma(1+\log(T)) + \frac{\alpha-1}{\alpha}M^{\frac{\alpha}{\alpha-1}}\lambda^{-\frac{1}{\alpha-1}}\sigma^{\frac{\alpha}{\alpha-1}}\omega_\Delta(\alpha)(1+\log(T)).$$

Let $\gamma = \frac{\lambda}{\alpha} > 0$. The inequality becomes

$$\begin{aligned}
\mathrm{Reg}_T(\mathring{\pi}) &\le \gamma\mathrm{Reg}_T(\mathring{\pi}) + \gamma C_{\mathrm{sb}} + \frac{8\alpha^2}{\alpha-1}\beta C^{2-2\alpha}SA^{1-\frac{1}{\alpha}}\sigma(1+\log(T)) + \gamma^{-\frac{1}{\alpha-1}}\alpha^{-\frac{1}{\alpha-1}}\frac{\alpha-1}{\alpha}M^{\frac{\alpha}{\alpha-1}}\sigma^{\frac{\alpha}{\alpha-1}}\omega_\Delta(\alpha)(1+\log(T))\\
&\le \gamma\mathrm{Reg}_T(\mathring{\pi}) + \gamma C_{\mathrm{sb}} + \kappa + \gamma^{-\frac{1}{\alpha-1}}\cdot M^{\frac{\alpha}{\alpha-1}}\sigma^{\frac{\alpha}{\alpha-1}}\omega_\Delta(\alpha)(1+\log(T))\\
&= \gamma\mathrm{Reg}_T(\mathring{\pi}) + \gamma C_{\mathrm{sb}} + \kappa + \gamma^{-\frac{1}{\alpha-1}}\Lambda,
\end{aligned}\tag{43}$$

where $\kappa = \frac{8\alpha^2}{\alpha-1}\beta C^{2-2\alpha}SA^{1-\frac{1}{\alpha}}\sigma(1+\log(T))$ and $\Lambda = M^{\frac{\alpha}{\alpha-1}}\sigma^{\frac{\alpha}{\alpha-1}}\omega_\Delta(\alpha)(1+\log(T))$.

Set $\gamma \in (0,1)$. We multiply both sides by $1/(1-\gamma)$ in both sides and rearrange the terms,

$$\mathrm{Reg}_T(\mathring{\pi}) \le \frac{\gamma}{1-\gamma}C_{\mathrm{sb}} + \frac{1}{1-\gamma}\kappa + \frac{\gamma^{-\frac{1}{\alpha-1}}}{1-\gamma}\Lambda \le \frac{\gamma}{1-\gamma}(C_{\mathrm{sb}} + \kappa) + \kappa + \frac{\gamma^{-\frac{1}{\alpha-1}}}{1-\gamma}\Lambda$$

Now we consider two cases. If $C_{\mathrm{sb}} = 0$, setting $\gamma = 1/2$ leads to

$$\mathrm{Reg}_T(\mathring{\pi}) \le 2\kappa + 2^{\frac{1}{\alpha-1}+1}\Lambda.\tag{44}$$

If $C_{\mathrm{sb}} > 0$, restricting $\gamma \in (0, 1/2]$ so that $1/(1-\gamma) \le 2$, we have from Eq. (43):

$$\mathrm{Reg}_T(\mathring{\pi}) \le 2\gamma C_{\mathrm{sb}} + 2\kappa + 2\gamma^{-\frac{1}{\alpha-1}}\Lambda.$$

We choose $\gamma$ to balance $\gamma C_{\mathrm{sb}}$ and $\gamma^{-\frac{1}{\alpha-1}}\Lambda$, capped at $1/2$:

$$\gamma = \min\left\{\frac{1}{2}, \left(\frac{\Lambda}{C_{\mathrm{sb}}}\right)^{\frac{\alpha-1}{\alpha}}\right\}.$$

**Sub-case 2a (non-saturating):** $(\Lambda/C_{\mathrm{sb}})^{(\alpha-1)/\alpha} \le 1/2$. With $\gamma = (\Lambda/C_{\mathrm{sb}})^{(\alpha-1)/\alpha}$, we have $\gamma C_{\mathrm{sb}} = C_{\mathrm{sb}}^{1/\alpha}\Lambda^{(\alpha-1)/\alpha} = \gamma^{-1/(\alpha-1)}\Lambda$, hence

$$\mathrm{Reg}_T(\mathring{\pi}) \le \mathcal{O}\left(C_{\mathrm{sb}}^{1/\alpha}\Lambda^{\frac{\alpha-1}{\alpha}} + \kappa\right).\tag{45}$$

**Sub-case 2b (saturated):** $(\Lambda/C_{\mathrm{sb}})^{(\alpha-1)/\alpha} > 1/2$. With $\gamma = 1/2$, the bound from Eq. (43) becomes

$$\mathrm{Reg}_T(\mathring{\pi}) \le 2\kappa + 2^{1/(\alpha-1)+1}\Lambda + C_{\mathrm{sb}},$$

which (combining with Eq. (45)) is dominated by Eq. (44) plus the mixed term. Combining Eq. (44) and Eq. (45), and substituting $\Lambda$, we obtain the final bound:

$$\begin{aligned}
\mathrm{Reg}_T(\mathring{\pi}) &\le \mathcal{O}\left(2\kappa + 2^{\frac{1}{\alpha-1}+1}\Lambda + C_{\mathrm{sb}}^{1/\alpha}\Lambda^{\frac{\alpha-1}{\alpha}}\right) \\
&= \mathcal{O}\Big(\frac{16\alpha^2}{\alpha-1}\beta C^{2-2\alpha}SA^{1-\frac{1}{\alpha}}\cdot\sigma(1+\log(T)) + 2^{\frac{1}{\alpha-1}+1}M^{\frac{\alpha}{\alpha-1}}\cdot\sigma^{\frac{\alpha}{\alpha-1}}\omega_\Delta(\alpha)(1+\log(T)) \\
&\qquad + C_{\mathrm{sb}}^{1/\alpha}\cdot M\cdot\sigma\omega_\Delta(\alpha)^{\frac{\alpha-1}{\alpha}}\cdot(1+\log(T))^{\frac{\alpha-1}{\alpha}}\Big)
\end{aligned}$$

Recall from Eq. (13) that $M = \mathcal{O}\big(H^{2-1/\alpha}S^{-1/\alpha}A^{-(2\alpha-1)/\alpha^2}\big) + \mathcal{O}\big(HS^{2-2/\alpha}A^{2-3/\alpha+1/\alpha^2}\big)$. Both contributions enter the $M^{\alpha/(\alpha-1)}$ Young term (via $(a+b)^p \le 2^{p-1}(a^p+b^p)$ for $p = \alpha/(\alpha-1) \ge 1$) and the $C_{\mathrm{sb}}^{1/\alpha}M$ mixed term additively. We obtain the regret bound for self-bounding regime:

$$\begin{aligned}
\mathrm{Reg}_T(\mathring{\pi}) \le \mathcal{O}\Bigg( & H^{1-\frac{1}{\alpha}}S^{2-\frac{1}{\alpha}}A^{3-\frac{4}{\alpha}+\frac{1}{\alpha^2}}\cdot\sigma(1+\log(T)) \\
& + H^{\frac{\alpha}{\alpha-1}}S^2A^{2-\frac{1}{\alpha}}\cdot\sigma^{\frac{\alpha}{\alpha-1}}\omega_\Delta(\alpha)(1+\log(T)) \\
& + H\cdot S^{2-\frac{2}{\alpha}}A^{2-\frac{3}{\alpha}+\frac{1}{\alpha^2}}\cdot C_{\mathrm{sb}}^{\frac{1}{\alpha}}\cdot\sigma\omega_\Delta(\alpha)^{1-\frac{1}{\alpha}}\left(1+\log(T)\right)^{1-\frac{1}{\alpha}} \\
& + H^{\frac{2\alpha-1}{\alpha-1}}\cdot S^{-\frac{1}{\alpha-1}}\cdot A^{-\frac{2\alpha-1}{\alpha(\alpha-1)}}\cdot\sigma^{\frac{\alpha}{\alpha-1}}\omega_\Delta(\alpha)(1+\log(T)) \\
& + H^{2-\frac{1}{\alpha}}\cdot S^{-\frac{1}{\alpha}}\cdot A^{-\frac{2\alpha-1}{\alpha^2}}\cdot C_{\mathrm{sb}}^{\frac{1}{\alpha}}\cdot\sigma\omega_\Delta(\alpha)^{1-\frac{1}{\alpha}}\left(1+\log(T)\right)^{1-\frac{1}{\alpha}}\Bigg)
\end{aligned}$$

$\square$

## C. Proof of Regret Bounds for Unknown Transition

In this section, we provide the detailed proof of the best-of-both-worlds guarantees for HT-FTRL-UOB (Algorithm 2). The formal version of Theorem 5.2 is provided below:

**Theorem C.1** (BoBW Guarantee for Unknown Transition Case). *By setting constant $\delta = 1/T^3$, $C$ in Eq. (17), $\beta$ in Eq. (18), and $D = H\sigma$, HT-FTRL-UOB (Algorithm 2) achieves BoBW regret bounds in the adversarial and self-bounding regimes simultaneously under Assumption 5.1:*

*1. Adversarial Regime:* *For any adversarial loss distributions and deterministic benchmark policy $\mathring{\pi}$,*

$$\mathrm{Reg}_T(\mathring{\pi}) \leq \widetilde{\mathcal{O}}\left(H^{1+\frac{1}{\alpha}}S^{4-\frac{3}{\alpha}}A^{4-\frac{4}{\alpha}+\frac{1}{\alpha^2}} \cdot \sigma T^{1/\alpha} + H^2 S^{2-\frac{2}{\alpha}}A^{2-\frac{3}{\alpha}+\frac{1}{\alpha^2}} \cdot \sigma T^{1/\alpha} + H\sigma S\sqrt{AT}\right)$$

*2. Self-Bounding Regime:* *If the environment satisfies the $(\mathring{\pi}, \Delta, C_{\mathrm{sb}})$ self-bounding constraint (Definition 3.1) and $\Delta_{\min} = \min_{(s,a):a\neq\mathring{\pi}(s)} \Delta(s,a)$ is the minimal suboptimal gap, then we have*

$$\begin{aligned}
\mathrm{Reg}_T(\mathring{\pi}) \leq \mathcal{O}\Big(& H^4\sigma^2 S^3 A^2 \log^2(\iota) + H^{1-\frac{1}{\alpha}}S^{3-\frac{1}{\alpha}}A^{4-\frac{4}{\alpha}+\frac{1}{\alpha^2}}\sigma \log^2(T) \\
& + H^6\sigma^2 S^2 \omega_\Delta(2)\log(\iota) + H^6\sigma^2 S^2 \Delta_{\min}^{-1}\log(\iota) + H^{\frac{2\alpha-1}{\alpha-1}}S^3 A^{3-\frac{1}{\alpha}} \cdot \sigma^{\frac{\alpha}{\alpha-1}}\omega_\Delta(\alpha)\log(T)^2 \\
& + H^3\sigma S\sqrt{\omega_\Delta(2)C_{\mathrm{sb}}\log(\iota)} + H^3\sigma S\sqrt{\Delta_{\min}^{-1}C_{\mathrm{sb}}\log(\iota)} \\
& + C_{\mathrm{sb}}^{1/\alpha} \cdot H^{2-\frac{1}{\alpha}} \cdot S^{3-\frac{3}{\alpha}}A^{3-\frac{4}{\alpha}+\frac{1}{\alpha^2}} \cdot \sigma \cdot \omega_\Delta(\alpha)^{\frac{\alpha-1}{\alpha}}\log^{\frac{2(\alpha-1)}{\alpha}}(T)\Big).
\end{aligned}$$

*where $\omega_\Delta(x) := \sum_{(s,a):a\neq\mathring{\pi}(s)} \Delta(s,a)^{-\frac{1}{x-1}}$, and $\iota = HSAT/\delta$.*

We detailed the proof for adversarial regime in Appendix C.6, and self-bounding regime in Appendix C.7. We then provide the formal proof of this theorem in the following sections.

### C.1. Regret Decomposition

We decompose the regret $\mathrm{Reg}_T(\mathring{\pi})$ against a reference policy $\mathring{\pi}$ into four parts: the transition error due to the unknown transition model (TRANSERR), the estimation error due to the FTRL algorithm on the estimated MDPs (ESTREG), the pessimism error (PESSERR), and the skipping error (SKIPERR). Recall the definition of $\ell_t^{\mathrm{pess}}(s,a) = \ell_t(s,a) - D \cdot B_i(s,a)$. The total regret for given reference policy $\mathring{\pi}$ can be decomposed as:

$$\mathrm{Reg}_T(\mathring{\pi}) = \mathbb{E}\left[\sum_{t=1}^T V^{\mathbb{P},\pi_t}(s_1;\boldsymbol{\ell}_t) - V^{\mathbb{P},\mathring{\pi}}(s_1;\boldsymbol{\ell}_t)\right]$$

$$= \mathbb{E}\left[\sum_{t=1}^T V^{\mathbb{P},\pi_t}(s_1;\boldsymbol{\ell}_t) - V^{\widehat{\mathbb{P}}_{i(t)},\pi_t}(s_1;\boldsymbol{\ell}_t^{\mathrm{pess}}) + V^{\widehat{\mathbb{P}}_{i(t)},\pi_t}(s_1;\boldsymbol{\ell}_t^{\mathrm{pess}}) - V^{\widehat{\mathbb{P}}_{i(t)},\mathring{\pi}}(s_1;\boldsymbol{\ell}_t^{\mathrm{pess}}) + V^{\widehat{\mathbb{P}}_{i(t)},\mathring{\pi}}(s_1;\boldsymbol{\ell}_t^{\mathrm{pess}}) - V^{\mathbb{P},\mathring{\pi}}(s_1;\boldsymbol{\ell}_t)\right]$$

$$= \mathbb{E}\left[\sum_{t=1}^T V^{\mathbb{P},\pi_t}(s_1;\boldsymbol{\ell}_t) - V^{\widehat{\mathbb{P}}_{i(t)},\pi_t}(s_1;\boldsymbol{\ell}_t^{\mathrm{pess}})\right] + \mathbb{E}\left[\sum_{t=1}^T V^{\widehat{\mathbb{P}}_{i(t)},\pi_t}(s_1;\widehat{\boldsymbol{\ell}}_t) - V^{\widehat{\mathbb{P}}_{i(t)},\mathring{\pi}}(s_1;\widehat{\boldsymbol{\ell}}_t)\right]$$

$$+ \mathbb{E}\left[\sum_{t=1}^T V^{\widehat{\mathbb{P}}_{i(t)},\pi_t}(s_1;\boldsymbol{\ell}_t^{\mathrm{pess}} - \widehat{\boldsymbol{\ell}}_t) - V^{\widehat{\mathbb{P}}_{i(t)},\mathring{\pi}}(s_1;\boldsymbol{\ell}_t^{\mathrm{pess}} - \widehat{\boldsymbol{\ell}}_t)\right] + \mathbb{E}\left[\sum_{t=1}^T V^{\widehat{\mathbb{P}}_{i(t)},\mathring{\pi}}(s_1;\boldsymbol{\ell}_t^{\mathrm{pess}}) - V^{\mathbb{P},\mathring{\pi}}(s_1;\boldsymbol{\ell}_t)\right]$$

$$= \underbrace{\mathbb{E}\left[\sum_{t=1}^T V^{\mathbb{P},\pi_t}(s_1;\boldsymbol{\ell}_t) - V^{\widehat{\mathbb{P}}_{i(t)},\pi_t}(s_1;\boldsymbol{\ell}_t^{\mathrm{pess}})\right]}_{\text{TRANSERR}} + \underbrace{\mathbb{E}\left[\sum_{t=1}^T V^{\widehat{\mathbb{P}}_{i(t)},\pi_t}(s_1;\widehat{\boldsymbol{\ell}}_t) - V^{\widehat{\mathbb{P}}_{i(t)},\mathring{\pi}}(s_1;\widehat{\boldsymbol{\ell}}_t)\right]}_{\text{ESTREG}}$$

$$+ \underbrace{\mathbb{E}\left[\sum_{t=1}^T V^{\widehat{\mathbb{P}}_{i(t)},\pi_t}(s_1;\boldsymbol{\ell}_t^{\mathrm{pess}} - \widehat{\boldsymbol{\ell}}_t) - V^{\widehat{\mathbb{P}}_{i(t)},\mathring{\pi}}(s_1;\boldsymbol{\ell}_t^{\mathrm{pess}} - \widehat{\boldsymbol{\ell}}_t)\right]}_{\text{SKIPERR}} + \underbrace{\mathbb{E}\left[\sum_{t=1}^T V^{\widehat{\mathbb{P}}_{i(t)},\mathring{\pi}}(s_1;\boldsymbol{\ell}_t^{\mathrm{pess}}) - V^{\mathbb{P},\mathring{\pi}}(s_1;\boldsymbol{\ell}_t)\right]}_{\text{PESSERR}}$$

## C.2. High-Probability Confidence Set

We define $\mathcal{E}_i$ as the event that the confidence set $\widehat{\mathcal{P}}_i$ contains the true transition probability $\mathbb{P}$, and $\mathcal{E} = \bigcap_{i=1}^{N} \mathcal{E}_i$.

Notice that we estimate the transition probability $\widehat{\mathbb{P}}_i$ following the popular doubling epoch schedule framework and Bernstein-type confidence set construction (Jin et al., 2020), we further can estimate the high-probability upper bound of the transition probability $\widehat{\mathbb{P}}_i$ by Jin et al. (2020, Lemma 2).

**Lemma C.2** (Jin et al. (2020, Lemma 2)). *With probability at least $1 - 4\delta$, we have event $\mathcal{E}$ holds:* $\Pr[\mathcal{E}] \geq 1 - 4\delta$.

## C.3. Bounding the Pessimistic Error (PESSERR)

Notice that $\mathbb{E}[|\ell_t(s,a)|^\alpha] \leq \sigma^\alpha$ for each $t, s, a$. Since $\alpha > 1$, by Jensen's inequality $\mathbb{E}[|\ell_t(s,a)| \mid \mathcal{F}_{t-1}] \leq (\mathbb{E}[|\ell_t(s,a)|^\alpha \mid \mathcal{F}_{t-1}])^{1/\alpha} \leq \sigma$. This will directly give the following two-sided control on the value function.

**Lemma C.3.** *Under Assumption 5.1, for any policy $\pi$, step $h \in [H]$, and state $s \in \mathcal{S}_h$,*

$$0 \leq \mathbb{E}[V^{\mathbb{P},\pi}(s; \boldsymbol{\ell}_t) \mid \mathcal{F}_{t-1}] \leq (H - h + 1)\sigma \leq H\sigma = D.$$

*Proof.* For each $(s', a)$, the upper bound on the moment yields $\mathbb{E}[\ell_t(s', a) \mid \mathcal{F}_{t-1}] \leq \mathbb{E}[|\ell_t(s', a)| \mid \mathcal{F}_{t-1}] \leq \sigma$. Conversely, Assumption 5.1 gives $\mathbb{E}_{\ell \sim \nu_t(s', a)}[\ell \cdot \mathbb{1}[|\ell| \leq M]] \geq 0$ for every $M > 0$; since $\mathbb{E}[|\ell_t(s', a)| \mid \mathcal{F}_{t-1}] \leq \sigma < \infty$, the dominated convergence theorem (with $|\ell| \in L^1$ as the dominating function) yields $\mathbb{E}[\ell_t(s', a) \mid \mathcal{F}_{t-1}] = \lim_{M \to \infty} \mathbb{E}[\ell_t(s', a) \cdot \mathbb{1}[|\ell_t(s', a)| \leq M] \mid \mathcal{F}_{t-1}] \geq 0$. Hence $\mathbb{E}[\ell_t(s', a) \mid \mathcal{F}_{t-1}] \in [0, \sigma]$ for all $(s', a)$.

Iterating the Bellman equation,

$$\mathbb{E}[V^{\mathbb{P},\pi}(s; \boldsymbol{\ell}_t) \mid \mathcal{F}_{t-1}] = \sum_{h'=h}^{H} \sum_{(s',a)} \Pr_{\mathbb{P},\pi}[(s_{h'}, a_{h'}) = (s', a) \mid s_h = s] \cdot \mathbb{E}[\ell_t(s', a) \mid \mathcal{F}_{t-1}],$$

where the marginal probabilities $\Pr_{\mathbb{P},\pi}[(s_{h'}, a_{h'}) = (s', a) \mid s_h = s] \geq 0$ sum to 1 over $(s', a)$ for each fixed $h' \geq h$. Combining with $\mathbb{E}[\ell_t(s', a) \mid \mathcal{F}_{t-1}] \in [0, \sigma]$, the right-hand side lies in $[0, (H - h + 1)\sigma] \subseteq [0, H\sigma]$. $\square$

Then, we are ready to establish the pessimistic value function.

**Lemma C.4** (Pessimistic Value Function). *Let $\mathcal{E}_{i(t)} := \{|\widehat{\mathbb{P}}_{i(t)}[s'|s,a] - \mathbb{P}[s'|s,a]| \leq B_{i(t)}(s,a,s')$ for all $(s,a,s')\}$ denote the per-epoch confidence event at time $t$. Since $\widehat{\mathbb{P}}_{i(t)}$ and $B_{i(t)}(\cdot, \cdot, \cdot)$ are constructed from data observed before epoch $i(t)$ begins, $\mathcal{E}_{i(t)} \in \mathcal{F}_{t-1}$ (i.e., $\mathbb{1}[\mathcal{E}_{i(t)}]$ is $\mathcal{F}_{t-1}$-measurable). On the event $\mathcal{E}_{i(t)}$, we have for any policy $\pi$ and any state $s$,*

$$\mathbb{E}[V^{\widehat{\mathbb{P}}_{i(t)},\pi}(s; \boldsymbol{\ell}_t^{pess}) \mid \mathcal{F}_{t-1}] \leq \mathbb{E}[V^{\mathbb{P},\pi}(s; \boldsymbol{\ell}_t) \mid \mathcal{F}_{t-1}]. \tag{46}$$

*Proof.* To prove the lemma, we need to show that for each $h$, and any state $s_h \in \mathcal{S}_h$, the pessimism for Q function holds:

$$\mathbb{E}[Q^{\widehat{\mathbb{P}}_{i(t)},\pi}(s_h, a; \boldsymbol{\ell}_t^{\text{pess}}) \mid \mathcal{F}_{t-1}] \leq \mathbb{E}[Q^{\mathbb{P},\pi}(s_h, a; \boldsymbol{\ell}_t) \mid \mathcal{F}_{t-1}]. \tag{47}$$

We prove this by induction on the horizon $h$.

- Base case: $h = H + 1$, we have $Q^{\widehat{\mathbb{P}}_{i(t)},\pi}(s_{H+1}, a; \boldsymbol{\ell}_t^{\text{pess}}) = 0 \leq 0 = Q^{\mathbb{P},\pi}(s_{H+1}, a; \boldsymbol{\ell}_t)$ holds for any $a \in \mathcal{A}$.

- Induction step: Suppose the lemma holds for $h + 1$, we have:

$$\mathbb{E}[Q^{\widehat{\mathbb{P}}_{i(t)},\pi}(s_h, a; \boldsymbol{\ell}_t^{\text{pess}}) \mid \mathcal{F}_{t-1}]$$

$$= \mathbb{E}[\ell_t(s_h, a) - D \cdot B_i(s_h, a) \mid \mathcal{F}_{t-1}] + \sum_{s' \in \mathcal{S}_{h+1}} \widehat{\mathbb{P}}_{i(t)}[s'|s_h, a] \mathbb{E}[V^{\widehat{\mathbb{P}}_{i(t)},\pi}(s'; \boldsymbol{\ell}_t^{\text{pess}}) \mid \mathcal{F}_{t-1}]$$

$$\leq \mathbb{E}[\ell_t(s_h, a) \mid \mathcal{F}_{t-1}] + \sum_{s' \in \mathcal{S}_{h+1}} \mathbb{P}[s'|s_h, a] \mathbb{E}[V^{\mathbb{P},\pi}(s'; \boldsymbol{\ell}_t) \mid \mathcal{F}_{t-1}]$$

$$- D \cdot B_i(s_h, a) + \sum_{s' \in \mathcal{S}_{h+1}} \left( \widehat{\mathbb{P}}_{i(t)}[s'|s_h, a] - \mathbb{P}[s'|s_h, a] \right) \mathbb{E}[V^{\mathbb{P},\pi}(s'; \boldsymbol{\ell}_t) \mid \mathcal{F}_{t-1}]$$

$$\leq \mathbb{E}[\ell_t(s_h, a) \mid \mathcal{F}_{t-1}] + \sum_{s' \in \mathcal{S}_{h+1}} \mathbb{P}[s'|s_h, a] \mathbb{E}[V^{\mathbb{P},\pi}(s'; \boldsymbol{\ell}_t) \mid \mathcal{F}_{t-1}]$$

$$= \mathbb{E}[Q^{\mathbb{P},\pi}(s_h, a; \boldsymbol{\ell}_t) \mid \mathcal{F}_{t-1}],$$

where the first inequality holds by induction hypothesis, and the second inequality follows from the sign argument below. By Lemma C.3, $\mathbb{E}[V^{\mathbb{P},\pi}(s'; \boldsymbol{\ell}_t) \mid \mathcal{F}_{t-1}] \in [0, D]$ for every $s' \in \mathcal{S}_{h+1}$. Splitting the sum into positive and negative parts of $(\widehat{\mathbb{P}}_{i(t)} - \mathbb{P})[s'|s_h, a]$ and dropping the non-positive contribution from indices with $\widehat{\mathbb{P}}_{i(t)} < \mathbb{P}$:

$$\sum_{s' \in \mathcal{S}_{h+1}} \left( \widehat{\mathbb{P}}_{i(t)}[s'|s_h, a] - \mathbb{P}[s'|s_h, a] \right) \mathbb{E}[V^{\mathbb{P},\pi}(s'; \boldsymbol{\ell}_t) \mid \mathcal{F}_{t-1}] \leq \sum_{s' \in \mathcal{S}_{h+1}} \left( \widehat{\mathbb{P}}_{i(t)}[s'|s_h, a] - \mathbb{P}[s'|s_h, a] \right)_+ \cdot D$$

$$\leq D \cdot \min \left\{ 1, \sum_{s' \in \mathcal{S}_{h+1}} B_i(s_h, a, s') \right\} = D \cdot B_i(s_h, a),$$

where the second step uses two facts: $(i)$ $\sum_{s'} (\widehat{\mathbb{P}}_{i(t)} - \mathbb{P})_+[s'|s_h, a] \leq \sum_{s'} \widehat{\mathbb{P}}_{i(t)}[s'|s_h, a] = 1$ since the positive part is bounded by $\widehat{\mathbb{P}}_{i(t)}[s'|s_h, a]$ itself, and $(ii)$ on the event $\mathcal{E}_{i(t)}$, $(\widehat{\mathbb{P}}_{i(t)} - \mathbb{P})_+[s'|s_h, a] \leq |\widehat{\mathbb{P}}_{i(t)} - \mathbb{P}|[s'|s_h, a] \leq B_i(s_h, a, s')$ for each $s'$, giving $\sum_{s'} (\widehat{\mathbb{P}}_{i(t)} - \mathbb{P})_+[s'|s_h, a] \leq \sum_{s'} B_i(s_h, a, s')$. Combining yields the inequality.

$\square$

Combined with the pessimism of estimated transition and loss function, we get:

**Lemma C.5.** *The pessimism error can be bounded by*

$$\mathbb{E}[\text{PessErr}] \leq O(H^2 \sigma T \delta)$$

*Proof.* Let $X_t := V^{\widehat{\mathbb{P}}_{i(t)},\hat{\pi}}(s_1; \boldsymbol{\ell}_t^{\text{pess}}) - V^{\mathbb{P},\hat{\pi}}(s_1; \boldsymbol{\ell}_t)$, so $\mathbb{E}[\text{PessErr}] = \sum_{t=1}^T \mathbb{E}[X_t]$. For each $t$, we split by the *per-epoch* confidence event $\mathcal{E}_{i(t)}$ defined in Lemma C.4, which is $\mathcal{F}_{t-1}$-measurable. Since $\mathcal{E} = \bigcap_i \mathcal{E}_i \subseteq \mathcal{E}_{i(t)}$, we have $\mathcal{E}_{i(t)}^c \subseteq \mathcal{E}^c$ and hence $\Pr[\mathcal{E}_{i(t)}^c] \leq \Pr[\mathcal{E}^c] \leq 4\delta$ (Jin et al. (2020), Lemma 2)). Therefore

$$\mathbb{E}[X_t] = \mathbb{E}[X_t \mathbb{1}[\mathcal{E}_{i(t)}]] + \mathbb{E}[X_t \mathbb{1}[\mathcal{E}_{i(t)}^c]].$$

**The $\mathcal{E}_{i(t)}$ part.** Since $\mathbb{1}[\mathcal{E}_{i(t)}] \in \mathcal{F}_{t-1}$, the tower property and Lemma C.4 give

$$\mathbb{E}[X_t \mathbb{1}[\mathcal{E}_{i(t)}]] = \mathbb{E}\left[ \mathbb{1}[\mathcal{E}_{i(t)}] \cdot \mathbb{E}[X_t \mid \mathcal{F}_{t-1}] \right] \leq \mathbb{E}[\mathbb{1}[\mathcal{E}_{i(t)}] \cdot 0] = 0.$$

**The $\mathcal{E}_{i(t)}^c$ part.** Under Assumption 5.1, $\mathbb{E}[\ell_t(s, a) \mid \mathcal{F}_{t-1}] \in [0, \sigma]$ (cf. Lemma C.3), and $DB_{i(t)}(s, a) \in [0, D]$ since $B_{i(t)} \in [0, 1]$. Hence $\mathbb{E}[\ell_t^{\text{pess}}(s, a) \mid \mathcal{F}_{t-1}] = \mathbb{E}[\ell_t \mid \mathcal{F}_{t-1}] - DB_{i(t)}(s, a) \in [-D, \sigma]$. Iterating the Bellman equation over the $\leq H$ remaining steps with $\widehat{\mathbb{P}}_{i(t)}$:

$$\left| \mathbb{E}[V^{\widehat{\mathbb{P}}_{i(t)},\hat{\pi}}(s_1; \boldsymbol{\ell}_t^{\text{pess}}) \mid \mathcal{F}_{t-1}] \right| \leq H \cdot \max\{\sigma, D\} = HD = H^2 \sigma,$$

using $D = H\sigma \geq \sigma$ for $H \geq 1$ and $\sigma \geq 0$. Combined with $|\mathbb{E}[V^{\mathbb{P},\hat{\pi}}(s_1; \ell_t) \mid \mathcal{F}_{t-1}]| \leq H^2\sigma$ from Lemma C.3, we obtain $|\mathbb{E}[X_t \mid \mathcal{F}_{t-1}]| \leq 2H^2\sigma$. Since $\mathbb{1}[\mathcal{E}^c_{i(t)}] \in \mathcal{F}_{t-1}$:

$$\left| \mathbb{E}[X_t \mathbb{1}[\mathcal{E}^c_{i(t)}]] \right| = \left| \mathbb{E}\left[ \mathbb{1}[\mathcal{E}^c_{i(t)}] \cdot \mathbb{E}[X_t \mid \mathcal{F}_{t-1}] \right] \right| \leq 2H^2\sigma \cdot \Pr[\mathcal{E}^c_{i(t)}] \leq 8H^2\sigma\delta.$$

**Combining both parts.** Summing over $t$:

$$\mathbb{E}[\text{PessErr}] = \sum_{t=1}^{T} \mathbb{E}[X_t] \leq \sum_{t=1}^{T} 0 + \sum_{t=1}^{T} 8H^2\sigma\delta = 8H^2\sigma T\delta = \mathcal{O}(H^2\sigma T\delta).$$

$\square$

## C.4. Bounding FTRL Regret (EstReg)

Since the algorithm resets FTRL in each epoch, EstReg is the sum of regrets over epochs. We define $\text{EstReg}_i$ as the regret over epoch $i$, which defined as

$$\text{EstReg}_i = \mathbb{E}\left[ \sum_{t=t_i}^{t_{i+1}-1} \left\langle \rho^{\widehat{\mathbb{P}}_i, \pi_t} - \rho^{\widehat{\mathbb{P}}_i, \hat{\pi}}, \widehat{\ell}_t \right\rangle \right]. \tag{48}$$

A key observation is that the learned occupancy measure $x_t$ in Algorithm 2 is exactly $\rho^{\widehat{\mathbb{P}}_i, \pi_t}$. Therefore, we apply similar techniques presented in Appendix B to bound the $\text{EstReg}_i$.

We first consider the bound of $\widehat{\ell}_t(s, a)$:

**Lemma C.6.** *Under good event $\mathcal{E}_{i(t)}$, we have*

$$|\widehat{\ell}_t(s, a)| \leq \left( C + C^{1-\alpha} \right) \cdot \sigma x_t(s, a)^{1/\alpha - 1} (t - t_{i(t)} + 1)^{1/\alpha} + H\sigma$$

*Proof.* By definition of $\widehat{\ell}_t(s, a)$, and definition of $\tau_t(s, a)$ and $b_t^{\text{skip}}(s, a)$ in Eqs. (7) and (8), we have

$$
\begin{aligned}
|\widehat{\ell}_t(s, a)| &= \left| \left( \frac{\ell_t^{\text{skip}}(s, a)}{u_t(s, a)} \right) \cdot \mathbb{1}_t[(s, a)] - b_t^{\text{skip}}(s, a) - D \cdot B_{i(t)}(s, a) \right| \\
&\leq \left| \frac{\ell_t^{\text{skip}}(s, a)}{u_t(s, a)} \right| + b_t^{\text{skip}}(s, a) + D \cdot B_{i(t)}(s, a) \\
&\leq \left| \frac{\tau_t(s, a)}{x_t(s, a)} \right| + b_t^{\text{skip}}(s, a) + D \\
&\leq \sigma x_t(s, a)^{1/\alpha - 1} \cdot \left( C(t - t_{i(t)} + 1)^{1/\alpha} + C^{1-\alpha}(t - t_{i(t)} + 1)^{1/\alpha - 1} \right) + H\sigma \\
&\leq \left( C + C^{1-\alpha} \right) \cdot \sigma x_t(s, a)^{1/\alpha - 1} (t - t_{i(t)} + 1)^{1/\alpha} + H\sigma,
\end{aligned}
$$

where the second inequality holds since $u_t(s, a) \geq x_t(s, a)$ (by Lemma C.10), and the last step uses $(t - t_{i(t)} + 1)^{1/\alpha - 1} \leq (t - t_{i(t)} + 1)^{1/\alpha}$ for $t \geq t_{i(t)}$. $\square$

**Lemma C.7** (Failure-event control). *Let*

$$\widehat{\ell}_t^{\text{IS}}(s, a) := \frac{\ell_t^{\text{skip}}(s, a)}{u_t(s, a)} \mathbb{1}_t[(s, a)]$$

*be the importance-weighted part of $\widehat{\ell}_t(s, a)$. For any epoch-$i$ confidence event $\mathcal{E}_i$ with $\mathbb{1}_{\mathcal{E}_i} \in \mathcal{F}_{t-1}$ and $\Pr[\mathcal{E}_i^c] \leq 4\delta$, and any nonnegative $\mathcal{F}_{t-1}$-measurable weights $w_t(s, a)$ satisfying $\sum_{s,a} w_t(s, a) \leq K$, we have*

$$\mathbb{E}\left[ \mathbb{1}_{\mathcal{E}_i^c} \sum_{s,a} w_t(s, a) \left| \widehat{\ell}_t^{\text{IS}}(s, a) \right| \right] \leq \mathcal{O}\left( KCS\sigma t (t - t_i + 1)^{1/\alpha} \delta \right). \tag{49}$$

*Moreover,*

$$\mathbb{E}\left[\mathbb{1}_{\mathcal{E}_i^c}\sum_{s,a} w_t(s,a)|\ell_t(s,a)|\right] \leq \mathcal{O}(K\sigma\delta), \tag{50}$$

*and hence the same bound holds with* $|\ell_t^{skip}(s,a)|$ *in place of* $|\ell_t(s,a)|$*. Finally,*

$$\mathbb{E}\left[\mathbb{1}_{\mathcal{E}_i^c}\sum_{s,a} w_t(s,a)DB_i(s,a)\right] \leq \mathcal{O}(KH\sigma\delta). \tag{51}$$

*Proof.* Following Jin et al. (2021, Lemma C.1.2), we have $u_t(s) \geq 1/(St)$. Since the policy is fixed when maximizing over transition kernels, the state-action UOB used by the algorithm satisfies

$$u_t(s,a) = \max_{\widehat{\mathbb{P}}\in\widehat{\mathcal{P}}_{i(t)}} \rho^{\widehat{\mathbb{P}},\pi_t}(s,a) = u_t(s)\pi_t(a|s).$$

Let $q_t(s,a) := \rho^{\mathbb{P},\pi_t}(s,a) = q_t(s)\pi_t(a|s)$. Using $q_t(s) \leq 1$ and the state-level lower bound,

$$\frac{\mathbb{1}_t[(s,a)]}{u_t(s,a)} = \frac{\mathbb{1}_t[(s,a)]}{u_t(s)\pi_t(a|s)} \leq St \cdot \frac{\mathbb{1}_t[(s,a)]}{q_t(s)\pi_t(a|s)} = St \cdot \frac{\mathbb{1}_t[(s,a)]}{q_t(s,a)},$$

where if $\pi_t(a|s) = 0$ then all ratios above are interpreted as zero. We also use the convention that the ratio is zero when $q_t(s,a) = 0$. Since $|\ell_t^{skip}(s,a)| \leq \tau_t(s,a) \leq C\sigma(t - t_i + 1)^{1/\alpha}$, we have

$$|\widehat{\ell}_t^{IS}(s,a)| \leq CS\sigma t\,(t - t_i + 1)^{1/\alpha} \frac{\mathbb{1}_t[(s,a)]}{q_t(s,a)}.$$

The event $\mathcal{E}_i^c$ is determined at the beginning of epoch $i$, so $\mathbb{1}_{\mathcal{E}_i^c}$ and $w_t$ are $\mathcal{F}_{t-1}$-measurable. Moreover, $\mathbb{E}[\mathbb{1}_t[(s,a)]/q_t(s,a) \mid \mathcal{F}_{t-1}] \leq 1$ under the same zero-ratio convention, with equality whenever $q_t(s,a) > 0$. Therefore,

$$\mathbb{E}\left[\mathbb{1}_{\mathcal{E}_i^c}\sum_{s,a} w_t(s,a)\frac{\mathbb{1}_t[(s,a)]}{q_t(s,a)} \,\bigg|\, \mathcal{F}_{t-1}\right] \leq \mathbb{1}_{\mathcal{E}_i^c}\sum_{s,a} w_t(s,a).$$

Taking expectation and using $\sum_{s,a} w_t(s,a) \leq K$ and $\Pr[\mathcal{E}_i^c] \leq 4\delta$ gives Eq. (49). For Eq. (50), Jensen's inequality and the $\alpha$-moment assumption give $\mathbb{E}[|\ell_t(s,a)| \mid \mathcal{F}_{t-1}] \leq \sigma$, so

$$\mathbb{E}\left[\mathbb{1}_{\mathcal{E}_i^c}\sum_{s,a} w_t(s,a)|\ell_t(s,a)| \,\bigg|\, \mathcal{F}_{t-1}\right] \leq \mathbb{1}_{\mathcal{E}_i^c}K\sigma.$$

The claim for $|\ell_t^{skip}|$ follows from $|\ell_t^{skip}| \leq |\ell_t|$. Finally, Eq. (51) follows from $B_i(s,a) \leq 1$, $D = H\sigma$, and $\Pr[\mathcal{E}_i^c] \leq 4\delta$. $\quad\square$

**Lemma C.8.** *Define the per-step dominant coefficient*

$$M := \underbrace{\mathcal{O}\big(H^{2-\frac{1}{\alpha}} \cdot S^{-\frac{1}{\alpha}} \cdot A^{-\frac{2\alpha-1}{\alpha^2}}\big)}_{\text{stability contribution}} + \underbrace{\mathcal{O}\big(H \cdot S^{2-\frac{2}{\alpha}} \cdot A^{2-\frac{3}{\alpha}+\frac{1}{\alpha^2}}\big)}_{\text{penalty contribution}}. \tag{52}$$

*We can control*

$$\begin{aligned}
\text{ESTREG}_i \leq O\Bigg( & M \cdot \sum_{t=t_i}^{t_{i+1}-1} \sigma(t - t_{i(t)} + 1)^{1/\alpha-1} \cdot \sum_{(s,a)\neq(s,\hat{\pi}(s))} x_t(s,a)^{1/\alpha}\Bigg) \\
& + O\Big(H^{1-\frac{1}{\alpha}}S^{2-\frac{1}{\alpha}}A^{3-\frac{4}{\alpha}+\frac{1}{\alpha^2}}\sigma\log(t_{i+1} - t_i)\Big) \\
& + O\big(HAS\sigma + (H\sigma SAT^2 + H^2\sigma)(t_{i+1} - t_i)\delta\big).
\end{aligned} \tag{53}$$

*Proof.* First we consider the case when the epoch confidence event $\mathcal{E}_i$ does not happen. We use Lemma C.7 with $w_t(s,a) = \rho^{\widehat{\mathbb{P}}_i, \pi_t}(s,a) + \rho^{\widehat{\mathbb{P}}_i, \mathring{\pi}}(s,a)$. Since $\sum_{s,a} w_t(s,a) \leq 2H$, the IS part and the transition-bonus part on $\mathcal{E}_i^c$ contribute at most

$$
\begin{aligned}
&\mathbb{E}\left[\sum_{t=t_i}^{t_{i+1}-1} \mathbb{1}_{\mathcal{E}_i^c} \sum_{s,a} w_t(s,a)\left(\left|\widehat{\ell}_t^{\mathrm{IS}}(s,a)\right| + DB_i(s,a)\right)\right] \\
&\leq \mathcal{O}\left(HCS\sigma T^{1+1/\alpha}(t_{i+1}-t_i)\delta + H^2\sigma(t_{i+1}-t_i)\delta\right) \\
&\leq \mathcal{O}\left((H\sigma SAT^2 + H^2\sigma)(t_{i+1}-t_i)\delta\right),
\end{aligned}
\tag{54}
$$

where the last inequality uses $C \leq 1$, $\alpha \in (1,2]$, and $T^{1+1/\alpha} \leq T^2$. The deterministic skipping-bonus part $b_t^{\mathrm{skip}}$ is not controlled through the failure probability; it is bounded by the same Tsallis penalty/suboptimal-mass argument used below and is included in the main term of Eq. (53).

Next we consider the case under $\mathcal{E}_i$. Recall the definition of $\widehat{\ell}_t(s,a)$, which can be seen as three parts: the importance sampling skipping loss $\widetilde{\ell}_t(s,a) = \ell_t^{\mathrm{skip}}(s,a)\mathbb{1}[(s,a)\text{ is visited}]/u_t(s,a)$, the skipping bonus term $b_t^{\mathrm{skip}}(s,a)$, and the transition bonus term $DB_i(s,a)$. We first apply standard FTRL regret decomposition, and get:

$$
\begin{aligned}
&\sum_{t=t_i}^{t_{i+1}-1}\left\langle \boldsymbol{x}_t - \boldsymbol{\rho}^{\widehat{\mathbb{P}}_i, \mathring{\pi}}, \widehat{\boldsymbol{\ell}}_t\right\rangle \\
&\leq \underbrace{\sum_{t=t_i}^{t_{i+1}-1}\left\langle \boldsymbol{x}_t - \boldsymbol{x}_{t+1}, \widehat{\boldsymbol{\ell}}_t\right\rangle - D_{\Psi_t}(\boldsymbol{x}_{t+1}, \boldsymbol{x}_t)}_{\text{Stability Term}} + \underbrace{\sum_{t=t_i}^{t_{i+1}-1}\left(\Psi_t(\boldsymbol{\rho}^{\widehat{\mathbb{P}}_i, \mathring{\pi}}) - \Psi_{t-1}(\boldsymbol{\rho}^{\widehat{\mathbb{P}}_i, \mathring{\pi}})\right) - (\Psi_t(\boldsymbol{x}_t) - \Psi_{t-1}(\boldsymbol{x}_t))}_{\text{Penalty Term}}
\end{aligned}
\tag{55}
$$

We further bound the stability term and penalty term separately.

**Bounding the Stability Term:** We apply similar methods in Lemma B.5. First we consider the shifted loss $\ell_t^{\mathrm{shift}}(s,a)$ defined as:

$$
\ell_t^{\mathrm{shift}}(s,a) := Q^{\widehat{\mathbb{P}}_{i(t)}, \pi_t}(s,a; \widetilde{\ell}_t(s,a)) - V^{\widehat{\mathbb{P}}_{i(t)}, \pi_t}(s; \widetilde{\ell}_t(s,a)),
$$

where $\widetilde{\ell}_t(s,a)$ is defined by

$$
\widetilde{\ell}_t(s,a) := \frac{\ell_t^{\mathrm{skip}}(s,a)\cdot\mathbb{1}[(s,a)\text{ is visited}]}{u_t(s,a)}.
$$

Therefore, we have

$$
\begin{aligned}
&\sum_{t=t_i}^{t_{i+1}-1}\left\langle \boldsymbol{x}_t - \boldsymbol{\rho}^{\widehat{\mathbb{P}}_i, \mathring{\pi}}, \widehat{\boldsymbol{\ell}}_t\right\rangle \\
&= \sum_{t=t_i}^{t_{i+1}-1}\sum_{(s,a)\in\mathcal{S}\times\mathcal{A}}(x_t(s,a) - \rho^{\widehat{\mathbb{P}}_i, \mathring{\pi}}(s,a))\cdot\left(\ell_t^{\mathrm{shift}}(s,a) - b_t^{\mathrm{skip}}(s,a) - D\cdot B_i(s,a)\right).
\end{aligned}
$$

Similar analysis in Lemma 4.3 shows that:

$$
\left|\ell_t^{\mathrm{shift}}(s,a)\right| \leq C(1 + HSA(1 + A^{1-1/\alpha}))\cdot\sigma x_t(s,a)^{1/\alpha-1}(t - t_{i(t)} + 1)^{1/\alpha}
$$

where, compared to the known transition case (Lemma B.2), three substitutions apply: $(i)$ the imprtance sampling denominator is $u_t(s,a)$, but under $\mathcal{E}_{i(t)}$ we have $u_t(s,a) \geq x_t(s,a)$, so the same pointwise upper bound $|\widetilde{\ell}_t(s,a)| \leq \tau_t(s,a)/x_t(s,a)$ holds; $(ii)$ the occupancy $x_t = \rho^{\widehat{\mathbb{P}}_{i(t)}, \pi_t}$ is taken under the empirical transition, but the centered $+\Delta V$ decomposition has identical structure with $H, S, A$ as the only state-action quantities; $(iii)$ the reachability weights live in

$\mathcal{Q}(\widehat{\mathbb{P}}_{i(t)})$ rather than $\mathcal{Q}(\mathbb{P})$, but the layer-by-layer bound $H \cdot \sum_{(s,a)} x_t(s,a)$ only uses the MDP layered structure, not the specific transition kernel. Therefore, by definition of $C$ and $\beta$ in Eq. (17) and Eq. (18), we have:

$$\alpha\eta_t \cdot \left| \ell_t^{\text{shift}}(s,a) - b_t^{\text{skip}}(s,a) \right|$$
$$\leq \alpha\beta \cdot \left( C(1 + HSA(1 + A^{1-1/\alpha})) + C^{1-\alpha} \right) \cdot x_t(s,a)^{1/\alpha-1}$$
$$\leq \frac{\alpha-1}{4\alpha} x_t(s,a)^{1/\alpha-1}$$

Then we can apply the methods in Lemma B.5. By the definition of $\widehat{\ell}_t$ and Eq. (55), we get:

$$\text{Stability Term} \leq \frac{8\alpha^2}{\alpha-1} \sum_{t=t_i}^{t_{i+1}-1} \eta_t \sum_{(s,a)\in\mathcal{S}\times\mathcal{A}} x_t(s,a)^{2-1/\alpha} \left( \left( \ell_t^{\text{shift}}(s,a) \right)^2 + (b_t^{\text{skip}}(s,a))^2 \right)$$
$$+ \sum_{(s,a)\in\mathcal{S}\times\mathcal{A}} \left( x_{t_{i+1}}(s,a) - x_{t_i}(s,a) \right) \cdot D \cdot B_i(s,a),$$

where the first part is by applying Lemma B.5 towards $(\boldsymbol{\ell}_t^{\text{shift}} - \boldsymbol{b}_t^{\text{skip}})$, and the second term is the telescoping summation over $D \cdot B_i(s,a)$ for each $(s,a) \in \mathcal{S} \times \mathcal{A}$. Notice that under $\mathcal{E}_{i(t)}$, we have $u_t(s,a) \geq \rho^{\mathbb{P},\pi_t}(s,a)$ and $u_t(s,a) \geq x_t(s,a) = \rho^{\widehat{\mathbb{P}}_{i(t)},\pi_t}(s,a)$ (by Lemma C.10). Combining the conditional independence $\ell_t \perp \mathbb{1}_t[(s,a)] \mid \mathcal{F}_{t-1}$ and the truncated moment bound $\mathbb{E}[(\ell_t^{\text{skip}}(s,a))^2 \mid \mathcal{F}_{t-1}] \leq \sigma^\alpha \tau_t(s,a)^{2-\alpha} = C^{2-\alpha}\sigma^2(t - t_{i(t)} + 1)^{(2-\alpha)/\alpha} x_t(s,a)^{(2-\alpha)/\alpha}$, the pointwise importance sampling variance satisfies

$$\mathbb{E}[\widetilde{\ell}_t(s,a)^2 \mid \mathcal{F}_{t-1}] = \frac{\rho^{\mathbb{P},\pi_t}(s,a)}{u_t(s,a)^2} \cdot \mathbb{E}[(\ell_t^{\text{skip}}(s,a))^2 \mid \mathcal{F}_{t-1}] \leq \frac{1}{x_t(s,a)} \cdot \mathbb{E}[(\ell_t^{\text{skip}}(s,a))^2 \mid \mathcal{F}_{t-1}],$$

where the inequality applies $\rho^{\mathbb{P},\pi_t} \leq u_t$ once in the numerator and $u_t \geq x_t$ once in the remaining $1/u_t$ factor. The right-hand side has the same form as the bound in the known transition case, so the analysis in Lemma B.6 applies verbatim with the substitutions $t \to t - t_{i(t)} + 1$ and $x_t = \rho^{\mathbb{P},\pi_t} \to x_t = \rho^{\widehat{\mathbb{P}}_{i(t)},\pi_t}$. The remaining steps of the bound (including the $(1 - \pi_t(a \mid s))$ factor that removes the optimal-action contribution) use only $x_t(s,a) = x_t(s)\pi_t(a \mid s)$ from $x_t \in \mathcal{Q}(\widehat{\mathbb{P}}_{i(t)})$, not the specific transition kernel. We obtain

$$\mathbb{E}\left[ \eta_t \sum_{(s,a)\in\mathcal{S}\times\mathcal{A}} x_t(s,a)^{2-1/\alpha} \left( \ell_t^{\text{shift}}(s,a) \right)^2 \right] \leq O\left( H^2\beta C^{2-\alpha} \cdot \sigma(t - t_{i(t)} + 1)^{1/\alpha-1} \cdot \sum_{(s,a)\neq(s,\mathring{\pi}(s))} x_t(s,a)^{1/\alpha} \right). \tag{56}$$

Moreover, for the skipping bonus term, by similar argument in Lemma B.7, we have

$$\mathbb{E}\left[ \sum_{t=t_i}^{t_{i+1}-1} \eta_t \sum_{(s,a)\in\mathcal{S}\times\mathcal{A}} x_t(s,a)^{2-1/\alpha} \left( b_t^{\text{skip}}(s,a) \right)^2 \right] \leq O\left( SA^{1-1/\alpha}\beta C^{2-2\alpha} \cdot \sigma\left(1 + \log(t_{i+1} - t_i)\right) \right). \tag{57}$$

Therefore, summing over $t$ in the epoch and incorporating the definitions of $C$ and $\beta$ in Eq. (17) and Eq. (18), we get

$$\mathbb{E}\left[\text{Stability Term}\right] \leq O\left( H^2\beta C^{2-\alpha} \cdot \sum_{t=t_i}^{t_{i+1}-1} \sigma(t - t_{i(t)} + 1)^{1/\alpha-1} \cdot \sum_{(s,a)\neq(s,\mathring{\pi}(s))} x_t(s,a)^{1/\alpha} \right)$$
$$+ O\left( SAH\sigma + SA^{1-1/\alpha}\beta C^{2-2\alpha} \cdot \sigma\left(1 + \log(t_{i+1} - t_i)\right) \right) \tag{58}$$

**Bounding the penalty term** We apply the same analysis as in Lemma B.9. Since the Tsallis penalty contribution scales as $\eta_t^{-1} = \sigma(t - t_{i(t)} + 1)^{1/\alpha}/\beta$ (cf. the convention in Eq. (18) and the proof of Lemma B.9), the $1/\beta$ factor must be retained:

$$\mathbb{E}\left[\text{Penalty Term}\right] \leq O\left( \frac{1 + H^{\frac{1}{\alpha}}S^{1-\frac{1}{\alpha}}}{\beta} \cdot \sum_{t=t_i}^{t_{i+1}-1} \sigma(t - t_{i(t)} + 1)^{1/\alpha-1} \cdot \sum_{(s,a)\neq(s,\mathring{\pi}(s))} x_t(s,a)^{1/\alpha} \right) \tag{59}$$

**Combining stability and penalty via direct $\beta$-substitution**  With both per-step coefficients now in hand (the $H^2\beta C^{2-\alpha}$ in equation 58 and $(1 + H^{1/\alpha}S^{1-1/\alpha})/\beta$ in equation 59), substituting $\beta = \mathcal{O}((HSA^{2-1/\alpha})^{-(1-1/\alpha)})$ from Eq. (18) gives:

- Penalty per-step:

$$\frac{1 + H^{1/\alpha}S^{1-1/\alpha}}{\beta} = \mathcal{O}\big(H \cdot S^{2-2/\alpha} \cdot A^{2-3/\alpha+1/\alpha^2}\big),$$

  where the second term dominates for $S \geq 1$.

- Stability time-dependent per-step:

$$H^2\beta C^{2-\alpha} = \mathcal{O}\big(H^{2-1/\alpha}S^{-1/\alpha}A^{-(2\alpha-1)/\alpha^2}\big).$$

- Stability skipping-bonus per-epoch:

$$SA^{1-1/\alpha}\beta C^{2-2\alpha}\sigma\log(t_{i+1}-t_i) = \mathcal{O}\big(H^{1-1/\alpha}S^{2-1/\alpha}A^{3-4/\alpha+1/\alpha^2}\sigma\log(t_{i+1}-t_i)\big),$$

  using $\beta C^{2-2\alpha} = \mathcal{O}((HSA^{2-1/\alpha})^{1-1/\alpha})$.

Mirroring the known-transition convention in Eq. (12), we take

$$M := \mathcal{O}\big(H^{2-1/\alpha} \cdot S^{-1/\alpha} \cdot A^{-(2\alpha-1)/\alpha^2}\big) + \mathcal{O}\big(H \cdot S^{2-2/\alpha} \cdot A^{2-3/\alpha+1/\alpha^2}\big),$$

as displayed in Eq. (52). The two contributions trade off across regimes: the stability term dominates when $H \gtrsim (SA)^{(2-1/\alpha)/(1-1/\alpha)}$, while the penalty term dominates otherwise. The skipping-bonus per-epoch term becomes a separate per-epoch additive contribution recorded in the second line of Eq. (53). Summing Eqs. (54), (58) and (59) then directly implies the result. $\qquad\square$

### C.5. Bounding Skipping Error (SKIPERR)

We further rewrite the Skipping Error term

$$\text{SKIPERR} = \mathbb{E}\left[\sum_{t=1}^T \left\langle \boldsymbol{\rho}^{\widehat{\mathbb{P}}_{i(t)},\pi_t} - \boldsymbol{\rho}^{\widehat{\mathbb{P}}_{i(t)},\mathring{\pi}}, \boldsymbol{\ell}_t^{\text{pess}} - \widehat{\boldsymbol{\ell}}_t \right\rangle\right]$$

**Lemma C.9.** *Under Assumption 5.1, with $M$ defined in Eq. (52), the unconditional expectation of the skipping error satisfies*

$$\mathbb{E}\left[\text{SKIPERR}\right] \leq \mathbb{E}\left[O\left(M \cdot \sum_{t=1}^T \sigma(t - t_{i(t)} + 1)^{1/\alpha-1} \cdot \sum_{(s,a)\neq(s,\mathring{\pi}(s))} x_t(s,a)^{1/\alpha}\right)\right]$$

$$+ \mathbb{E}\left[\sigma\sum_{t=1}^T \sum_{(s,a)\in\mathcal{S}\times\mathcal{A}} |u_t(s,a) - \rho^{\mathbb{P},\pi_t}(s,a)|\right] + \mathcal{O}\big(H^2\sigma SAT^3\delta\big). \tag{60}$$

*Proof.* Notice that by the definition of $\ell_t^{\text{pess}}$ and $\widehat{\ell}_t$, the $D \cdot B_i(s,a)$ terms cancel and we have

$$\ell_t^{\text{pess}}(s,a) - \widehat{\ell}_t(s,a) = \ell_t(s,a) - \frac{\ell_t^{\text{skip}}(s,a)}{u_t(s,a)} \cdot \mathbb{1}_t[(s,a)] + b_t^{\text{skip}}(s,a).$$

On the per-epoch event $\mathcal{E}_{i(t)} \in \mathcal{F}_{t-1}$, using $x_t(s,a) = \rho^{\widehat{\mathbb{P}}_{i(t)},\pi_t}(s,a)$ and the conditional independence of $\ell_t(s,a)$ and

$\mathbb{1}_t[(s,a)]$ given $\mathcal{F}_{t-1}$, we decompose the conditional expectation as

$$
\mathbb{E}\left[\left\langle \boldsymbol{\rho}^{\widehat{\mathbb{P}}_{i(t)},\pi_t} - \boldsymbol{\rho}^{\widehat{\mathbb{P}}_{i(t)},\mathring{\pi}}, \boldsymbol{\ell}_t^{\text{pess}} - \widehat{\boldsymbol{\ell}}_t \right\rangle \mid \mathcal{F}_{t-1}\right]
$$
$$
= \underbrace{\sum_{(s,a)\in\mathcal{S}\times\mathcal{A}} \left(x_t(s,a) - \rho^{\widehat{\mathbb{P}}_{i(t)},\mathring{\pi}}(s,a)\right) \cdot \mathbb{E}\left[\ell_t(s,a) - \ell_t^{\text{skip}}(s,a) + b_t^{\text{skip}}(s,a) \mid \mathcal{F}_{t-1}\right]}_{=:\mathtt{I}_t}
$$
$$
+ \underbrace{\sum_{(s,a)\in\mathcal{S}\times\mathcal{A}} \left(x_t(s,a) - \rho^{\widehat{\mathbb{P}}_{i(t)},\mathring{\pi}}(s,a)\right) \cdot \mathbb{E}\left[\ell_t^{\text{skip}}(s,a) - \frac{\ell_t^{\text{skip}}(s,a)}{u_t(s,a)} \cdot \mathbb{1}_t[(s,a)] \mid \mathcal{F}_{t-1}\right]}_{=:\mathtt{J}_t},
$$

where we used $\mathbb{E}\left[\ell_t^{\text{skip}}(s,a) \cdot \mathbb{1}_t[(s,a)] \mid \mathcal{F}_{t-1}\right] = \rho^{\mathbb{P},\pi_t}(s,a) \cdot \mathbb{E}\left[\ell_t^{\text{skip}}(s,a) \mid \mathcal{F}_{t-1}\right]$ in the decomposition.

**Bounding $\mathtt{I}_t$:** The first term $\mathtt{I}_t$ has the same structure as the skipping error in the known transition case (see Lemma B.10), with $\widehat{\mathbb{P}}_{i(t)}$ playing the role of $\mathbb{P}$ and epoch-level time $t - t_{i(t)} + 1$ replacing $t$. The truncation moment bound (Eq. (40) in Lemma B.10) gives

$$
\mathbb{E}\left[|\ell_t(s,a) - \ell_t^{\text{skip}}(s,a) + b_t^{\text{skip}}(s,a)| \mid \mathcal{F}_{t-1}\right] \leq 2C^{1-\alpha}\sigma(t - t_{i(t)} + 1)^{1/\alpha-1}x_t(s,a)^{1/\alpha-1},
$$

and the suboptimal mass propagation Lemma 4.5 applies under $\widehat{\mathbb{P}}_{i(t)}$ (since it is purely a combinatorial layered-MDP argument independent of the specific transition kernel), yielding

$$
\sum_s (x_t(s) - \rho^{\widehat{\mathbb{P}}_{i(t)},\mathring{\pi}}(s))_+^{1/\alpha} \leq H^{1/\alpha}S^{1-1/\alpha} \cdot \sum_{(s,a)\neq(s,\mathring{\pi}(s))} x_t(s,a)^{1/\alpha}.
$$

Combining with the truncation moment bound, the per-step constant is $2C^{1-\alpha}(1 + H^{1/\alpha}S^{1-1/\alpha}) = \mathcal{O}(H \cdot S^{2-2/\alpha} \cdot A^{2-3/\alpha+1/\alpha^2}) \leq \mathcal{O}(M)$ (using $C^{1-\alpha} = \mathcal{O}((HSA^{2-1/\alpha})^{1-1/\alpha})$ from Eq. (17), with $M$ defined in Eq. (52)). Therefore

$$
\sum_{t=1}^T \mathbb{E}\left[\mathtt{I}_t \mid \mathcal{F}_{t-1}\right] \leq O\left(M \cdot \sum_{t=1}^T \sigma(t - t_{i(t)} + 1)^{1/\alpha-1} \cdot \sum_{(s,a)\neq(s,\mathring{\pi}(s))} x_t(s,a)^{1/\alpha}\right). \tag{61}
$$

**Bounding $\mathtt{J}_t$:** For the second term, we compute the conditional expectation:

$$
\mathbb{E}\left[\ell_t^{\text{skip}}(s,a) - \frac{\ell_t^{\text{skip}}(s,a)}{u_t(s,a)} \cdot \mathbb{1}_t[(s,a)] \mid \mathcal{F}_{t-1}\right] = \mathbb{E}\left[\ell_t^{\text{skip}}(s,a) \mid \mathcal{F}_{t-1}\right] \cdot \frac{u_t(s,a) - \rho^{\mathbb{P},\pi_t}(s,a)}{u_t(s,a)}.
$$

On the per-epoch event $\mathcal{E}_{i(t)}$ (which guarantees $\mathbb{P} \in \widehat{\mathcal{P}}_{i(t)}$), the UOB property gives $u_t(s,a) \geq \rho^{\mathbb{P},\pi_t}(s,a)$, so $\frac{u_t(s,a)-\rho^{\mathbb{P},\pi_t}(s,a)}{u_t(s,a)} \geq 0$. Moreover, Assumption 5.1 gives $\mathbb{E}\left[\ell_t^{\text{skip}}(s,a) \mid \mathcal{F}_{t-1}\right] = \mathbb{E}_{\ell\sim\nu_t(s,a)}[\ell \cdot \mathbb{1}[|\ell| \leq \tau_t(s,a)]] \geq 0$. Thus the product $\mathbb{E}\left[\ell_t^{\text{skip}}(s,a) \mid \mathcal{F}_{t-1}\right] \cdot \frac{u_t(s,a)-\rho^{\mathbb{P},\pi_t}(s,a)}{u_t(s,a)}$ is nonnegative for every $(s,a)$, which means the contribution of $-\rho^{\widehat{\mathbb{P}}_{i(t)},\mathring{\pi}}(s,a)$ in the $(x_t - \rho^{\widehat{\mathbb{P}}_{i(t)},\mathring{\pi}})$ factor is nonpositive. Dropping this nonpositive contribution yields the upper bound

$$
\mathtt{J}_t \leq \sum_{(s,a)\in\mathcal{S}\times\mathcal{A}} x_t(s,a) \cdot \mathbb{E}\left[\ell_t^{\text{skip}}(s,a) \mid \mathcal{F}_{t-1}\right] \cdot \frac{u_t(s,a) - \rho^{\mathbb{P},\pi_t}(s,a)}{u_t(s,a)}.
$$

Since $u_t(s,a) \geq x_t(s,a)$ by definition of the UOB, we have $\frac{x_t(s,a)}{u_t(s,a)} \leq 1$. Furthermore, $0 \leq \mathbb{E}\left[\ell_t^{\text{skip}}(s,a) \mid \mathcal{F}_{t-1}\right] \leq \mathbb{E}\left[|\ell_t(s,a)| \mid \mathcal{F}_{t-1}\right] \leq \sigma$, where the last step follows from Jensen's inequality and $\mathbb{E}[|\ell_t(s,a)|^\alpha] \leq \sigma^\alpha$. Thus,

$$
\mathtt{J}_t \leq \sigma \sum_{(s,a)\in\mathcal{S}\times\mathcal{A}} |u_t(s,a) - \rho^{\mathbb{P},\pi_t}(s,a)|. \tag{62}
$$

**Combining the bounds:** The bounds Eqs. (61) and (62) on $\mathbb{E}[\mathtt{I}_t + \mathtt{J}_t \mid \mathcal{F}_{t-1}]$ are derived on the per-epoch event $\mathcal{E}_{i(t)}$, and the bounds themselves are non-negative. Since $\mathbb{1}_{\mathcal{E}_{i(t)}} \in \mathcal{F}_{t-1}$, the tower property applies to the per-time indicator splitting:

$$\mathbb{E}\left[\text{SKIPERR}\right] = \mathbb{E}\left[\sum_{t=1}^{T}\mathbb{1}_{\mathcal{E}_{i(t)}}\mathbb{E}\left[\left\langle\boldsymbol{\rho}^{\widehat{\mathbb{P}}_{i(t)},\pi_t} - \boldsymbol{\rho}^{\widehat{\mathbb{P}}_{i(t)},\mathring{\pi}}, \boldsymbol{\ell}_t^{\text{pess}} - \widehat{\boldsymbol{\ell}}_t\right\rangle \mid \mathcal{F}_{t-1}\right]\right]$$

$$+ \mathbb{E}\left[\sum_{t=1}^{T}\mathbb{1}_{\mathcal{E}_{i(t)}^c}\left\langle\boldsymbol{\rho}^{\widehat{\mathbb{P}}_{i(t)},\pi_t} - \boldsymbol{\rho}^{\widehat{\mathbb{P}}_{i(t)},\mathring{\pi}}, \boldsymbol{\ell}_t^{\text{pess}} - \widehat{\boldsymbol{\ell}}_t\right\rangle\right]$$

$$\leq \mathbb{E}\left[\sum_{t=1}^{T}(\mathtt{I}_t + \mathtt{J}_t)\right] + \mathcal{O}\big(H^2\sigma SAT^3\delta\big),$$

where the second line is bounded by Lemma C.7 applied with the weights $w_t(s,a) = \rho^{\widehat{\mathbb{P}}_{i(t)},\pi_t}(s,a) + \rho^{\widehat{\mathbb{P}}_{i(t)},\mathring{\pi}}(s,a)$ (so $\sum_{s,a}w_t \leq 2H$, giving $K = 2H$), summed over $t \in [T]$, analogous to the $\mathcal{E}^c$ bookkeeping in Eq. (54). Summing Eqs. (61) and (62) over $t$ yields the desired result. $\qquad\square$

### C.6. Proof for Adversarial Regime

We now decompose the transition error as

$$\mathbb{E}[\text{TRANSERR}] = \mathbb{E}\left[\sum_{t=1}^{T}\left\langle\boldsymbol{\rho}^{\mathbb{P},\pi_t}, \boldsymbol{\ell}_t\right\rangle - \left\langle\boldsymbol{\rho}^{\widehat{\mathbb{P}}_{i(t)},\pi_t}, \boldsymbol{\ell}_t^{\text{pess}}\right\rangle\right]$$

$$\leq \mathbb{E}\left[\sum_{t=1}^{T}\sum_{(s,a)\in\mathcal{S}\times\mathcal{A}}\big(\rho^{\mathbb{P},\pi_t}(s,a) - x_t(s,a)\big)\cdot\ell_t(s,a)\right]$$

$$+ \mathbb{E}\left[\sum_{t=1}^{T}\sum_{(s,a)\in\mathcal{S}\times\mathcal{A}}x_t(s,a)\cdot D\cdot B_{i(t)}(s,a)\right]$$

By the analysis in Jin et al. (2021, Eqs.(36-38) in Appendix C.2) we get the following

$$\mathbb{E}\left[\sum_{t=1}^{T}\sum_{(s,a)\in\mathcal{S}\times\mathcal{A}}\big(\rho^{\mathbb{P},\pi_t}(s,a) - x_t(s,a)\big)\cdot\ell_t(s,a)\right] \leq \sigma\cdot\mathbb{E}\left[\sum_{t=1}^{T}\sum_{(s,a)\in\mathcal{S}\times\mathcal{A}}\big|\rho^{\mathbb{P},\pi_t}(s,a) - x_t(s,a)\big|\right]$$

$$\leq \mathcal{O}\left(H\sigma S\sqrt{AT\log(\iota)} + H^2\sigma S^3A^3\log^2(\iota) + \sigma SAT\delta\right)$$

$$\mathbb{E}\left[\sum_{t=1}^{T}\sum_{(s,a)\in\mathcal{S}\times\mathcal{A}}x_t(s,a)\cdot D\cdot B_{i(t)}(s,a)\right] \leq \mathcal{O}\left(H\sigma S\sqrt{AT\log(\iota)} + H^2\sigma S^2A\log^2(\iota)\right)$$

Notice that by the doubling epoch argument (Jin et al., 2020), we have the epoch number $N \leq \mathcal{O}(SA\log(T))$. Then, the estimation regret shown in Lemma C.8 implies:

$$\mathbb{E}\left[\text{ESTREG}\right] = \mathbb{E}\left[\sum_{i=1}^{N}\text{ESTREG}_i\right]$$

$$\leq \mathbb{E}\left[\mathcal{O}\left(M\cdot\sum_{t=1}^{T}\sigma(t - t_{i(t)} + 1)^{1/\alpha-1}\cdot\sum_{(s,a)\neq(s,\mathring{\pi}(s))}x_t(s,a)^{1/\alpha}\right)\right]$$

$$+ \mathcal{O}\left(HS^2A^2\sigma\log(T) + H\sigma SAT^3\delta + H^2\sigma T\delta\right),$$

where $M$ is the per-step dominant coefficient defined in Eq. (52). For the first term, by Hölder's inequality (as in the proof of Theorem B.1), $\sum_{(s,a)\neq(s,\mathring{\pi}(s))}x_t(s,a)^{1/\alpha} \leq H^{1/\alpha}(SA)^{1-1/\alpha}$. Combined with $\sum_{t=t_i}^{t_{i+1}-1}(t - t_i + 1)^{1/\alpha-1} \leq \alpha(t_{i+1} - t_i)^{1/\alpha} \leq \alpha T^{1/\alpha}$ and $N \leq \mathcal{O}(SA\log(T))$, we now aggregate the two contributions of $M$ from Eq. (52) separately:

- *Penalty contribution.* Using the second piece of $M$, namely $\mathcal{O}(H \cdot S^{2-2/\alpha} \cdot A^{2-3/\alpha+1/\alpha^2})$,

$$H \cdot S^{2-\frac{2}{\alpha}} \cdot A^{2-\frac{3}{\alpha}+\frac{1}{\alpha^2}} \sum_{t=1}^{T} \sigma(t - t_{i(t)} + 1)^{1/\alpha-1} \cdot \sum_{(s,a)\neq(s,\hat{\pi}(s))} x_t(s,a)^{1/\alpha}$$

$$\leq H^{1+\frac{1}{\alpha}} \cdot S^{3-\frac{3}{\alpha}} \cdot A^{3-\frac{4}{\alpha}+\frac{1}{\alpha^2}} \sigma \cdot \alpha N T^{1/\alpha} = \mathcal{O}\left(\sigma H^{1+\frac{1}{\alpha}} \cdot S^{4-\frac{3}{\alpha}} \cdot A^{4-\frac{4}{\alpha}+\frac{1}{\alpha^2}} T^{1/\alpha} \log(T)\right).$$

- *Stability contribution.* Using the first piece of $M$, namely $\mathcal{O}(H^{2-1/\alpha} \cdot S^{-1/\alpha} \cdot A^{-(2\alpha-1)/\alpha^2})$,

$$H^{2-\frac{1}{\alpha}} \cdot S^{-\frac{1}{\alpha}} \cdot A^{-\frac{2\alpha-1}{\alpha^2}} \sum_{t=1}^{T} \sigma(t - t_{i(t)} + 1)^{1/\alpha-1} \cdot \sum_{(s,a)\neq(s,\hat{\pi}(s))} x_t(s,a)^{1/\alpha}$$

$$\leq H^{(2-\frac{1}{\alpha})+\frac{1}{\alpha}} \cdot S^{-\frac{1}{\alpha}+(1-\frac{1}{\alpha})+1} \cdot A^{-\frac{2\alpha-1}{\alpha^2}+(1-\frac{1}{\alpha})+1} \cdot \sigma \cdot \alpha T^{1/\alpha} \log(T)$$

$$= \mathcal{O}\left(H^2 \cdot S^{2-\frac{2}{\alpha}} \cdot A^{2-\frac{3}{\alpha}+\frac{1}{\alpha^2}} \cdot \sigma T^{1/\alpha} \log(T)\right).$$

Therefore:

$$\mathbb{E}\left[\text{EstReg}\right] \leq \mathcal{O}\Big(\sigma H^{1+\frac{1}{\alpha}} \cdot S^{4-\frac{3}{\alpha}} \cdot A^{4-\frac{4}{\alpha}+\frac{1}{\alpha^2}} T^{1/\alpha} \log(T) + \sigma H^2 \cdot S^{2-\frac{2}{\alpha}} \cdot A^{2-\frac{3}{\alpha}+\frac{1}{\alpha^2}} T^{1/\alpha} \log(T)$$

$$+ H\sigma S A T^3 \delta + H^2 \sigma T \delta\Big).$$

The skipping error shown in Lemma C.9, together with the failure-event control in Lemma C.7, gives the unconditional bound below. Here the $\ell_t - \ell_t^{\text{skip}} + b_t^{\text{skip}}$ component is handled by the same deterministic skipping-bias argument as in Lemma C.9; only the $J_t$ component, namely $\ell_t^{\text{skip}} - \widehat{\ell}_t^{\text{IS}}$, needs the extra $\mathcal{E}^c$ control. Eq. (50) and Eq. (49) bound these two terms under $\mathcal{E}^c$ with weights $x_t(s,a) + \rho^{\widehat{\mathbb{P}}_{i(t)},\hat{\pi}}(s,a)$, whose total mass is at most $2H$.

$$\mathbb{E}\left[\text{SkipErr}\right] \leq \mathbb{E}\left[O\left(M \cdot \sum_{i=1}^{N} \sum_{t=t_i}^{t_{i+1}-1} \sigma(t - t_{i(t)} + 1)^{1/\alpha-1} \cdot \sum_{(s,a)\neq(s,\hat{\pi}(s))} x_t(s,a)^{1/\alpha}\right)\right]$$

$$+ \mathbb{E}\left[2\sigma \sum_{t=1}^{T} \sum_{(s,a)\in\mathcal{S}\times\mathcal{A}} |u_t(s,a) - \rho^{\mathbb{P},\pi_t}(s,a)|\right] + \mathcal{O}(H\sigma S A T^3 \delta)$$

$$\leq \mathcal{O}\Big(\sigma H^{1+\frac{1}{\alpha}} \cdot S^{4-\frac{3}{\alpha}} \cdot A^{4-\frac{4}{\alpha}+\frac{1}{\alpha^2}} T^{1/\alpha} \log(T) + \sigma H^2 \cdot S^{2-\frac{2}{\alpha}} \cdot A^{2-\frac{3}{\alpha}+\frac{1}{\alpha^2}} T^{1/\alpha} \log(T)$$

$$+ H\sigma S A T^3 \delta\Big) + 2\sigma \cdot \mathbb{E}\left[\sum_{t=1}^{T} \sum_{(s,a)\in\mathcal{S}\times\mathcal{A}} |u_t(s,a) - \rho^{\mathbb{P},\pi_t}(s,a)|\right],$$

where the aggregate calculation parallels that for $\mathbb{E}[\text{EstReg}]$ above (with $M$ as in Eq. (52)), splitting into the penalty and stability contributions identically. By the analysis in Appendix C.2 in Jin et al. (2021), we further have

$$\mathbb{E}\left[\sum_{t=1}^{T} \sum_{(s,a)\in\mathcal{S}\times\mathcal{A}} |u_t(s,a) - \rho^{\mathbb{P},\pi_t}(s,a)|\right] \leq \mathcal{O}(HS\sqrt{AT\log(\iota)} + H^2 S^3 A^2 \log^2(\iota) + SAT\delta).$$

Therefore, we have

$$\mathbb{E}\left[\text{SkipErr}\right] \leq \mathcal{O}\Big(\sigma H^{1+\frac{1}{\alpha}} \cdot S^{4-\frac{3}{\alpha}} \cdot A^{4-\frac{4}{\alpha}+\frac{1}{\alpha^2}} T^{1/\alpha} \log(T) + \sigma H^2 \cdot S^{2-\frac{2}{\alpha}} \cdot A^{2-\frac{3}{\alpha}+\frac{1}{\alpha^2}} T^{1/\alpha} \log(T) + H\sigma S A T^3 \delta\Big)$$

$$+ \mathcal{O}\left(H\sigma S\sqrt{AT\log(\iota)} + H^2 \sigma S^3 A^2 \log^2(\iota) + \sigma SAT\delta\right).$$

By Lemma C.5, we have

$$\mathbb{E}[\text{PessErr}] \leq O(H^2 \sigma T \delta).$$

Combining the bounds for $\mathbb{E}[\mathrm{EstReg}], \mathbb{E}[\mathrm{SkipErr}], \mathbb{E}[\mathrm{TransErr}]$, and $\mathbb{E}[\mathrm{PessErr}]$ above: with $\delta = 1/T^3$, all $\delta$-dependent terms are independent of $T$ up to problem-dependent polynomial factors and are absorbed by the $\widetilde{\mathcal{O}}$ notation. The remaining contributions are: (i) the leading $T^{1/\alpha}$ *penalty* term $\widetilde{\mathcal{O}}(\sigma H^{1+1/\alpha} S^{4-3/\alpha} A^{4-4/\alpha+1/\alpha^2} T^{1/\alpha})$ from EstReg and SkipErr; (ii) the $T^{1/\alpha}$ *stability* term $\widetilde{\mathcal{O}}(\sigma H^2 S^{2-2/\alpha} A^{2-3/\alpha+1/\alpha^2} T^{1/\alpha})$ from EstReg and SkipErr; (iii) the $\sqrt{T}$ term $\widetilde{\mathcal{O}}(H\sigma S\sqrt{AT})$ from the transition concentration in TransErr and SkipErr's UOB component; and (iv) the polylogarithmic-in-$T$ term $\mathcal{O}(H^2 \sigma S^3 A^3 \log^2 \iota)$, which is absorbed by the displayed $\widetilde{\mathcal{O}}$ bound. We obtain the regret bound in the adversarial regime:

$$\mathrm{Reg}_T(\mathring{\pi}) \le \widetilde{\mathcal{O}}\left(H^{1+\frac{1}{\alpha}} S^{4-\frac{3}{\alpha}} A^{4-\frac{4}{\alpha}+\frac{1}{\alpha^2}} \cdot \sigma T^{1/\alpha} + H^2 S^{2-\frac{2}{\alpha}} A^{2-\frac{3}{\alpha}+\frac{1}{\alpha^2}} \cdot \sigma T^{1/\alpha} + H\sigma S\sqrt{AT}\right)$$

## C.7. Proof for Self-Bounding Regime

Following Jin et al. (2021, Appendix C), we introduce the *path-dependent occupancy* $q_t^\star : \mathcal{S} \times \mathcal{A} \to [0,1]$:

$$q_t^\star(s,a) := \pi_t(a \mid s) \cdot \rho^{\mathbb{P},\mathring{\pi}}(s), \qquad \text{where } \rho^{\mathbb{P},\mathring{\pi}}(s) := \sum_{a' \in \mathcal{A}} \rho^{\mathbb{P},\mathring{\pi}}(s,a'). \qquad (63)$$

Intuitively, $q_t^\star(s,a)$ is the probability of visiting $(s,a)$ when the trajectory follows $\mathring{\pi}$ up to (but not including) the layer of $s$, then takes action $a$ under the algorithm's policy $\pi_t$. Since $\pi_t$ is determined by $\widehat{\ell}_1, \ldots, \widehat{\ell}_{t-1}$ and $\mathring{\pi}$ is fixed, $q_t^\star \in \mathcal{F}_{t-1}$.

For brevity we also write $q_t(s,a) := \rho^{\mathbb{P},\pi_t}(s,a)$ and $\widehat{q}_t(s,a) := \rho^{\widehat{\mathbb{P}}_{i(t)},\pi_t}(s,a)$.

Similar to the analysis in adversarial regime, by setting $\delta = T^{-3}$, the regret contribution from $\mathcal{E}^c$ is independent of $T$ up to problem-dependent polynomial factors and is absorbed by the final $\mathcal{O}(\cdot)$ terms. Therefore, we only consider the case when $\mathcal{E}$ happens.

In self-bounding regime, we consider the $(\mathring{\pi}, \Delta, C_{\mathrm{sb}})$ self-bounding environment (Definition 3.1). We first analyze the estimation regret term and skipping error term. By Lemmas C.8 and C.9 and similar arguments in Appendix C.6, the master sum reads

$$\mathbb{E}\left[\mathrm{SkipErr} + \mathrm{EstReg}\right] \le \mathbb{E}\left[\mathcal{O}\left(M \cdot \sum_{i=1}^{N} \sum_{t=t_i}^{t_{i+1}-1} \sigma(t - t_{i(t)} + 1)^{1/\alpha - 1} \cdot \sum_{(s,a) \ne (s,\mathring{\pi}(s))} x_t(s,a)^{1/\alpha}\right)\right]$$

$$+ 2\sigma \cdot \mathbb{E}\left[\sum_{t=1}^{T} \sum_{(s,a) \in \mathcal{S} \times \mathcal{A}} |u_t(s,a) - \rho^{\mathbb{P},\pi_t}(s,a)|\right]$$

$$+ \mathcal{O}\left(H^2 \sigma S A T^3 \delta + H S^2 A^2 \sigma \log(T) + H^{1-\frac{1}{\alpha}} S^{3-\frac{1}{\alpha}} A^{4-\frac{4}{\alpha}+\frac{1}{\alpha^2}} \sigma \log^2(T)\right),$$

where the per-step dominant coefficient is from Eq. (52).

$$M := \mathcal{O}\left(H^{2-\frac{1}{\alpha}} \cdot S^{-\frac{1}{\alpha}} \cdot A^{-\frac{2\alpha-1}{\alpha^2}}\right) + \mathcal{O}\left(H \cdot S^{2-\frac{2}{\alpha}} \cdot A^{2-\frac{3}{\alpha}+\frac{1}{\alpha^2}}\right).$$

The empirical master sum in the first line is controlled by Lemma C.16. This lemma isolates the only point where the empirical occupancy $x_t = \rho^{\widehat{\mathbb{P}}_{i(t)},\pi_t}$ must be compared with the true occupancy $\rho^{\mathbb{P},\pi_t}$, and gives the bound in Eq. (70).

Notice that Jin et al. (2021) gives the instance-dependent bound over the last term. By Appendix C.3 in Jin et al. (2021), we have

$$\mathbb{E}\left[\sum_{t=1}^{T} \sum_{(s,a) \in \mathcal{S} \times \mathcal{A}} |u_t(s,a) - \rho^{\mathbb{P},\pi_t}(s,a)|\right] \le 16\mathbb{E}[\mathbb{G}_3(\log(\iota))] + \mathcal{O}(H^2 S^3 A^2 \log^2(\iota)), \qquad (64)$$

where $\mathbb{G}_3(\iota)$ is bounded by Lemma C.15.

Next, we use the definition of self-bounding terms and the related lemmas from Jin et al. (2021) to bound $\mathbb{E}[\mathrm{TransErr} +$

PESSERR]. Especially, by the analysis in Appendix C.3 in Jin et al. (2021), we can decompose

$$\mathbb{E}[\text{TRANSERR} + \text{PESSERR}]$$

$$= \mathbb{E}\left[\sum_{t=1}^{T} \sum_{(s,a):a \neq \mathring{\pi}(s)} q_t(s,a)\, \widehat{E}_t^{\mathring{\pi}}(s,a)\right] \tag{ERRSUB}$$

$$+ \mathbb{E}\left[\sum_{t=1}^{T} \sum_{(s,a):a = \mathring{\pi}(s)} (q_t(s,a) - q_t^{\star}(s,a))\, \widehat{E}_t^{\mathring{\pi}}(s,a)\right] \tag{ERROPT}$$

$$+ \mathbb{E}\left[\sum_{t=1}^{T} \sum_{(s,a):a \neq \mathring{\pi}(s)} (q_t(s,a) - \widehat{q}_t(s,a)) \left(Q^{\widehat{\mathbb{P}}_{i(t)},\mathring{\pi}}(s,a;\boldsymbol{\ell}_t^{\text{pess}}) - V^{\widehat{\mathbb{P}}_{i(t)},\mathring{\pi}}(s;\boldsymbol{\ell}_t^{\text{pess}})\right)\right] \tag{OCCDIFF}$$

$$+ \mathbb{E}\left[\sum_{t=1}^{T} \sum_{(s,a):a \neq \mathring{\pi}(s)} q_t^{\star}(s,a) \left(V^{\widehat{\mathbb{P}}_{i(t)},\mathring{\pi}}(s;\boldsymbol{\ell}_t^{\text{pess}}) - V^{\mathbb{P},\mathring{\pi}}(s;\boldsymbol{\ell}_t)\right)\right], \tag{BIAS}$$

where for $h \in [H]$ and $(s_h, a) \in \mathcal{S}_h \times \mathcal{A}$,

$$\widehat{E}_t^{\mathring{\pi}}(s_h, a) = \ell_t(s_h, a) + \sum_{s' \in \mathcal{S}_{h+1}} \mathbb{P}[s' \mid s_h, a] V^{\widehat{\mathbb{P}}_{i(t)},\mathring{\pi}}(s';\boldsymbol{\ell}_t^{\text{pess}}) - Q^{\widehat{\mathbb{P}}_{i(t)},\mathring{\pi}}(s_h, a;\boldsymbol{\ell}_t^{\text{pess}}).$$

This is the standard decomposition of Jin et al. (2021, Corollary C.1).

Notice that under $\mathcal{E}_{i(t)}$, the algebraic identity holds:

$$\widehat{E}_t^{\mathring{\pi}}(s_h, a) = \ell_t(s_h, a) + \sum_{s' \in \mathcal{S}_{h+1}} \mathbb{P}[s' \mid s_h, a] V^{\widehat{\mathbb{P}}_{i(t)},\mathring{\pi}}(s';\boldsymbol{\ell}_t^{\text{pess}}) - Q^{\widehat{\mathbb{P}}_{i(t)},\mathring{\pi}}(s_h, a;\boldsymbol{\ell}_t^{\text{pess}})$$

$$= H\sigma \cdot B_{i(t)}(s_h, a) + \sum_{s' \in \mathcal{S}_{h+1}} (\mathbb{P}[s' \mid s_h, a] - \widehat{\mathbb{P}}_{i(t)}[s' \mid s_h, a]) V^{\widehat{\mathbb{P}}_{i(t)},\mathring{\pi}}(s';\boldsymbol{\ell}_t^{\text{pess}}),$$

Taking conditional expectation given $\mathcal{F}_{t-1}$ and bounding by $|\mathbb{E}[V^{\widehat{\mathbb{P}}_{i(t)},\mathring{\pi}}(s';\boldsymbol{\ell}_t^{\text{pess}}) \mid \mathcal{F}_{t-1}]| \leq H \cdot \max\{\sigma, H\sigma\} = H^2\sigma$, under $\mathcal{E}_{i(t)}$ we have $|\mathbb{P} - \widehat{\mathbb{P}}_{i(t)}| \leq B_{i(t)}(s_h, a, s')$ entrywise. Combining with $\sum_{s'} B_{i(t)}(s_h, a, s') \leq 1$ when $B_{i(t)}(s_h, a) = \sum_{s'} B_{i(t)}(s_h, a, s')$ (and $B_{i(t)}(s_h, a) = 1$ otherwise, with $\sum |\mathbb{P} - \widehat{\mathbb{P}}| \leq 2$), and applying the sign argument analogous to Lemma C.4:

$$\mathbb{E}[\widehat{E}_t^{\mathring{\pi}}(s_h, a) \mid \mathcal{F}_{t-1}] \leq H\sigma \cdot B_{i(t)}(s_h, a) + 2H^2\sigma \cdot B_{i(t)}(s_h, a) = (1 + 2H)H\sigma \cdot B_{i(t)}(s_h, a) \leq 3H^2\sigma \cdot B_{i(t)}(s_h, a),$$

where the last inequality uses $H \geq 1$ to obtain $(1 + 2H) \leq 3H$. This $3H^2\sigma$ multiplier is precisely the heavy-tail analog of Jin et al.'s pointwise value bound $\mathcal{O}(L^2)$ in the bounded-loss setting, and matches the squared multiplier needed for the downstream self-bounding term argument $\mathbb{G}_1(\mathcal{O}(H^4\sigma^2 S \log \iota)) = \mathbb{G}_1(D^2 H^2 S \log \iota)$ stated below.

Combined with the $\mathcal{F}_{t-1}$-measurability of $q_t$ and $q_t^{\star}$ (both built from $\pi_t \in \mathcal{F}_{t-1}$ and the fixed $\mathring{\pi}$ via equation 63), the tower property yields

$$\mathbb{E}[\text{ERROPT}] = \mathbb{E}\left[\sum_{t=1}^{T} \sum_{s \in \mathcal{S}} (q_t(s, \mathring{\pi}(s)) - q_t^{\star}(s, \mathring{\pi}(s))) \cdot \mathbb{E}[\widehat{E}_t^{\mathring{\pi}}(s, \mathring{\pi}(s)) \mid \mathcal{F}_{t-1}]\right]$$

$$\leq \mathbb{E}\left[\sum_{t=1}^{T} \sum_{s \in \mathcal{S}} |q_t(s, \mathring{\pi}(s)) - q_t^{\star}(s, \mathring{\pi}(s))| \cdot |\mathbb{E}[\widehat{E}_t^{\mathring{\pi}}(s, \mathring{\pi}(s)) \mid \mathcal{F}_{t-1}]|\right],$$

which is the absolute-value form of $\mathbb{G}_2$ in Definition C.12. The $\mathcal{E}_{i(t)}^c$ contribution is absorbed by the standard $\mathcal{O}(H^2\sigma T\delta)$ bookkeeping with $\delta = 1/T^3$ (cf. Lemma C.5). The analogous step for ERRSUB is automatic: $q_t(s, a) \geq 0$ on the sub-optimal index set, so the one-sided upper bound above suffices without needing the lower direction.

By the same process in Appendix B.3 in Jin et al. (2021), we have

$$\mathbb{E}\left[\text{ERRSUB}\right] \leq \mathcal{O}\left(\mathbb{E}\left[\mathbb{G}_1(D^2 H^2 S \log(\iota))\right] + D^2 S^2 A \log^2(\iota)\right), \tag{65}$$

$$\mathbb{E}\left[\text{ERROPT}\right] \leq \mathcal{O}\left(\mathbb{E}\left[\mathbb{G}_2(D^2 H^2 S \log(\iota))\right] + D^2 S^2 A \log^2(\iota)\right). \tag{66}$$

We note that under $\mathcal{E}_{i(t)}$,

$$\left|\mathbb{E}\left[Q^{\widehat{\mathbb{P}}_{i(t)},\mathring{\hat{\pi}}}(s,a;\ell_t^{\text{pess}}) - V^{\widehat{\mathbb{P}}_{i(t)},\mathring{\hat{\pi}}}(s;\ell_t^{\text{pess}}) \mid \mathcal{F}_{t-1}\right]\right| \leq 3H^2\sigma,$$

by the triangle inequality applied to the two conditional value bounds $\left|\mathbb{E}[Q^{\widehat{\mathbb{P}}_{i(t)},\mathring{\hat{\pi}}}(s,a;\ell_t^{\text{pess}}) \mid \mathcal{F}_{t-1}]\right| \leq H^2\sigma + H\sigma \leq 2H^2\sigma$ and $\left|\mathbb{E}[V^{\widehat{\mathbb{P}}_{i(t)},\mathring{\hat{\pi}}}(s;\ell_t^{\text{pess}}) \mid \mathcal{F}_{t-1}]\right| \leq H^2\sigma$ (Lemma C.5). Then by the same analysis in Appendix B.3 in Jin et al. (2021), we have

$$\mathbb{E}\left[\text{OCCDIFF}\right] \leq \mathcal{O}\left(H^4\sigma^2 S^3 A^2 \log^2(\iota) + \mathbb{E}[\mathbb{G}_3(H^4\sigma^2 \log(\iota))]\right) \tag{67}$$

Moreover, on the per-epoch event $\mathcal{E}_{i(t)} \in \mathcal{F}_{t-1}$, Lemma C.4 yields $\mathbb{E}[V^{\widehat{\mathbb{P}}_{i(t)},\mathring{\hat{\pi}}}(s;\ell_t^{\text{pess}}) - V^{\mathbb{P},\mathring{\hat{\pi}}}(s;\ell_t) \mid \mathcal{F}_{t-1}] \leq 0$. Combined with $q_t^\star(s,a) \geq 0$ entrywise and the $\mathcal{F}_{t-1}$-measurability of $q_t^\star$, the tower property gives $\mathbb{E}[\text{BIAS}] \leq 0$ (with the $\mathcal{E}_{i(t)}^c$ tail absorbed by the standard $\mathcal{O}(\cdot)$ bookkeeping as in Lemma C.5). Therefore, we can drop BIAS from the right-hand side.

We now combine the established bounds for $\mathbb{G}_1, \mathbb{G}_2, \mathbb{G}_3$ (Lemmas C.13 to C.15) and sum up Eqs. (64) to (67) and (70). Throughout this combination, the symbols $\alpha', \beta', \gamma' \in (0,1]$ are unified scalars to be chosen later. Each individual invocation of Lemmas C.13 to C.15 uses internal parameters tailored to the specific term being bounded: for instance, Lemma C.16 invokes Lemma C.15 with $\alpha_0 = \alpha'/(cH\sigma\gamma')$, $\beta_0 = \beta'/(cH\sigma\gamma')$ for an absolute constant $c$, and then multiplies the resulting bound by $\gamma'$ (from a Young inequality). All such internal rescalings only contribute absolute-constant factors, which are absorbed into the $\Lambda_1, \Lambda_2, \Lambda_3$ coefficients defined below. With this convention we obtain

$$\text{Reg}_T(\mathring{\hat{\pi}}) \leq \mathbb{E}[\text{ESTREG} + \text{SKIPERR}] + \mathbb{E}[\text{TRANSERR} + \text{PESSERR}]$$
$$\leq \mathcal{O}\Big(H^4\sigma^2 S^3 A^2 \log^2(\iota) + H^{1-\frac{1}{\alpha}} S^{3-\frac{1}{\alpha}} A^{4-\frac{4}{\alpha}+\frac{1}{\alpha^2}} \sigma \log^2(T) + (\alpha' + \beta' + \gamma')(\text{Reg}_T(\mathring{\hat{\pi}}) + C_{\text{sb}})$$
$$+ {\alpha'}^{-1} H^6 \sigma^2 S^2 \omega_\Delta(2) \log(\iota) + {\beta'}^{-1} H^6 \sigma^2 S^2 \Delta_{\min}^{-1} \log(\iota) + {\gamma'}^{-\frac{1}{\alpha-1}} \cdot H^{\frac{2\alpha-1}{\alpha-1}} S^3 A^{3-\frac{1}{\alpha}} \cdot \sigma^{\frac{\alpha}{\alpha-1}} \omega_\Delta(\alpha) \log(T)^2\Big),$$

where $\alpha', \beta', \gamma' > 0$ are parameters to be chosen, and

$$\omega_\Delta(x) = \sum_{(s,a):a\neq\mathring{\pi}(s)} \Delta(s,a)^{-\frac{1}{x-1}}, \quad \Delta_{\min} = \min_{(s,a):a\neq\mathring{\pi}(s)} \Delta(s,a).$$

Rearranging the terms, we have

$$(1 - (\alpha' + \beta' + \gamma'))\text{Reg}_T(\mathring{\hat{\pi}}) \leq (\alpha' + \beta' + \gamma')C_{\text{sb}} + \kappa + {\alpha'}^{-1}\Lambda_1 + {\beta'}^{-1}\Lambda_2 + {\gamma'}^{-\frac{1}{\alpha-1}}\Lambda_3, \tag{68}$$

where we define the following constants:

$$\kappa = \mathcal{O}\left(H^4\sigma^2 S^3 A^2 \log^2(\iota) + H^{1-\frac{1}{\alpha}} S^{3-\frac{1}{\alpha}} A^{4-\frac{4}{\alpha}+\frac{1}{\alpha^2}} \sigma \log^2(T)\right),$$
$$\Lambda_1 = \mathcal{O}\left(H^6\sigma^2 S^2 \omega_\Delta(2)\log(\iota)\right),$$
$$\Lambda_2 = \mathcal{O}\left(H^6\sigma^2 S^2 \Delta_{\min}^{-1}\log(\iota)\right),$$
$$\Lambda_3 = \mathcal{O}\left(H^{\frac{2\alpha-1}{\alpha-1}} S^3 A^{3-\frac{1}{\alpha}} \cdot \sigma^{\frac{\alpha}{\alpha-1}} \omega_\Delta(\alpha)\log(T)^2\right).$$

**Case 1:** $C_{\text{sb}} = 0$. Setting $\alpha' = \beta' = \gamma' = 1/6$, we can absorb the regret term to the left side:

$$\text{Reg}_T(\mathring{\hat{\pi}}) \leq 2\kappa + 12\Lambda_1 + 12\Lambda_2 + 6^{\frac{1}{\alpha-1}}\Lambda_3.$$

**Case 2: $C_{\mathrm{sb}} \geq C^\star$.** Define the threshold

$$C^\star := \max\left\{36\Lambda_1,\ 36\Lambda_2,\ 6^{\alpha/(\alpha-1)}\Lambda_3\right\}.$$

When $C_{\mathrm{sb}} \geq C^\star$, we balance each pair of terms in Eq. (68):

- For $\alpha'$: balancing $\alpha' C_{\mathrm{sb}}$ and $\alpha'^{-1}\Lambda_1$, set $\alpha' = \sqrt{\Lambda_1/C_{\mathrm{sb}}}$ yielding $2\sqrt{\Lambda_1 C_{\mathrm{sb}}}$; the condition $C_{\mathrm{sb}} \geq 36\Lambda_1$ ensures $\alpha' \leq 1/6$.

- For $\beta'$: balancing $\beta' C_{\mathrm{sb}}$ and $\beta'^{-1}\Lambda_2$, set $\beta' = \sqrt{\Lambda_2/C_{\mathrm{sb}}}$ yielding $2\sqrt{\Lambda_2 C_{\mathrm{sb}}}$; the condition $C_{\mathrm{sb}} \geq 36\Lambda_2$ ensures $\beta' \leq 1/6$.

- For $\gamma'$: balancing $\gamma' C_{\mathrm{sb}}$ and $\gamma'^{-\frac{1}{\alpha-1}}\Lambda_3$, set $\gamma' = (\Lambda_3/C_{\mathrm{sb}})^{\frac{\alpha-1}{\alpha}}$ yielding $2C_{\mathrm{sb}}^{1/\alpha}\Lambda_3^{\frac{\alpha-1}{\alpha}}$; the condition $C_{\mathrm{sb}} \geq 6^{\alpha/(\alpha-1)}\Lambda_3$ ensures $\gamma' \leq 1/6$.

The three constraints together imply $\alpha' + \beta' + \gamma' \leq 1/2$, so the rearrangement of Eq. (68) yields

$$\mathrm{Reg}_T(\mathring{\pi}) \leq 2\kappa + 4\sqrt{\Lambda_1 C_{\mathrm{sb}}} + 4\sqrt{\Lambda_2 C_{\mathrm{sb}}} + 4C_{\mathrm{sb}}^{1/\alpha}\Lambda_3^{\frac{\alpha-1}{\alpha}}.$$

**Case 3: $0 < C_{\mathrm{sb}} < C^\star$.** In this intermediate regime, the balanced choices above would violate $\alpha', \beta', \gamma' \leq 1/6$. We instead fall back to the Case 1 choices $\alpha' = \beta' = \gamma' = 1/6$ (so $\alpha' + \beta' + \gamma' = 1/2$), giving

$$\mathrm{Reg}_T(\mathring{\pi}) \leq 2\kappa + 12\Lambda_1 + 12\Lambda_2 + 2 \cdot 6^{1/(\alpha-1)}\Lambda_3 + C_{\mathrm{sb}}.$$

Since $C_{\mathrm{sb}} < C^\star \leq \mathcal{O}(\Lambda_1 + \Lambda_2 + \Lambda_3)$ for any fixed $\alpha \in (1,2]$, the $C_{\mathrm{sb}}$ term is absorbed into the $\Lambda_i$ contributions.

**Final Bound.** Combining Case 1, Case 2, and Case 3, we have

$$\mathrm{Reg}_T(\mathring{\pi}) \leq \mathcal{O}\left(\kappa + \Lambda_1 + \Lambda_2 + \Lambda_3 + \sqrt{\Lambda_1 C_{\mathrm{sb}}} + \sqrt{\Lambda_2 C_{\mathrm{sb}}} + C_{\mathrm{sb}}^{1/\alpha}\Lambda_3^{\frac{\alpha-1}{\alpha}}\right).$$

Substituting the definitions of $\kappa, \Lambda_1, \Lambda_2, \Lambda_3$, we obtain the final self-bounding regret bound:

$$\begin{aligned}
\mathrm{Reg}_T(\mathring{\pi}) \leq \mathcal{O}\Big( &H^4\sigma^2 S^3 A^2 \log^2(\iota) + H^{1-\frac{1}{\alpha}}S^{3-\frac{1}{\alpha}}A^{4-\frac{4}{\alpha}+\frac{1}{\alpha^2}}\sigma\log^2(T) \\
&+ H^6\sigma^2 S^2\omega_\Delta(2)\log(\iota) + H^6\sigma^2 S^2\Delta_{\min}^{-1}\log(\iota) + H^{\frac{2\alpha-1}{\alpha-1}}S^3 A^{3-\frac{1}{\alpha}}\cdot\sigma^{\frac{\alpha}{\alpha-1}}\omega_\Delta(\alpha)\log(T)^2 \\
&+ H^3\sigma S\sqrt{\omega_\Delta(2)C_{\mathrm{sb}}\log(\iota)} + H^3\sigma S\sqrt{\Delta_{\min}^{-1}C_{\mathrm{sb}}\log(\iota)} \\
&+ C_{\mathrm{sb}}^{1/\alpha}\cdot H^{2-\frac{1}{\alpha}}\cdot S^{3-\frac{3}{\alpha}}A^{3-\frac{4}{\alpha}+\frac{1}{\alpha^2}}\cdot\sigma\cdot\omega_\Delta(\alpha)^{\frac{\alpha-1}{\alpha}}\log^{\frac{2(\alpha-1)}{\alpha}}(T)\Big).
\end{aligned}$$

## C.8. Auxiliary Lemmas

**Lemma C.10** (UOB construction property). *By construction of the upper occupancy bound, $\widehat{\mathbb{P}}_{i(t)} \in \widehat{\mathcal{P}}_{i(t)}$ is itself a member of the confidence set, so the max defining $u_t$ over $\widehat{\mathcal{P}}_{i(t)}$ includes the algorithm's empirical kernel. Therefore*

$$u_t(s,a) \geq x_t(s,a) = \rho^{\widehat{\mathbb{P}}_{i(t)},\pi_t}(s,a) \quad \text{for all } t \geq 1,\ s \in \mathcal{S},\ a \in \mathcal{A}.$$

**Lemma C.11** (Lemma C.1.8 in Jin et al. (2021)).

$$\mathbb{E}\left[\sum_{t=1}^{T}\sum_{(s,a)\in\mathcal{S}\times\mathcal{A}} x_t(s,a)\cdot(B_{i(t)}(s,a))^2\right] = O\left(H^2 S^3 A^2\log(\iota)^2 + SAT\delta\right), \tag{69}$$

*where $\log(\iota) = \log(HSAT/\delta)$.*

We then involve the definitions of self-bounding terms in Jin et al. (2021) for analyzing the self-bounding regime:

**Definition C.12** (Self-bounding Terms). For some mapping $\mathring{\pi}: S \to A$, define the following:

$$\mathbb{G}_1(J) = \sum_{t=1}^{T} \sum_{(s,a) \neq (s,\mathring{\pi}(s))} \rho^{\mathbb{P},\pi_t}(s,a) \cdot \sqrt{\frac{J}{\max\{m_{i(t)}(s,a),1\}}}$$

$$\mathbb{G}_2(J) = \sum_{t=1}^{T} \sum_{s \in \mathcal{S}} \left| q_t(s,\mathring{\pi}(s)) - q_t^{\star}(s,\mathring{\pi}(s)) \right| \cdot \sqrt{\frac{J}{\max\{m_{i(t)}(s,\mathring{\pi}(s)),1\}}}$$

$$\mathbb{G}_3(J) = \sum_{t=1}^{T} \sum_{h=1}^{H} \sum_{\substack{s \in \mathcal{S}_h \\ a \neq \mathring{\pi}(s)}} \sum_{h'=1}^{h-1} \sum_{\substack{u \in \mathcal{S}_{h'}, v \in \mathcal{A} \\ w \in \mathcal{S}_{h'+1}}} \rho^{\mathbb{P},\pi_t}(u,v) \cdot \sqrt{\frac{\mathbb{P}[w \mid u,v] \cdot J}{\max\{m_{i(t)}(u,v),1\}}} \, q_t(s,a \mid w),$$

where $q_t(s,a \mid w)$ is the probability that starting from state $w$ and visit $(s,a)$ under transition $\mathbb{P}$ and policy $\pi_t$.

Jin et al. (2021) then gives the bounds for these self-bounding terms:

**Lemma C.13** (Lemma D.2.2 in Jin et al. (2021)). *Under Definition 3.1, for any $\alpha_0 \in \mathbb{R}_+$ (we use the subscript to avoid notational clash with the Tsallis exponent $\alpha \in (1,2]$), we have*

$$\mathbb{E}[\mathbb{G}_1(J)] \leq \alpha_0 \cdot (\text{Reg}_T(\mathring{\pi}) + C_{\text{sb}}) + \frac{1}{\alpha_0} \sum_{(s,a) \neq (s,\mathring{\pi}(s))} \frac{8J}{\Delta(s,a)}.$$

**Lemma C.14** (Lemma D.2.3 in Jin et al. (2021)). *Under Definition 3.1, for any $\beta_0 \in \mathbb{R}_+$, we have*

$$\mathbb{E}[\mathbb{G}_2(J)] \leq \beta_0 \cdot (\text{Reg}_T(\mathring{\pi}) + C_{\text{sb}}) + \frac{1}{\beta_0} \frac{8SHJ}{\Delta_{\min}}.$$

**Lemma C.15** (Lemma D.2.4 in Jin et al. (2021)). *Under Definition 3.1, for any $\alpha_0, \beta_0 \in \mathbb{R}_+$, we have*

$$\mathbb{E}[\mathbb{G}_3(J)] \leq (\alpha_0 + \beta_0) \cdot (\text{Reg}_T(\mathring{\pi}) + C_{\text{sb}}) + \frac{1}{\alpha_0} \sum_{(s,a) \neq (s,\mathring{\pi}(s))} \frac{8H^2 SJ}{\Delta(s,a)} + \frac{1}{\beta_0} \cdot \frac{8H^2 S^2 J}{\Delta_{\min}}.$$

**Lemma C.16** (Empirical-to-true self-bounding reduction). *Suppose Definition 3.1 holds. We use a normalized representative of the gap function such that $\Delta(s,a) \leq \mathcal{O}(H\sigma)$ for every suboptimal pair $(s,a)$. Let $M$ be the per-step dominant coefficient defined in Eq. (52), i.e.,*

$$M := \mathcal{O}\big(H^{2-\frac{1}{\alpha}} \cdot S^{-\frac{1}{\alpha}} \cdot A^{-\frac{2\alpha-1}{\alpha^2}}\big) + \mathcal{O}\big(H \cdot S^{2-\frac{2}{\alpha}} \cdot A^{2-\frac{3}{\alpha}+\frac{1}{\alpha^2}}\big).$$

*Then for any $\alpha', \beta', \gamma' \in (0,1]$,*

$$\mathbb{E}\Big[\mathcal{O}\Big(M \sum_i \sum_t \sigma(t - t_{i(t)} + 1)^{1/\alpha - 1} \sum_{(s,a) \neq (s,\mathring{\pi}(s))} x_t(s,a)^{1/\alpha}\Big)\Big]$$

$$\leq \mathcal{O}\Big((\alpha' + \beta' + \gamma')\big(\text{Reg}_T(\mathring{\pi}) + C_{\text{sb}}\big) + \gamma'^{-\frac{1}{\alpha-1}} H^{\frac{2\alpha-1}{\alpha-1}} S^3 A^{3-\frac{1}{\alpha}} \sigma^{\alpha/(\alpha-1)} \omega_\Delta(\alpha) \log^2 T \tag{70}$$

$$+ \alpha'^{-1} H^4 \sigma^2 S^2 \omega_\Delta(2) \log \iota + \beta'^{-1} H^4 \sigma^2 S^2 \Delta_{\min}^{-1} \log \iota + H^4 \sigma^2 S^3 A^2 \log^2 \iota\Big).$$

*Proof.* Define

$$y_t^{\text{emp}} := \sum_{(s,a) \neq (s,\mathring{\pi}(s))} \Delta(s,a)\, x_t(s,a), \qquad y_t^{\text{true}} := \sum_{(s,a) \neq (s,\mathring{\pi}(s))} \Delta(s,a)\, \rho^{\mathbb{P},\pi_t}(s,a).$$

If necessary, replacing $\Delta$ by its clipped version $\min\{\Delta, \mathcal{O}(H\sigma)\}$ only decreases the left-hand side of the self-bounding condition; we relabel this clipped gap as $\Delta$.

By Hölder's inequality with conjugate exponents $(\alpha, \alpha/(\alpha-1))$,

$$\sum_{(s,a) \neq (s,\mathring{\pi}(s))} x_t(s,a)^{1/\alpha} \leq \big(y_t^{\text{emp}}\big)^{1/\alpha} \omega_\Delta(\alpha)^{(\alpha-1)/\alpha}.$$

Applying Young's inequality with parameter $\gamma' > 0$, and using the epoch bound $N \le \mathcal{O}(SA \log T)$ together with $\sum_{t=t_i}^{t_{i+1}-1}(t - t_i + 1)^{-1} \le 1 + \log T$, gives

$$\mathbb{E}\Big[\mathcal{O}\Big(M \sum_i \sum_t \sigma(t - t_{i(t)} + 1)^{1/\alpha - 1} \sum_{(s,a)\neq(s,\mathring{\pi}(s))} x_t(s,a)^{1/\alpha}\Big)\Big]$$

$$\le \mathcal{O}\Big(\gamma' \mathbb{E}\Big[\sum_{t=1}^T y_t^{\mathrm{emp}}\Big] + \gamma'^{-\frac{1}{\alpha-1}} M^{\alpha/(\alpha-1)} \sigma^{\alpha/(\alpha-1)} \omega_\Delta(\alpha) SA \log^2 T\Big). \tag{71}$$

Since $M = M_{\mathrm{stab}} + M_{\mathrm{pen}}$ with $M_{\mathrm{stab}} = \mathcal{O}(H^{2-1/\alpha}S^{-1/\alpha}A^{-(2\alpha-1)/\alpha^2})$ and $M_{\mathrm{pen}} = \mathcal{O}(HS^{2-2/\alpha}A^{2-3/\alpha+1/\alpha^2})$ from Eq. (52), the basic inequality $(a+b)^p \le 2^p(a^p+b^p)$ for $p = \alpha/(\alpha-1) \ge 1$ yields $M^{\alpha/(\alpha-1)} \le \mathcal{O}(M_{\mathrm{stab}}^{\alpha/(\alpha-1)}+M_{\mathrm{pen}}^{\alpha/(\alpha-1)})$. We bound the two contributions separately.

*Penalty contribution.* Raising $M_{\mathrm{pen}}$ to the $\alpha/(\alpha-1)$-th power and multiplying by the extra $SA$ factor from the time-summation gives

$$M_{\mathrm{pen}}^{\alpha/(\alpha-1)} \cdot SA = \mathcal{O}\big(H^{\alpha/(\alpha-1)} \cdot S^3 \cdot A^{3-1/\alpha}\big).$$

*Stability contribution.* Raising $M_{\mathrm{stab}}$ to the same power and multiplying by $SA$ and using $S, A \ge 1$ to relax the smaller $S, A$ exponents up to 3 and $3 - 1/\alpha$ respectively,

$$M_{\mathrm{stab}}^{\alpha/(\alpha-1)} \cdot SA \le \mathcal{O}\big(H^{\alpha/(\alpha-1)+1} \cdot S^3 \cdot A^{3-1/\alpha}\big) \le \mathcal{O}\big(H^{\alpha/(\alpha-1)} \cdot H \cdot S^3 \cdot A^{3-1/\alpha}\big).$$

The extra factor $H$ from the stability contribution is propagated explicitly to the displayed $\Lambda_3$ in Eq. (68): $\Lambda_3 = \mathcal{O}(H^{(2\alpha-1)/(\alpha-1)}S^3 A^{3-1/\alpha}\sigma^{\alpha/(\alpha-1)}\omega_\Delta(\alpha)\log^2 T)$.

It remains to replace $y_t^{\mathrm{emp}}$ by $y_t^{\mathrm{true}}$. The residual occupancy bound of Jin et al. (2021, Appendices C.1–C.2), applied to $(\mathbb{P}, \widehat{\mathbb{P}}_{i(t)})$ under the same policy $\pi_t$, yields

$$\big|\rho^{\mathbb{P},\pi_t}(s,a) - x_t(s,a)\big| \le r_t(s,a) + 4 \sum_{h'=1}^{h(s)-1} \sum_{\substack{u\in\mathcal{S}_{h'},v\in\mathcal{A}\\w\in\mathcal{S}_{h'+1}}} \rho^{\mathbb{P},\pi_t}(u,v)\sqrt{\frac{\mathbb{P}[w \mid u,v]\log\iota}{\max\{m_{i(t)}(u,v),1\}}} q_t(s,a \mid w),$$

where $h(s)$ denotes the layer index of state $s$ in the layered MDP (so $s \in \mathcal{S}_{h(s)}$), $q_t(s,a \mid w)$ denotes the visitation probability of $(s,a)$ under policy $\pi_t$ and true transition $\mathbb{P}$ starting from state $w$. The error term $r_t$ collects the lower-order quadratic contributions from the layered Bernstein expansion and satisfies

$$\mathbb{E}\left[\sum_{t=1}^T \sum_{(s,a)\neq(s,\mathring{\pi}(s))} r_t(s,a)\right] \le \mathcal{O}(H^2 S^3 A^2 \log^2 \iota).$$

Since $\sum_t(y_t^{\mathrm{emp}} - y_t^{\mathrm{true}}) \le \sum_{t,(s,a)\neq(s,\mathring{\pi}(s))}\Delta(s,a)|x_t(s,a) - \rho^{\mathbb{P},\pi_t}(s,a)|$, multiplying by $\Delta(s,a) \le \mathcal{O}(H\sigma)$ and summing over suboptimal pairs gives

$$\mathbb{E}\Big[\sum_t(y_t^{\mathrm{emp}} - y_t^{\mathrm{true}})\Big] \le \mathcal{O}\Big(H\sigma\mathbb{E}[\mathbb{G}_3(\log\iota)] + H^3\sigma S^3 A^2 \log^2 \iota\Big).$$

Applying Lemma C.15 with $J = \log\iota$ and parameters $\alpha_0 = \alpha'/(cH\sigma\gamma')$, $\beta_0 = \beta'/(cH\sigma\gamma')$ for a sufficiently large universal constant $c$, and then multiplying by $\gamma'$, we obtain

$$\gamma'\mathbb{E}\Big[\sum_t(y_t^{\mathrm{emp}} - y_t^{\mathrm{true}})\Big] \le \mathcal{O}\Big((\alpha' + \beta')\big(\mathrm{Reg}_T(\mathring{\pi}) + C_{\mathrm{sb}}\big)$$

$$+ \alpha'^{-1}H^4\sigma^2 S^2 \omega_\Delta(2)\log\iota + \beta'^{-1}H^4\sigma^2 S^2 \Delta_{\min}^{-1}\log\iota + H^4\sigma^2 S^3 A^2 \log^2 \iota\Big), \tag{72}$$

where the inverse-$\alpha'$ coefficient $H^4\sigma^2 S^2 \omega_\Delta(2)\log\iota$ is obtained from the tight bound $\mathcal{O}(H^4\sigma^2 S\omega_\Delta(2)\log\iota)$ (derived directly from $8H^2 SJ/\alpha_0$ in Lemma C.15) by the elementary inequality $S \le S^2$, so that the displayed coefficient matches the $\Lambda_1$ definition stated in Eq. (68). We additionally used $\gamma' \le 1$ and $S \ge 1$.

Finally, Definition 3.1 gives $\mathbb{E}[\sum_t y_t^{\mathrm{true}}] \le \mathrm{Reg}_T(\mathring{\pi}) + C_{\mathrm{sb}}$. Combining this with Eq. (71) and Eq. (72) proves the lemma. $\qquad\square$

