# Best-of-Both-Worlds for Heavy-Tailed Markov Decision Processes

## Abstract

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

 5.1 is the first instance-dependent regret bound of algorithms for heavy-tailed MDPs with unknown transitions. Compared to the bounded loss setting, Theorem 5.1 recovers

previous results in Jin et al. (2021) and Ito et al. (2025) with respect to $T$ and suboptimal gap terms when we set $\alpha = 2$.

**Technical Innovations.** To analyze the regret bound for Algorithm 2, we derive an innovative regret decomposition scheme for HTMDPs with unknown transitions. Our novel pessimistic estimator allows us to decompose the regret into four terms: transition error, estimation error, skipping error, and pessimism error. This decomposition is different from previous works with similar algorithm framework (Jin et al., 2020; 2021; Ito et al., 2025), since the heavy-tailed losses and skipping tuning technique considered in heavy-tailed settings introduce extra skipping error terms. Controlling this term requires careful design of the transition kernel and UOB deviation term. To describe our innovative regret decomposition scheme for HTMDPs with unknown transitions, we first introduce the auxiliary loss function $\ell_t^{\text{pess}}$:

$$\ell_t^{\text{pess}}(s,a) := \ell_t(s,a) - D \cdot B_i(s,a). \quad (10)$$

This term is crucial for regret decomposition, since we need to carefully arrange the decomposition order of the extra operations we did in Line 8 of Algorithm 2 to construct the pessimistic estimator $\widehat{\ell}_t(s,a)$. Specifically, we have the following regret decomposition: $\text{Reg}_T(\mathring{\pi}) = \mathbb{E}\left[\sum_{t=1}^T V^{\mathbb{P},\pi_t}(s_1;\ell_t) - V^{\mathbb{P},\mathring{\pi}}(s_1;\ell_t)\right] = \text{TRANSERR} + \text{ESTREG} + \text{SKIPERR} + \text{PESSERR}$, where we denote these four terms as:

$$\text{TRANSERR} = \mathbb{E}\left[\sum_{t=1}^T V^{\mathbb{P},\pi_t}(s_1;\ell_t) - V^{\widehat{\mathbb{P}}_{i(t)},\pi_t}(s_1;\ell_t^{\text{pess}})\right],$$

$$\text{ESTREG} = \mathbb{E}\left[\sum_{t=1}^T V^{\widehat{\mathbb{P}}_{i(t)},\pi_t}(s_1;\widehat{\ell}_t) - V^{\widehat{\mathbb{P}}_{i(t)},\mathring{\pi}}(s_1;\widehat{\ell}_t)\right],$$

$$\text{SKIPERR} = \mathbb{E}\left[\sum_{t=1}^T V^{\widehat{\mathbb{P}}_{i(t)},\pi_t}(s_1;\ell_t^{\text{pess}} - \widehat{\ell}_t)\right.$$
$$\left. - V^{\widehat{\mathbb{P}}_{i(t)},\mathring{\pi}}(s_1;\ell_t^{\text{pess}} - \widehat{\ell}_t)\right],$$

$$\text{PESSERR} = \mathbb{E}\left[\sum_{t=1}^T V^{\widehat{\mathbb{P}}_{i(t)},\mathring{\pi}}(s_1;\ell_t^{\text{pess}}) - V^{\mathbb{P},\mathring{\pi}}(s_1;\ell_t)\right].$$

This decomposition order is crucial, since we can handle ESTREG by the similar methods we conducted for the known transition case, and handle PESSERR by the Bernstein-type concentration argument (given in Jin et al. (2020)). Therefore, we can focus on bounding the TRANSERR and SKIPERR terms. The TRANSERR term can be solved by applying similar techniques introduced by Jin et al. (2021). We highlight the procedure of bounding the TRANSERR term here.

We first note that the FTRL result $x_t$ is the occupancy measure $\rho^{\widehat{\mathbb{P}}_{i(t)},\pi_t}$, which does not match the true occupancy mea-

sure $\rho^{\mathbb{P},\pi_t}$ in the unknown transition case. Therefore, we can only handle the skipping error term SKIPERR by incorporating the skipping bonus to compensate for the skipping bias under the empirical model $\widehat{\mathbb{P}}_{i(t)}$ given the definition of $b_t^{\text{skip}}(s,a)$ in Eq. (8), which provides the high-level intuition of constructing this regret decomposition. Note that in Line 8 of Algorithm 2, the skipping loss $\ell_t^{\text{skip}}(s,a)$ is weighted by the optimistic importance sampling factor $u_t(s,a)$, which is not the exact occupancy $x_t(s,a)$ in the known transition case. Thus, we should involve the difference on UOB term when bounding the deviation between the "true" loss $\ell_t$ and estimated loss $\widehat{\ell}_t$.

**Lemma 5.2.** *With high probability, we have*

$$\text{SKIPERR} \le \mathbb{E}\left[\mathcal{O}\left(\sum_{t=1}^T \sigma t^{1/\alpha-1} \sum_{(s,a):a\neq\mathring{\pi}(s)} x_t(s,a)^{1/\alpha}\right)\right]$$
$$+ \mathbb{E}\left[\mathcal{O}\left(\sum_{t=1}^T \sum_{(s,a)} \sigma|u_t(s,a) - \rho^{\mathbb{P},\pi_t}(s,a)|\right)\right].$$

Here the first term is the desired form that excludes the reference action term $\mathring{\pi}(s)$, which can further derive instance-dependent and instance-independent regret bounds. The second term is the UOB deviation term, which can be bounded by the residual UOB argument in Jin et al. (2021). We refer to the formal version of this lemma in Lemma C.9.

# 6. Conclusion

In this paper, we present the first Best-of-Both-Worlds (BoBW) framework for episodic Markov decision processes with heavy-tailed feedback (HTMDPs). For settings with known transitions, we propose `HT-FTRL-OM`, an algorithm leveraging the FTRL framework with $1/\alpha$-Tsallis entropy regularization and a novel skipping loss estimator. By introducing innovative techniques such as local control for heavy-tailed shifted losses and suboptimal mass propagation, we establish that `HT-FTRL-OM` achieves nearly minimax-optimal instance-independent regret $\widetilde{\mathcal{O}}(T^{1/\alpha})$

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

*Proof.*

$$\mathbb{E}[\text{PessErr}] \leq \mathbb{E}\left[\left.\sum_{t=1}^{T} V^{\widehat{\mathbb{P}}_{i(t)},\mathring{\pi}}(s_1;\boldsymbol{\ell}_t^{\text{pess}}) - V^{\mathbb{P},\mathring{\pi}}(s_1;\boldsymbol{\ell}_t)\right|\mathcal{E}\right] + \mathbb{E}\left[\left.\sum_{t=1}^{T} V^{\widehat{\mathbb{P}}_{i(t)},\mathring{\pi}}(s_1;\boldsymbol{\ell}_t^{\text{pess}}) - V^{\mathbb{P},\mathring{\pi}}(s_1;\boldsymbol{\ell}_t)\right|\mathcal{E}^c\right]$$

$$\leq \mathbb{E}\left[\left.\sum_{t=1}^{T} \mathbb{E}[V^{\widehat{\mathbb{P}}_{i(t)},\mathring{\pi}}(s_1;\boldsymbol{\ell}_t^{\text{pess}}) \mid \mathcal{F}_{t-1}] - \mathbb{E}[V^{\mathbb{P},\mathring{\pi}}(s_1;\boldsymbol{\ell}_t) \mid \mathcal{F}_{t-1}]\right|\mathcal{E}\right] + O(H\sigma T\delta),$$

where the last inequality holds by Lemma C.3. Notice that under $\mathcal{E}$, by Lemma C.4, we have

$$\mathbb{E}[V^{\widehat{\mathbb{P}}_{i(t)},\pi}(s;\boldsymbol{\ell}_t^{\text{pess}}) \mid \mathcal{F}_{t-1}] \leq \mathbb{E}[V^{\mathbb{P},\pi}(s;\boldsymbol{\ell}_t) \mid \mathcal{F}_{t-1}].$$

which implies the result. $\qquad\square$

## C.4. Bounding FTRL Regret (EstReg)

Since the algorithm resets FTRL in each epoch, EstReg is the sum of regrets over epochs. We define $\text{EstReg}_i$ as the regret over epoch $i$, which defined as

$$\text{EstReg}_i = \mathbb{E}\left[\sum_{t=t_i}^{t_{i+1}-1} \left\langle \boldsymbol{\rho}^{\widehat{\mathbb{P}}_i,\pi_t} - \boldsymbol{\rho}^{\widehat{\mathbb{P}}_i,\mathring{\pi}}, \widehat{\boldsymbol{\ell}}_t \right\rangle\right]. \tag{47}$$

A key observation is that the learned occupancy measure $\boldsymbol{x}_t$ in Algorithm 2 is exactly $\rho^{\widehat{\mathbb{P}}_i,\pi_t}$. Therefore, we apply similar techniques presented in Appendix B to bound the $\text{EstReg}_i$.

We first consider the bound of $\widehat{\ell}_t(s,a)$:

**Lemma C.6.** *Under good event $\mathcal{E}$, we have*

$$|\widehat{\ell}_t(s,a)| \leq \left(C + C^{1-\alpha}\right) \cdot \sigma x_t(s,a)^{1/\alpha-1}(t - t_{i(t)} + 1)^{1/\alpha} + H\sigma\left(1 + C^{1-\alpha}S^{1-\alpha}\right)$$

*Proof.* By definition of $\widehat{\ell}_t(s,a)$, and definition of $\tau_t(s,a)$ and $b_t^{\text{skip}}(s,a)$ in Eqs. (7) and (8), we have

$$|\widehat{\ell}_t(s,a)| = \left|\left(\frac{\ell_t^{\text{skip}}(s,a)}{u_t(s,a)}\right) \cdot \mathbb{1}_t[(s,a)] - b_t^{\text{skip}}(s,a) - D \cdot B_{i(t)}(s,a)\right|$$

$$\leq \left|\frac{\ell_t^{\text{skip}}(s,a)}{u_t(s,a)}\right| + b_t^{\text{skip}}(s,a) + D \cdot B_{i(t)}(s,a)$$

$$\leq \left|\frac{\tau_t(s,a)}{x_t(s,a)}\right| + b_t^{\text{skip}}(s,a) + D$$

$$\leq \sigma x_t(s,a)^{1/\alpha-1} \cdot \left(C(t - t_{i(t)} + 1)^{1/\alpha} + C^{1-\alpha}(t - t_{i(t)} + 1)^{1/\alpha-1}\right) + H\sigma\left(1 + C^{1-\alpha}S^{1-\alpha}\right)$$

$$\leq \left(C + C^{1-\alpha}\right) \cdot \sigma x_t(s,a)^{1/\alpha-1}(t - t_{i(t)} + 1)^{1/\alpha} + H\sigma\left(1 + C^{1-\alpha}S^{1-\alpha}\right),$$

where the second inequality holds since $u_t(s,a) \geq x_t(s,a)$ under $\mathcal{E}$. $\qquad\square$

**Lemma C.7.** *If $\mathcal{E}$ does not hold, we have*

$$|\widehat{\ell}_t(s,a)| \leq O(t^2) \tag{48}$$

*Proof.* Notice that $u_t(s,a) \geq 1/St$ holds. Therefore, we can bound $|\widehat{\ell}_t(s,a)|$ by

$$|\widehat{\ell}_t(s,a)| \leq tS \cdot \tau_t(s,a) + b_t^{\text{skip}}(s,a) + D \leq \mathcal{O}(t^2)$$

$\qquad\square$

**Lemma C.8.** *We can control*

$$
\text{ESTREG}_i \leq O\left( H^{1-\frac{1}{\alpha}} \sum_{t=t_i}^{t_{i+1}-1} \sigma(t - t_{i(t)} + 1)^{1/\alpha - 1} \cdot \sum_{(s,a) \neq (s, \mathring{\pi}(s))} x_t(s,a)^{1/\alpha} \right)
$$

$$
+ O\left( \mathbb{E}\left[ H^2 \sigma^2 \sum_{t=t_i}^{t_{i+1}-1} \eta_t \sum_{(s,a) \in \mathcal{S} \times \mathcal{A}} x_t(s,a)^{2-1/\alpha} \left( B_i(s,a) \right)^2 \right] \right) \tag{49}
$$

$$
+ O\left( H^{1-\frac{1}{\alpha}} S^{2-\frac{1}{\alpha}} A^{5-\frac{5}{\alpha}} \sigma \log(t_{i+1} - t_i) + H\sigma S^{2-\alpha} A T^2 (t_{i+1} - t_i)\delta \right).
$$

*Proof.* First we consider the case when $\mathcal{E}$ does not happen. In this case, we can derive the naive bound:

$$
\mathbb{E}[\text{ESTREG}_i] = \mathbb{E}\left[ \sum_{t=t_i}^{t_{i+1}-1} \left\langle \rho^{\widehat{\mathbb{P}}_i, \pi_t} - \rho^{\widehat{\mathbb{P}}_i, \mathring{\pi}}, \widehat{\boldsymbol{\ell}}_t \right\rangle \right]
$$

$$
\leq \mathbb{E}\left[ \sum_{t=t_i}^{t_{i+1}-1} \sum_{(s,a) \in \mathcal{S} \times \mathcal{A}} \left( \rho^{\widehat{\mathbb{P}}_i, \pi_t}(s,a) + \rho^{\widehat{\mathbb{P}}_i, \mathring{\pi}}(s,a) \right) \cdot |\widehat{\ell}_t(s,a)| \right] \tag{50}
$$

$$
\leq 2SA(t_{i+1} - t_i) \cdot \left( D + (C + C^{1-\alpha})\sigma(t - t_{i(t)} + 1)^{1/\alpha + 1} \right)
$$

$$
\leq O(H\sigma S^{2-\alpha} A T^2 (t_{i+1} - t_i)).
$$

where the first inequality holds by Lemma C.7, and the second inequality is due to Lemma C.10 and ignoring the constant terms.

Next we consider the case under $\mathcal{E}$. Recall the definition of $\widehat{\ell}_t(s,a)$, which can be seen as three parts: the importance sampling skipping loss $\widetilde{\ell}_t(s,a) = \ell_t^{\text{skip}}(s,a) \mathbb{1}[(s,a) \text{ is visited}]/u_t(s,a)$, the skipping bonus term $b_t^{\text{skip}}(s,a)$, and the transition bonus term $DB_i(s,a)$. We first apply standard FTRL regret decomposition, and get:

$$
\sum_{t=t_i}^{t_{i+1}-1} \left\langle \boldsymbol{x}_t - \rho^{\widehat{\mathbb{P}}_i, \mathring{\pi}}, \widehat{\boldsymbol{\ell}}_t \right\rangle
$$

$$
\leq \underbrace{\sum_{t=t_i}^{t_{i+1}-1} \left\langle \boldsymbol{x}_t - \boldsymbol{x}_{t+1}, \widehat{\boldsymbol{\ell}}_t \right\rangle - D_{\Psi_t}(\boldsymbol{x}_{t+1}, \boldsymbol{x}_t)}_{\text{Stability Term}} + \underbrace{\sum_{t=t_i}^{t_{i+1}-1} \left( \Psi_t(\rho^{\mathbb{P}, \mathring{\pi}}) - \Psi_{t-1}(\rho^{\mathbb{P}, \mathring{\pi}}) \right) - \left( \Psi_t(\boldsymbol{x}_t) - \Psi_{t-1}(\boldsymbol{x}_t) \right)}_{\text{Penalty Term}} \tag{51}
$$

We further bound the stability term and penalty term separately.

**Bounding the Stability Term:** We apply similar methods in Lemma B.5. First we consider the shifted loss $\ell_t^{\text{shift}}(s,a)$ defined as:

$$
\ell_t^{\text{shift}}(s,a) := Q^{\widehat{\mathbb{P}}_{i(t)}, \pi_t}(s, a; \widetilde{\ell}_t(s,a)) - V^{\widehat{\mathbb{P}}_{i(t)}, \pi_t}(s; \widetilde{\ell}_t(s,a)),
$$

where $\widetilde{\ell}_t(s,a)$ is defined by

$$
\widetilde{\ell}_t(s, a) := \frac{\ell_t^{\text{skip}}(s,a) \cdot \mathbb{1}[(s,a) \text{ is visited}]}{u_t(s,a)}.
$$

Therefore, we have

$$
\sum_{t=t_i}^{t_{i+1}-1} \left\langle \boldsymbol{x}_t - \rho^{\widehat{\mathbb{P}}_{i(t)}, \mathring{\pi}}, \widehat{\boldsymbol{\ell}}_t \right\rangle
$$

$$
= \sum_{t=t_i}^{t_{i+1}-1} \sum_{(s,a) \in \mathcal{S} \times \mathcal{A}} (x_t(s,a) - \rho^{\widehat{\mathbb{P}}_{i(t)}, \mathring{\pi}}(s,a)) \cdot \left( \ell_t^{\text{shift}}(s,a) - b_t^{\text{skip}}(s,a) - D \cdot B_i(s,a) \right).
$$

Similar analysis in Lemma 4.2 shows that:

$$\left| \ell_t^{\text{shift}}(s,a) \right| \leq C(1 + HSA(1 + A)) \cdot \sigma x_t(s,a)^{1/\alpha - 1}(t - t_{i(t)} + 1)^{1/\alpha}$$

Therefore, by definition of $C$ and $\beta$ in Eq. (16) and Eq. (17), we have:

$$\alpha \eta_t \cdot \left( \ell_t^{\text{shift}}(s,a) - b_t^{\text{skip}}(s,a) - D \cdot B_i(s,a) \right)$$

$$\leq \alpha \beta \cdot (C(1 + HSA(1 + A^{1-1/\alpha})) + C^{1-\alpha}) \cdot x_t(s,a)^{1/\alpha - 1}$$

$$\leq \frac{\alpha - 1}{4\alpha^2} x_t(s,a)^{1/\alpha - 1}$$

Then we can apply the methods in Lemma B.5, we get:

$$\text{Stability Term} \leq \frac{8\alpha^2}{\alpha - 1} \sum_{t=t_i}^{t_{i+1}-1} \eta_t \sum_{(s,a) \in \mathcal{S} \times \mathcal{A}} x_t(s,a)^{2-1/\alpha} \left( \left( \ell_t^{\text{shift}}(s,a) \right)^2 + (b_t^{\text{skip}}(s,a))^2 + D^2 \left( B_i(s,a) \right)^2 \right).$$

Notice that by $u_t(s,a) \geq x_t(s,a)$ under $\mathcal{E}$, we can apply similar analysis in Lemma B.6 and get:

$$\mathbb{E}\left[ \eta_t \sum_{(s,a) \in \mathcal{S} \times \mathcal{A}} x_t(s,a)^{2-1/\alpha} \left( \ell_t^{\text{shift}}(s,a) \right)^2 \right] \leq O\left( H\beta C^{2-\alpha} \cdot \sigma(t - t_{i(t)} + 1)^{1/\alpha - 1} \cdot \sum_{(s,a) \neq (s, \hat{\pi}(s))} x_t(s,a)^{1/\alpha} \right). \tag{52}$$

Moreover, for the skipping bonus term, by similar argument in Lemma B.7, we have

$$\mathbb{E}\left[ \sum_{t=t_i}^{t_{i+1}-1} \eta_t \sum_{(s,a) \in \mathcal{S} \times \mathcal{A}} x_t(s,a)^{2-1/\alpha} \left( b_t^{\text{skip}}(s,a) \right)^2 \right] \leq O\left( SA^{1-1/\alpha} \beta C^{2-2\alpha} \cdot \sigma \left( \log(t_{i+1} - t_i) \right) \right). \tag{53}$$

Therefore, we get

$$\mathbb{E}\left[ \text{Stability Term} \right] \leq O\left( \sum_{t=t_i}^{t_{i+1}-1} H\beta C^{2-\alpha} \cdot \sigma(t - t_{i(t)} + 1)^{1/\alpha - 1} \cdot \sum_{(s,a) \neq (s, \hat{\pi}(s))} x_t(s,a)^{1/\alpha} \right)$$

$$+ O\left( \mathbb{E}\left[ D^2 \sum_{t=t_i}^{t_{i+1}-1} \eta_t \sum_{(s,a) \in \mathcal{S} \times \mathcal{A}} x_t(s,a)^{2-1/\alpha} \left( B_i(s,a) \right)^2 \right] \right) \tag{54}$$

$$+ O\left( SA^{1-1/\alpha} \beta C^{2-2\alpha} \cdot \sigma \log(t_{i+1} - t_i) \right).$$

We can bound the second term under $\mathcal{E}$ by Lemma C.11, and incorporate the definition of $C$ and $\beta$ in Eq. (16) and Eq. (17):

$$\mathbb{E}\left[ \text{Stability Term} \right] \leq O\left( H^{\frac{1}{\alpha}} \sum_{t=t_i}^{t_{i+1}-1} \sigma(t - t_{i(t)} + 1)^{1/\alpha - 1} \cdot \sum_{(s,a) \neq (s, \hat{\pi}(s))} x_t(s,a)^{1/\alpha} \right)$$

$$+ O\left( \mathbb{E}\left[ H^2 \sigma^2 \sum_{t=t_i}^{t_{i+1}-1} \eta_t \sum_{(s,a) \in \mathcal{S} \times \mathcal{A}} x_t(s,a)^{2-1/\alpha} \left( B_i(s,a) \right)^2 \right] \right) \tag{55}$$

$$+ O\left( H^{1-\frac{1}{\alpha}} S^{2-\frac{1}{\alpha}} A^{5-\frac{5}{\alpha}} \sigma \log(t_{i+1} - t_i) \right).$$

**Bounding the penalty term**  Penalty term is much easier. We apply the same analysis in Lemma B.9 and get

$$\mathbb{E}\left[ \text{Penalty Term} \right] \leq O\left( H^{\frac{2}{\alpha}-1} \cdot \sum_{t=t_i}^{t_{i+1}-1} \sigma(t - t_{i(t)} + 1)^{1/\alpha - 1} \cdot \sum_{(s,a) \neq (s, \hat{\pi}(s))} x_t(s,a)^{1/\alpha} \right) \tag{56}$$

Therefore, summing Eq. (50), Eq. (55), and Eq. (56) directly implies the result. $\qquad \square$

### C.5. Bounding Skipping Error (SKIPERR)

We further rewrite the Skipping Error term

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

Notice that by the doubling epoch argument (Jin et al., 2020), we have the epoch number $N \leq \mathcal{O}(SA\log(T))$. Then, the estimation regret shown in Lemma C.8 implies:

$$\mathbb{E}[\text{ESTREG}] = \mathbb{E}\left[\sum_{i=1}^{N} \text{ESTREG}_i \right]$$

$$\leq \mathbb{E}\left[ \mathcal{O}\left( H^{1-\frac{1}{\alpha}} \sum_{i=1}^{N} \sum_{t=t_i}^{t_{i+1}-1} \sigma(t - t_{i(t)} + 1)^{1/\alpha - 1} \cdot \sum_{(s,a)\neq(s,\mathring{\pi}(s))} x_t(s,a)^{1/\alpha} \right) \right]$$

$$+ \mathbb{E}\left[ \mathcal{O}\left( H^2\sigma^2 \sum_{t=1}^{T} \eta_t \sum_{(s,a)\in\mathcal{S}\times\mathcal{A}} x_t(s,a)^{2-1/\alpha} \left( B_i(s,a) \right)^2 \right) \right]$$

$$+ \mathcal{O}\left( H^{1-\frac{1}{\alpha}} S^{3-\frac{1}{\alpha}} A^{6-\frac{5}{\alpha}} \sigma \log(T)^2 + H\sigma S^{2-\alpha} AT^3 \delta \right).$$

$$\leq \mathcal{O}\left( \sigma H(SA)^{1-1/\alpha} T^{1/\alpha} \cdot SA\log(T) \right)$$

$$+ \mathcal{O}\left( H^4\sigma^2 S^3 A^2 \log(\iota)^2 + H^2\sigma^2 SAT\delta \right)$$

$$+ \mathcal{O}\left( H^{1-\frac{1}{\alpha}} S^{3-\frac{1}{\alpha}} A^{6-\frac{5}{\alpha}} \sigma \log(T)^2 + H\sigma S^{2-\alpha} AT^3 \delta \right).$$

where the last inequality is shown by Holder inequality similar to the proof of Theorem B.1. The skipping error shown in Lemma C.9, we have

$$\mathbb{E}[\text{SKIPERR}] \leq \mathbb{E}\left[ O\left( H^{\alpha+1-\frac{1}{\alpha}} A^{3-\frac{3}{\alpha}} \sum_{i=1}^{N} \sum_{t=t_i}^{t_{i+1}-1} \sigma(t - t_{i(t)} + 1)^{1/\alpha - 1} \cdot \sum_{(s,a)\neq(s,\mathring{\pi}(s))} x_t(s,a)^{1/\alpha} \right) \right]$$

$$+ \mathbb{E}\left[ 2\sigma \sum_{t=1}^{T} \sum_{(s,a)\in\mathcal{S}\times\mathcal{A}} \left| u_t(s,a) - \rho^{\mathbb{P},\pi_t}(s,a) \right| \right] + 4\delta \cdot \mathbb{E}\left[ \text{SKIPERR} \mid \mathcal{E}^c \right]$$

$$\leq \mathcal{O}\left(\sigma H^{\alpha+1}S^{1-\frac{1}{\alpha}}A^{4-\frac{4}{\alpha}}T^{1/\alpha} \cdot SA\log(T) + H\sigma S^{2-\alpha}AT^3\delta\right) + 2\sigma \cdot \mathbb{E}\left[\sum_{t=1}^{T}\sum_{(s,a)\in\mathcal{S}\times\mathcal{A}}|u_t(s,a) - \rho^{\mathbb{P},\pi_t}(s,a)|\right].$$

By the analysis in Appendix C.2 in Jin et al. (2021), we further have

$$\mathbb{E}\left[\sum_{t=1}^{T}\sum_{(s,a)\in\mathcal{S}\times\mathcal{A}}|u_t(s,a) - \rho^{\mathbb{P},\pi_t}(s,a)|\right] \leq \mathcal{O}(HS\sqrt{AT\log(\iota)} + H^2S^3A^2\log^2(\iota) + SAT\delta).$$

Therefore, we have

$$\mathbb{E}\left[\text{SKIPERR}\right] \leq \mathcal{O}\left(H^{\alpha+1}S^{2-\frac{1}{\alpha}}A^{5-\frac{4}{\alpha}} \cdot \sigma T^{1/\alpha}\log(T) + H\sigma S^{2-\alpha}AT^3\delta\right)$$
$$+ \mathcal{O}\left(H\sigma S\sqrt{AT\log(\iota)} + H^2\sigma S^3A^2\log^2(\iota) + \sigma SAT\delta\right).$$

By Lemma C.5, we have

$$\mathbb{E}[\text{PESSERR}] \leq O(\sigma T^2\delta + H\sigma S^{1-\alpha}T\delta).$$

Combine each term, we finally get the regret bound in adversarial regime:

$$\text{Reg}_T(\mathring{\pi}) \leq \widetilde{\mathcal{O}}\left(H^{\alpha+1}S^{2-\frac{1}{\alpha}}A^{5-\frac{4}{\alpha}} \cdot \sigma T^{1/\alpha} + H\sigma S\sqrt{AT}\right)$$

## C.7. Proof for Self-Bounding Regime

Similar to the analysis in adversarial regime, by setting $\delta = T^{-3}$, the regret when $\mathcal{E}$ does not hold is a constant that does not relate to $T$. Therefore, we only consider the case when $\mathcal{E}$ happens.

In self-bounding regime, we consider the $(\mathring{\pi}, \Delta, C_{\text{sb}})$ self-bounding environment (Definition 3.1). We first analyze the estimation regret term and skipping error term. By Lemmas C.8 and C.9 and similar arguments in Appendix C.6, we have

$$\mathbb{E}\left[\text{SKIPERR} + \text{ESTREG}\right] \leq \mathbb{E}\left[\mathcal{O}\left(H^{\alpha+1-\frac{1}{\alpha}}A^{3-\frac{3}{\alpha}} \cdot \sum_{i=1}^{N}\sum_{t=t_i}^{t_{i+1}-1}\sigma(t-t_{i(t)}+1)^{1/\alpha-1} \cdot \sum_{(s,a)\neq(s,\mathring{\pi}(s))}x_t(s,a)^{1/\alpha}\right)\right]$$
$$+ 2\sigma \cdot \mathbb{E}\left[\sum_{t=1}^{T}\sum_{(s,a)\in\mathcal{S}\times\mathcal{A}}|u_t(s,a) - \rho^{\mathbb{P},\pi_t}(s,a)|\right]$$
$$+ \mathcal{O}\left(H^2\sigma S^{2-\alpha}AT^3\delta + H^4\sigma^2 S^3A^2\log(\iota)^2 + H^{1-\frac{1}{\alpha}}S^{3-\frac{1}{\alpha}}A^{6-\frac{5}{\alpha}}\sigma\log(T)^2\right)$$

By similar argument in the proof of Appendix B.6, we have for any constant $\gamma' > 0$, set $M = \mathcal{O}(H^{\alpha+1-\frac{1}{\alpha}}A^{3-\frac{3}{\alpha}})$

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

Combining these bounds, when $\alpha' + \beta' + \gamma' \leq 1/2$ (which holds when $C_{\mathrm{sb}}$ is not too small), we obtain:

$$\mathrm{Reg}_T(\mathring{\pi}) \leq 2\kappa + 4\sqrt{\Lambda_1 C_{\mathrm{sb}}} + 4\sqrt{\Lambda_2 C_{\mathrm{sb}}} + 4C_{\mathrm{sb}}^{1/\alpha}\Lambda_3^{\frac{\alpha-1}{\alpha}}.$$

**Final Bound.** Combining both cases, we have

$$\mathrm{Reg}_T(\mathring{\pi}) \leq \mathcal{O}\left(\kappa + \Lambda_1 + \Lambda_2 + \Lambda_3 + \sqrt{\Lambda_1 C_{\mathrm{sb}}} + \sqrt{\Lambda_2 C_{\mathrm{sb}}} + C_{\mathrm{sb}}^{1/\alpha}\Lambda_3^{\frac{\alpha-1}{\alpha}}\right).$$

Substituting the definitions of $\kappa, \Lambda_1, \Lambda_2, \Lambda_3$, we obtain the final self-bounding regret bound:

$$\begin{aligned}
\mathrm{Reg}_T(\mathring{\pi}) \leq \mathcal{O}\Big(&H^4\sigma^2 S^3 A^2 \log^2(\iota) \\
&+ H^4\sigma^2 S^2 \omega_\Delta(2)\log(\iota) + H^4\sigma^2 S^2 \Delta_{\min}^{-1}\log(\iota) + H^{\frac{\alpha^2}{\alpha-1}+1}S^2 A^4 \cdot \sigma^{\frac{\alpha}{\alpha-1}}\omega_\Delta(\alpha)\log(T)^2 \\
&+ H^2\sigma S\sqrt{\omega_\Delta(2)C_{\mathrm{sb}}\log(\iota)} + H^2\sigma S\sqrt{\Delta_{\min}^{-1}C_{\mathrm{sb}}\log(\iota)} \\
&+ C_{\mathrm{sb}}^{1/\alpha}\cdot H^{\alpha+1-\frac{1}{\alpha}}S^{2-\frac{2}{\alpha}}A^{4-\frac{4}{\alpha}}\cdot\sigma\cdot\omega_\Delta(\alpha)^{\frac{\alpha-1}{\alpha}}\log^{\frac{\alpha-1}{\alpha}}(T)\Big).
\end{aligned}$$

## C.8. Auxiliary Lemmas

**Lemma C.10** (Lemma C.1.2 in Jin et al. (2021)). *Algorithm 2 ensures $u_t(s) \geq 1/St$ for all $t$ and $s$.*

**Lemma C.11** (Lemma C.1.8 in Jin et al. (2021)).

$$\mathbb{E}\left[\sum_{t=1}^T \sum_{(s,a)\in\mathcal{S}\times\mathcal{A}} x_t(s,a)\cdot(B_{i(t)}(s,a))^2\right] = O\left(H^2 S^3 A^2 \log(\iota)^2 + SAT\delta\right), \tag{64}$$

*where $\log(\iota) = \log(HSAT/\delta)$.*

We then involve the definitions of self-bounding terms in Jin et al. (2021) for analyzing the self-bounding regime:

**Definition C.12** (Self-bounding Terms). For some mapping $\mathring{\pi} : S \to A$, define the following:

$$\mathbb{G}_1(J) = \sum_{t=1}^{T} \sum_{(s,a)\neq(s,\mathring{\pi}(s))} \rho^{\mathbb{P},\pi_t}(s,a) \cdot \frac{J}{\max\left\{m_{i(t)}(s,a),1\right\}}$$

$$\mathbb{G}_2(J) = \sum_{t=1}^{T} \sum_{(s,a)=(s,\mathring{\pi}(s))} (\rho^{\mathbb{P},\pi_t}(s,a) - \rho^{\mathbb{P},\mathring{\pi}}(s,a)) \cdot \frac{J}{\max\left\{m_{i(t)}(s,a),1\right\}}$$

$$\mathbb{G}_3(J) = \sum_{t=1}^{T} \sum_{s\in\mathcal{S}_h,a\neq\mathring{\pi}(s)} \sum_{h'=1}^{h-1} \sum_{u\in\mathcal{S}_{h'},v\in\mathcal{A},w\in\mathcal{S}_{h'+1}} \rho^{\mathbb{P},\pi_t}(u,v) \cdot \frac{\mathbb{P}[w \mid u,v] \cdot J}{\max\{m_{i(t)}(u,v),1\}} q_t(s,a \mid w),$$

where $q_t(s,a \mid w)$ is the probability that starting from state $w$ and visit $(s,a)$ under transition $\mathbb{P}$ and policy $\pi_t$.

Jin et al. (2021) then gives the bounds for these self-bounding terms:

**Lemma C.13** (Lemma D.2.2 in Jin et al. (2021)). *Under Definition 3.1, for any $\alpha \in \mathbb{R}_+$, we have*

$$\mathbb{E}[\mathbb{G}_1(J)] \leq \alpha \cdot (\mathrm{Reg}_T(\mathring{\pi}) + C_{\mathrm{sb}}) + \frac{1}{\alpha} \sum_{(s,a)\neq(s,\mathring{\pi}(s))} \frac{8J}{\Delta(s,a)}.$$

**Lemma C.14** (Lemma D.2.3 in Jin et al. (2021)). *Under Definition 3.1, for any $\beta \in \mathbb{R}_+$, we have*

$$\mathbb{E}[\mathbb{G}_2(J)] \leq \beta \cdot (\mathrm{Reg}_T(\mathring{\pi}) + C_{\mathrm{sb}}) + \frac{1}{\beta} \frac{8SHJ}{\Delta_{\min}}.$$

**Lemma C.15** (Lemma D.2.4 in Jin et al. (2021)). *Under Definition 3.1, for any $\alpha, \beta \in \mathbb{R}_+$, we have*

$$\mathbb{E}[\mathbb{G}_3(J)] \leq (\alpha + \beta) \cdot (\mathrm{Reg}_T(\mathring{\pi}) + C_{\mathrm{sb}}) + \frac{1}{\alpha} \sum_{(s,a)\neq(s,\mathring{\pi}(s))} \frac{8H^2SJ}{\Delta(s,a)} + \frac{1}{\beta} \cdot \frac{8H^2S^2J}{\Delta_{\min}}.$$