# OpenReview forum: "Best-of-Both-Worlds for Heavy-Tailed Markov Decision Processes"
_ICML.cc/2026/Conference — ICML 2026 regular_

### Official Review · Reviewer_7XJk · 2026-03-02

**Soundness:** 2
**Presentation:** 3
**Significance:** 3
**Originality:** 4
**Overall Recommendation:** 4
**Confidence:** 4

**Summary:**

This paper proposes two algorithms for minimizing regret in episodic MDPs with heavy-tailed feedback, where the loss distributions are assumed to have bounded $\alpha$-th raw moments. The authors study both the known-transition and unknown-transition settings. When the transition dynamics are known, Algorithm 1 achieves an instance-independent regret bound of $\tilde{O}(T^{1/\alpha})$ in adversarial regimes, and an instance-dependent logarithmic regret of $O(\log T)$ in stochastic regimes. When the transition dynamics are unknown, Algorithm 2 achieves an instance-independent regret bound of $\tilde{O}(T^{1/\alpha} + \sqrt{T})$ in adversarial regimes, and an instance-dependent regret bound of $O(\log^2 T)$ in stochastic regimes.

**Compliance With Llm Reviewing Policy:**

Affirmed.

**Final Justification:**

The rebuttal has fully addressed my most concerns and questions, especially the statements related to the classical lower bounds. Therefore, I would like to raise my score from 3 to 4.

**Key Questions For Authors:**

1. **Convexity of the Occupancy Measure Set**

   Why is the occupancy measure set $\mathcal{Q}(P)$ defined in Eq. (1) a convex polytope? A brief explanation or reference clarifying the linear constraints that induce this polyhedral structure would improve readability.

2. **Motivation for the Adversarial and Self-Bounding Regimes**

   What are the practical motivations or representative applications for considering the adversarial and self-bounding regimes in heavy-tailed MDPs? It would be helpful if the authors could elaborate on concrete scenarios where each regime is meaningful and explain the significance of analyzing both regimes in this setting.

3. **Relationship Between the Self-Bounding and Stochastic Regimes**

   What is the precise relationship between the self-bounding regime and the stochastic regime? The current presentation suggests that the stochastic regime is a special case of the self-bounding regime, but this connection is not stated explicitly. A clearer statement explaining how the regret bound in the self-bounding regime implies the bound in the stochastic regime would help readers better understand the logical flow.

4. **Minimax-Optimality Claim for Algorithm 1 (Known Transitions)**

   The paper claims that Algorithm 1 achieves a nearly minimax-optimal regret bound in adversarial regimes when the transition dynamics are known. However, no explicit lower bound is stated or proved under the same heavy-tailed adversarial MDP setting.

   The claim appears to rely on known heavy-tailed bandit lower bounds (e.g., Bubeck et al., 2013), but the paper does not explicitly provide a reduction argument or a formal justification showing that these lower bounds directly apply to the heavy-tailed MDP setting considered here. It would improve clarity and rigor if the authors could include a formal reduction or an explicit lower-bound statement establishing the matching rate in the adversarial regime.

5. **Minimax-Optimality Claim for Algorithm 2 (Unknown Transitions)**

   Similarly, the paper claims that Algorithm 2 achieves a nearly minimax-optimal regret bound in adversarial regimes when the transition dynamics are unknown. Again, no explicit lower bound is provided under the adversarial heavy-tailed MDP setting with unknown transitions.

   The argument seems to rely on lower bounds from Zhang & Sui (2021), but that work focuses on stochastic environments rather than adversarial regimes. It would strengthen the claim if the authors could provide a formal reduction argument or a rigorous lower-bound proof tailored to the adversarial setting with unknown transitions.

6. **Change in Instance-Dependent Regret Between Known and Unknown Transitions**

   When the transition is unknown, Algorithm 2 achieves a regret bound of $\tilde{O}(T^{1/\alpha} + \sqrt{T})$ in the adversarial regime, compared to $\tilde{O}(T^{1/\alpha})$ when the transition is known. The additional $\sqrt{T}$ term due to transition estimation is understandable.

   However, in the stochastic/self-bounding regime, the instance-dependent regret changes from $O(\log T)$ (known transitions) to $O(\log^2 T)$ (unknown transitions). It would be helpful if the authors could provide intuition for this extra logarithmic factor.

7. **Implications of the Instance-Dependent Regret Bound in the Stochastic Regime**

   Does achieving the instance-dependent logarithmic regret bound require additional structural assumptions on the MDP, such as gap conditions or self-bounding properties? More specifically, can the authors provide a clear statement of the form: if the MDP satisfies certain structural conditions, then the algorithm guarantees a logarithmic regret bound?

   If such conditions are indeed necessary, it would be helpful to explicitly state them and clarify their role in the analysis. Additionally, a discussion of whether these assumptions are realistic, verifiable, or likely to hold in practical applications would further strengthen the paper.

**Limitations:**

First, the instance-dependent logarithmic regret guarantees rely on structural conditions such as gap assumptions or self-bounding properties. These conditions may not hold in many practical MDPs. A clearer discussion of when such assumptions are realistic or verifiable in practice would improve the transparency of the contribution.

Second, the claim of near-minimax optimality in adversarial regimes is not accompanied by explicit lower bounds under the same heavy-tailed MDP setting.

Third, the heavy-tailed assumption is motivated by robustness considerations, but the paper does not discuss significance of considering the adversarial setting.

**Strengths And Weaknesses:**

**Strengths**

- To the best of my knowledge, this is the first work that studies adversarial regimes together with best-of-both-worlds (BoBW) guarantees in the heavy-tailed online decision-making setting.
- The theoretical analysis addresses several technical challenges inherent to heavy-tailed MDPs by adapting existing techniques and developing new analytical tools.
- The paper provides a comprehensive discussion of related work, and the overall presentation is clear and well-structured.

**Weaknesses**

- The paper appears to overstate some of its contributions. In particular, the authors claim that their algorithms achieve nearly minimax-optimal regret in adversarial regimes, both when the transition dynamics are known and unknown. However, the paper does not provide a clear matching lower bound under the same heavy-tailed adversarial setting. Without an explicit lower-bound argument or a precise reduction to existing results, the minimax-optimality claim is not fully substantiated.

---

### Official Review · Reviewer_cGnx · 2026-03-12

**Soundness:** 2
**Presentation:** 3
**Significance:** 2
**Originality:** 2
**Overall Recommendation:** 4
**Confidence:** 4

**Summary:**

This paper studies episodic heavy-tailed MDPs under an $\alpha$-moment assumption on losses, aiming for best-of-both-worlds guarantees across adversarial and self-bounding/stochastic regimes. For the known-transition setting, it proposes HT-FTRL-OM, which combines occupancy-measure FTRL with Tsallis regularization and a skipping/truncation-based loss estimator. For the unknown-transition setting, it proposes HT-FTRL-UOB, which adds empirical transitions, pessimistic bonuses, and upper occupancy bounds. The paper claims $\widetilde O(T^{1/\alpha})$ adversarial regret and $O(\log T)$ self-bounding regret in the known-transition case, and $\widetilde O(T^{1/\alpha}+\sqrt T)$ adversarial regret and $O(\log^2 T)$ self-bounding regret in the unknown-transition case. The main technical ingredients are a heavy-tailed shifted-loss analysis, a suboptimal-mass propagation lemma, and a regret decomposition for the unknown-transition setting.

**Compliance With Llm Reviewing Policy:**

Affirmed.

**Final Justification:**

The responses on Lemmas C.8 and C.9 largely address my main soundness concerns, so I am willing to raise my score from 3 to 4.

**Key Questions For Authors:**

1. In Lemma C.8, can the authors provide a complete derivation of the small-gradient condition needed for the local Tsallis-stability lemma, explicitly retaining the $D\,B_i(s,a)$ term? A rigorous fix here would substantially improve my soundness assessment.

2. In Lemma C.9, how is $\rho_{\hat P_{i(t)},\pi^\circ}$ controlled using $u_t$, given that $u_t$ is defined as an upper occupancy bound for $\pi_t$? If there is an omitted property of the Comp-UOB procedure or a different inequality should be used, please state it explicitly. Resolving this would materially affect my view of the unknown-transition result.

3. Can the authors provide a concrete empirical or semi-synthetic benchmark in which episodic heavy-tailed losses are genuinely present? This would help clarify whether the setting is mainly of theoretical interest or also practically relevant, and it would improve my assessment of significance.

4. How essential is the tabular assumption to the arguments? Are the main obstacles to extending the approach to linear or function-approximation settings conceptual or merely technical? A convincing discussion would improve my view of the broader impact of the work.

**Limitations:**

No. The paper should more clearly discuss the limitations of focusing only on tabular MDPs, the lack of empirical validation or concrete heavy-tailed RL benchmarks (at least showing that they exist), and the dependence on known heavy-tail parameters and occupancy-measure optimization.

**Strengths And Weaknesses:**

The paper tackles an interesting problem at the intersection of heavy-tailed online learning and best-of-both-worlds RL. Extending BoBW guarantees to heavy-tailed episodic MDPs is a natural question, and the occupancy-measure FTRL viewpoint is a reasonable starting point. The paper is also ambitious in covering both known and unknown transitions, and the high-level decomposition is consistent with the existing BoBW-MDP literature.

My main concern is soundness. In the unknown-transition analysis, I do not think the proof of the main theorem is currently complete. In Lemma C.8, the local Tsallis-stability argument seems to require a pointwise “small gradient” bound on
$$
\alpha \eta_t\bigl(\ell_t^{\mathrm{shift}}(s,a)-b_t^{\mathrm{skip}}(s,a)-D\,B_i(s,a)\bigr).
$$
However, on p. 33 around lines 1765--1769, the displayed bound appears to account only for the shifted-loss and skipping-bonus terms, while the $D\,B_i(s,a)$ term effectively disappears. I do not see a deterministic inequality that upper-bounds $D\,B_i(s,a)$ by the required occupancy-scaled quantity $x_t(s,a)^{1/\alpha-1}$. Since $B_i$ is defined from count-based confidence widths rather than the occupancy $x_t$, this alignment is not immediate. Because this condition is exactly what is needed to invoke the local stability lemma, the ESTREG proof for the unknown-transition case appears incomplete.

A second soundness issue appears in Lemma C.9. Around p. 34, lines 1848--1855, the proof turns a term involving
$$
\frac{u_t(s,a)-\rho_{P,\pi_t}(s,a)}{u_t(s,a)}\Bigl(x_t(s,a)-\rho_{\hat P_{i(t)},\pi^\circ}(s,a)\Bigr)
$$
into a bound proportional to $\sum_{s,a}|u_t(s,a)-\rho_{P,\pi_t}(s,a)|$. I do not see why this is valid. The quantity $u_t$ is introduced as an upper occupancy bound for the policy $\pi_t$, not for the reference policy $\pi^\circ$. So I do not see how $\rho_{\hat P_{i(t)},\pi^\circ}$ is controlled by $u_t$, and hence I do not see why the multiplicative factor is bounded by a universal constant. Unless an additional property of the UOB construction is missing from the proof, this part also looks incomplete.

On presentation, the paper is understandable at a high level and includes pseudocode, but the most delicate parts of the proof are written too tersely. Several places in the unknown-transition analysis rely on phrases like “similar analysis” even though extra terms are introduced. This makes the paper difficult to verify, especially when the contribution is primarily theoretical.

On significance, the problem is important, but the practical case is underdeveloped. The paper motivates heavy tails through domains like finance and networking, yet it does not present a concrete RL benchmark or dataset where episodic heavy-tailed losses actually arise. I understand the lack of experiment is acceptable in an RL theory paper, but it is important to see the setting actually taking place in real world problems. In addition, the whole paper is limited to the tabular setting. Given that prior heavy-tailed RL work already considers function approximation in stochastic settings, the tabular restriction substantially narrows the scope.

On originality, the combination of occupancy-measure FTRL with truncation/skipping for heavy-tailed losses is reasonable, but to me the contribution feels more incremental than fundamental. Algorithmically, the paper largely wraps existing BoBW-FTRL templates with heavy-tail truncation/skipping machinery and corresponding bonuses. That can still be publishable if the theory is clean and compelling, but here the proof issues and lack of practical evidence make the overall contribution less convincing.

Overall, I see clear merit in the problem choice, but at present the weaknesses outweigh the strengths. That said, I can raise the score if my concerns are addressed.

---

### Official Review · Reviewer_JsX2 · 2026-03-12

**Soundness:** 3
**Presentation:** 2
**Significance:** 2
**Originality:** 3
**Overall Recommendation:** 4
**Confidence:** 2

**Summary:**

This paper studies episodic MDPs with heavy-tailed losses. It proposes two FTRL-based algorithms, one for the known-transition setting and the other for the unknown-transition setting, and shows that both achieve best-of-both-worlds guarantees. To obtain these results, the paper develops several novel technical ideas to address the challenges posed by heavy-tailed losses. The resulting regret bounds are near-optimal, and they also recover several existing results with respect to $T$ in the literature.

**Compliance With Llm Reviewing Policy:**

Affirmed.

**Final Justification:**

The rebuttal addresses my main concerns, and I will keep my current evaluation.

**Key Questions For Authors:**

Key questions:

1. Can you compare the dependence on $H, S, A$ in the results with the bounded-loss BoBW results?
2. Is the self-bounding condition from the bounded-loss BoBW literature suitable for the heavy-tailed environments?

Minor comments:

1. The paper uses the terms heavy-tailed feedback and heavy-tailed losses in different places. For clarity and consistency, it may be preferable to use a single term throughout.

2. The notation $\Delta$ for the gap function is overloaded and may be confusing to readers.

3. In Section 4.1, it would be helpful to mention explicitly that the skipping loss estimator is introduced shortly afterward in Section 4.2.

**Limitations:**

There is no discussion on practical side of the algorithms, such as the computational complexity of solving a convex program over the occupancy measure polytope.

**Strengths And Weaknesses:**

Strengths:
- **Soundness**: This paper appears technically solid.
- **Presentation**: At a high level, this paper is structured clearly and the main contributions are easy to find.
- **Significance**: This paper extends the best-of-both-worlds guarantees to episodic MDPs with heavy-tailed losses, which is a meaningful problem in the theory of RL.
- **Originality**: This paper develop several novel techniques when handling the setting of episodic MDPs.

Weakness:
- **Soundness**: The regret bounds hide potentially very large polynomial factors in these quantities behind $\text{poly}(H,S,A)$ notation. The paper does not discuss whether these dependencies are tight or improvable, nor does it compare them against the $H, S, A$ factors in the bounded-loss BoBW results.
- **Presentation**: This paper is quite technically dense.
- **Significance**: The significance is primarily theoretical, with limited discussion of computational or practical implications. No experiments.

---

### Official Review · Reviewer_Ugbw · 2026-03-12

**Soundness:** 2
**Presentation:** 2
**Significance:** 3
**Originality:** 3
**Overall Recommendation:** 3
**Confidence:** 3

**Summary:**

This paper gives a policy for MDPs that has an uniform guarantee in two environments: if the environment is adversarial, the policy achieves regret of O(T^1/alpha), if the environment is self-bounding (possibly unbounded, with heavy tail losses), it achieves regret of O(log T). Another policy is presented when the transition probabilities are unknown, with similar guarantees.

**Compliance With Llm Reviewing Policy:**

Affirmed.

**Key Questions For Authors:**

Happy to reconsider if the above issues are addressed.

**Limitations:**

Yes

**Strengths And Weaknesses:**

I think there's an important assumption missing: as presented, the adversarial regime allows unbounded losses: the loss only has to be F_{t-1}-measurable. Another problem: Lemma 4.3 doesn't have conditional expectation but Lemma B.3 has a conditional expectation. There are some steps missing.

Presentation could be improved: e.g., sometimes, the state is denoted s^t_h, other times, it's s_h. Why is the episode index t missing?

Significance: this paper considers an important problem because unbounded losses or heavy-tail losses do appear in practice.

Originality: There is good potential for originality that can be gleaned once the soundness and presentation are improved.

---

### Decision · Program_Chairs · 2026-04-30

**Decision:**

Accept (regular)

**Comment:**

This paper proposes two FTRL-based algorithms for episodic MDPs with heavy-tailed losses, achieving best-of-both-worlds guarantees under both known and unknown transitions. Reviewers agreed this is the first work achieving BoBW guarantees under heavy-tailed feedback, and that the technical contributions are non-trivial and of value to the community.

Three main concerns were raised during review. First, two gaps were identified in Lemmas C.8 and C.9. The authors provided fixes, restructuring the Tsallis-stability argument and introducing a mild truncated nonnegativity assumption, which reviewers found satisfactory. Second, a reviewer noted missing explicit lower-bound statements; the authors clarified these follow from standard reductions and committed to adding them, satisfying the reviewer. Third, all reviewers noted suboptimal polynomial dependence on H, S, A relative to bounded-loss baselines, which the authors attributed to the shifted-loss analysis and acknowledged as an open problem.

Three of four reviewers converged on weak accept after the rebuttal. The remaining weak reject reflects presentation concerns rather than fundamental flaws. The paper is technically sound and advances a well-motivated research direction.

I recommend acceptance conditional on the revision incorporating the corrected proofs for Lemmas C.8 and C.9 clearly into the paper with the truncated nonnegativity assumption stated explicitly in the main text; formal lower-bound statements; and the notation and presentation fixes agreed upon during the discussion.